# Evolution of Discriminator and Generator Gradients in GAN Training: From Fitting to Collapse

**Weiguo Gao**  *wggao@fudan.edu.cn*
*School of Mathematical Sciences & School of Data Science, Fudan University*
*Shanghai Key Laboratory of Contemporary Applied Mathematics*

**Ming Li**  *mingli23@m.fudan.edu.cn*
*School of Mathematical Sciences, Fudan University*

**Reviewed on OpenReview:** *https://openreview.net/forum?id=58gPkcVbFL*

## Abstract

Generative Adversarial Networks (GANs) are powerful generative models but often suffer from mode mixture and mode collapse. We propose a perspective that views GAN training as a two-phase progression from fitting to collapse, where mode mixture and mode collapse are treated as inter-connected. Inspired by the particle model interpretation of GANs, we leverage the *discriminator gradient* to analyze particle movement and the *generator gradient*, specifically "steepness," to quantify the severity of mode mixture by measuring the generator's sensitivity to changes in the latent space. Using these theoretical insights into *evolution of gradients*, we design a specialized metric that integrates both gradients to detect the transition from fitting to collapse. This metric forms the basis of an early stopping algorithm, which stops training at a point that retains sample quality and diversity. Experiments on synthetic and real-world datasets, including MNIST, Fashion MNIST, and CIFAR-10, validate our theoretical findings and demonstrate the effectiveness of the proposed algorithm.

## 1 Introduction

Generative Adversarial Networks (GANs) serve as a popular technique for unsupervisedly learning generative models of structured and complicated data (Goodfellow et al., 2014; Nowozin et al., 2016; Arjovsky et al., 2017; Goodfellow, 2017; Li et al., 2017; Nguyen et al., 2017; Ghosh et al., 2018; Luo & Yang, 2024). GANs typically involve a generator that generates samples resembling real samples, and a discriminator that differentiates between real and generated samples. Through adversarial training, the generator learns to produce increasingly realistic samples, while the discriminator enhances its ability to distinguish them, resulting in refined models.

One of the primary challenges in training GANs is fine-tuning the interactive dynamics between the generator and the discriminator. If these dynamics are not well-aligned, several problematic behaviors can arise. Among the most common issues are mode collapse (Goodfellow, 2017) and mode mixture (An et al., 2020). Mode collapse occurs when the generator produces limited varieties of samples, collapsing to very few modes, while mode mixture involves blending distinct modes, resulting in unrealistic or ambiguous outputs. Numerous GAN variants have been proposed to address mode collapse (Nowozin et al., 2016; Arjovsky et al., 2017; Li et al., 2017; Nguyen et al., 2017; Ghosh et al., 2018; Luo & Yang, 2024), alongside theoretical insights (Sun et al., 2020; Becker et al., 2022). For mode mixture, research has focused on mitigation strategies, particularly within the framework of optimal transport (Lei et al., 2019; An et al., 2020; Gu et al., 2021) and rejection sampling (Tanielian et al., 2020).

Despite extensive research on GAN training, current studies share two common limitations: (i) mode collapse (Goodfellow, 2017) and mode mixture (An et al., 2020; Tanielian et al., 2020) are typically treated

as separate, independent issues, and (ii) mode collapse is frequently viewed as an indicator of training failure (Arjovsky et al., 2017; Luo & Yang, 2024), and the current literature primarily focuses on techniques to prevent or mitigate this phenomenon (Nowozin et al., 2016; Arjovsky et al., 2017; Li et al., 2017; Nguyen et al., 2017; Ghosh et al., 2018; Luo & Yang, 2024). In contrast, we propose a perspective that views GAN training as a two-phase progression *from fitting to collapse.* In this view, mode mixture is an integral part of the fitting phase, where the generator increasingly aligns with the real data distribution. During this alignment, some generated samples may fall outside the modes, reflecting the presence of mode mixture. Mode collapse does not signify outright failure. Instead, it occurs when the model tries to alleviate mode mixture. In this process, generated samples are pushed away from the modes, leading to a loss of diversity. By identifying the transition from fitting to collapse, early stopping can prevent the loss of diversity while preserving sample quality.

To illustrate this progression, we train the Non-Saturating GAN (NSGAN) (Goodfellow et al., 2014) on a 3-dimensional Gaussian mixture dataset and MNIST (LeCun et al., 1998), recording generated samples as shown in fig. 1. In the first row, the orange dots represent real samples drawn from the Gaussian mixture, while the blue dots show generated samples. Initially, the generated samples cluster near the origin (subfigure 1). As training progresses, these samples spread out and align with the modes of the real distribution (subfigure 2–3). This marks the fitting phase, where the generator increasingly captures the structure of the real distribution. Nevertheless, during this phase, some generated samples may fall outside the modes, indicating the presence of mode mixture. As training continues, the generated samples start to collapse into fewer modes (subfigure 3–6), eventually leading to a severe loss of diversity. This *collapse* phase is evident in the Gaussian mixture dataset, highlighting the need for early stopping to preserve diversity. In the second row, we map both real MNIST images and generated images into a common 3-dimensional space using UMAP (Uniform Manifold Approximation and Projection) (McInnes et al., 2018), where we observe a similar progression.

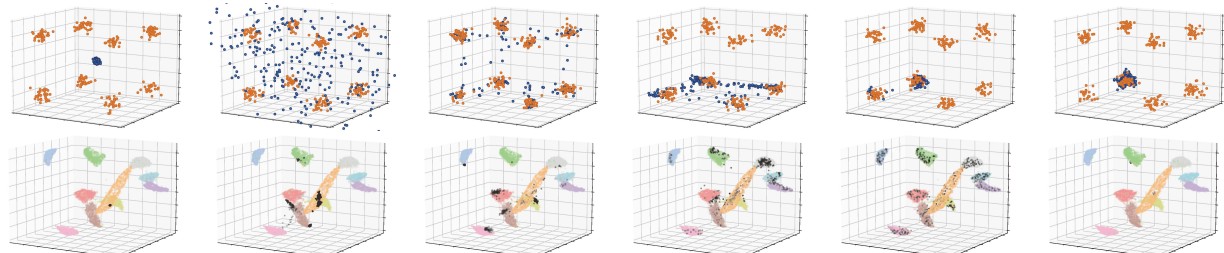

Figure 1: The real and generated samples by training NSGAN on a 3-dimensional Gaussian mixture dataset and MNIST. **First row: Gaussian mixture dataset.** Orange: Real samples. Blue: Generated samples. Epochs from left to right: 0, 15, 60, 450, 850, 980. Initially, the generated samples cluster near the origin, then spread out and align with the real modes. However, instead of becoming more refined, they eventually collapse to part of the modes. **Second row: MNIST embedded in a 3-dimensional space.** Colored: Real samples. Black: Generated samples. Epochs (Batches) from left to right: 0(0), 0(8), 0(32), 0(64), 32(0), 47(0). Similar progression has been observed.

Generalizing these observations, we propose a perspective that views GAN training as a two-phase progression from fitting to collapse. Our main tool for analyzing the two phases is the study of *gradient dynamics*, as gradients of the discriminator and generator functions with respect to their inputs provide insight into how generated samples evolve. Table 1 provides an overview of each phase. Identifying the transition from fitting to collapse is crucial, as stopping training at this point can retain sample quality and diversity: stopping too early yields unrealistic samples, while stopping too late leads to reduced diversity.

Table 1: An overview of the two phases: fitting, and collapse, which includes a brief description and the roles of discriminator and generator gradients.

| | **Fitting** | **Collapse** |
| --- | --- | --- |
| Description | Particles move toward and converge around the modes, reducing their spread and mitigating mode mixture. | Particles near mode boundaries are pushed away, eventually leading to mode collapse. |
| Discriminator gradient | Guides particles from the initial noise prior toward regions close to the modes. | Pushes particles near mode boundaries away with significant force and magnitude. |
| Generator gradient | Measures how the generator maps nearby points in the latent space to distant points in the output space, quantitatively measure the severity of mode mixture. | Drops in magnitude as particles near mode boundaries are pushed away and concentrate around fewer modes. |

Our contributions can be summarized as follows:

- **We propose a perspective that views GAN training as a two-phase progression from fitting (section 3) to collapse (section 4)**, where mode mixture and mode collapse are treated as interconnected. Notably, we highlight the underexplored idea that mode collapse (i.e., the collapse phase) may emerge in the later stages of a converging GAN (i.e., the fitting phase).

- **We employ gradient-based tools to analyze each phase, using the discriminator gradient which guides particle movement and the generator gradient, termed "steepness," to quantify mode mixture severity.** These tools are detailed in section 2.

- **We develop an early stopping algorithm to optimize GAN training by detecting the transition from fitting to collapse.** The early stopping algorithm, outlined in section 4.3, uses a metric based on discriminator and generator gradients. By intrinsically capturing GAN training dynamics without direct dependence on generated or real images, it identifies a stopping point where both sample quality and diversity are retained, as empirically demonstrated in section 6.

## 2 Technical Preliminaries and Basic Assumptions

In this section, we provide the technical preliminaries and basic assumptions. We begin with an overview of the gradient dynamics in section 2.1, focusing on how the generator and discriminator gradients shape the behavior of generated samples across the two phases, as summarized in table 1. In section 2.2, we present an interpretation of GANs as particle models, where the discriminator gradient guides the movement of generated samples as particles. In section 2.3, we introduce the concept of steepness, derived from the generator gradient, to quantify the severity of mode mixture. Finally, in section 2.4, we outline the assumptions we make regarding the real data distribution and the noise prior.

### 2.1 An Overview of Gradient Dynamics in GANs

In this work, the main tools we use to analyze the proposed two phases of GAN training are the generator and discriminator gradients. To clarify, we consider gradients as derivatives of the generator and discriminator functions with respect to their inputs, rather than with respect to network parameters. In section 2.2, we interpret the divergence GANs as *particle models*, where generated samples are viewed as particles, each moving based on a function of the discriminator's gradient. During the fitting phase, this gradient guides particles from the initial noise prior towards regions near the modes. In section 2.3, we define *steepness* based on the generator's gradient. Steepness quantifies how the generator maps nearby points in the latent space to potentially distant points in the output space, providing a measure of mode mixture severity that enables a quantitative analysis of this phenomenon in the fitting phase. The collapse phase is characterized by two distinctive behaviors. From a particle perspective, certain particles near the mode boundaries start

---

**Algorithm 1** Interpretation of Non-Saturating GAN as a Particle Model (c.f. (Yi et al., 2023))

---

**Require:** The discriminator $d_\omega$ (with $\omega$ denoting the discriminator's parameters) and the generator $g_\theta$ (with $\theta$ denoting the generator's parameters), the noise prior $p_z$, batch size $m > 0$, step size $s > 0$
1: **for** number of training iterations **do**
2:     Train the discriminator $d_\omega$ as in (Goodfellow et al., 2014).
3:     Sample $\boldsymbol{z}_i$'s from the noise prior $p_z(\boldsymbol{z})$.
4:     Generate particles $\boldsymbol{Z}_i = g_\theta(\boldsymbol{z}_i)$, $(1 \le i \le m)$.
5:     Update the particles $\hat{\boldsymbol{Z}}_i = \boldsymbol{Z}_i + s \cdot \nabla d_\omega(\boldsymbol{Z}_i)/d_\omega(\boldsymbol{Z}_i)$, $(1 \le i \le m)$.
6:     Apply the *stop gradient operator* to $\hat{Z}_i$ and update $g_\theta$ by descending $\nabla_\theta \frac{1}{m} \sum_{i=1}^{m} \left\| g_\theta(\boldsymbol{z}_i) - \hat{\boldsymbol{Z}}_i \right\|_2^2$.
7: **end for**

---

to "escape" from these modes, a phenomenon we analyze using the discriminator's gradient. From a global perspective, generated particles begin to concentrate around only a few modes. We use the generator's gradient to characterize this concentration effect, giving a comprehensive view of the collapse phase.

## 2.2 Discriminator Gradient: Guiding Particle Movement

Divergence GANs such as Vanilla GAN (Goodfellow et al., 2014), NSGAN (Goodfellow et al., 2014) and $f$-GAN (Nowozin et al., 2016) can be interpreted as *particle models* (Gao et al., 2019; Johnson & Zhang, 2019; Franceschi et al., 2023; Huang & Zhang, 2023; Yi et al., 2023). This paper focuses on the NSGAN for its practicality and conciseness. And we outline the methodology for other Divergence GANs in appendix H. The pseudocode of NSGAN as a particle model is presented in algorithm 1, which is fundamentally grounded in the work of Yi et al. (2023). This interpretation is essentially equivalent to the original NSGAN (Yi et al., 2023, theorem 3.2). Accordingly, we refer to the generated samples as *particles* throughout this paper. Unless otherwise stated, we assume the discriminator is optimal[1], i.e., $d_*(\boldsymbol{x}) = p_{\text{data}}(\boldsymbol{x})/\big(p_{\text{data}}(\boldsymbol{x}) + p_g(\boldsymbol{x})\big)$, as established by Goodfellow et al. (2014) (we omit the subscript $\omega$ for brevity hereafter). Consequently, the vector field $\nabla d(\boldsymbol{x})/d(\boldsymbol{x})$ that *guides particle movement* can be reformulated in terms of the density ratio $r(\boldsymbol{x}) = p_{\text{data}}(\boldsymbol{x})/p_g(\boldsymbol{x})$ as

$$\frac{\nabla d_*(\boldsymbol{x})}{d_*(\boldsymbol{x})} = \nabla r(\boldsymbol{x}) \cdot \frac{1}{r(\boldsymbol{x})(1 + r(\boldsymbol{x}))}. \tag{1}$$

## 2.3 Generator Gradient: Measuring Mode Mixture Severity

In addition to the discriminator's role in guiding particle movement, the generator's gradient provides a measure of mode mixture severity. As described in algorithm 1, a particle $\boldsymbol{x}$ near a mode is updated in the direction of $\nabla d_*(\boldsymbol{x})/d_*(\boldsymbol{x})$, typically pointing towards the nearest mode (see section 3). However, between adjacent modes, critical points exist where nearby particles are pushed apart in opposite directions. During training, two close latent points $\boldsymbol{z}_1$ and $\boldsymbol{z}_2$ may map to outputs $g_\theta(\boldsymbol{z}_1)$ and $g_\theta(\boldsymbol{z}_2)$ that are far apart. This behavior reflects high sensitivity in the generator's mapping, indicated by a large spectral norm of the Jacobian of $g_\theta$. This motivates the concept of *steepness* (see definition 2.1), which quantifies the generator's sensitivity across the latent space. Higher steepness corresponds to regions where small differences in latent points produce large separations in the output space, mitigating the severity of mode mixture. Importantly, this definition is invariant under orthogonal coordinate transformations, ensuring that steepness $\mathcal{S}_g$ captures intrinsic properties of the generator's mapping. This notion of steepness is related to several works that study the conditioning and Lipschitz constants of the generator Jacobian, which we discuss in section 5.

**Definition 2.1.** *Let $g \colon \mathbb{R}^n \to \mathbb{R}^n$ be continuously differentiable. The steepness of $g$ at a point $\boldsymbol{x}$, denoted by $\mathcal{S}_g(\boldsymbol{x})$, is defined as the spectral norm of the Jacobian of $g$ at $\boldsymbol{x}$:*

$$\mathcal{S}_g(\boldsymbol{x}) = \|J_g(\boldsymbol{x})\|_2. \tag{2}$$

---

[1] For the sake of completeness, we also provide an analysis of a class of *suboptimal* discriminators in appendix D.

## 2.4 Assumptions on Real Data and Noise Prior

Gaussian smoothing of the data, as applied in assumption 2.1, is a commonly used approach in machine learning to transform discrete datasets into continuous probability distributions. This method enables mathematical analysis, aligns with standard data preprocessing practices, and is widely adopted in generative modeling (Goldfeld et al., 2020; Ho et al., 2020; Song et al., 2021; Karras et al., 2022). By employing kernel density estimation with a Gaussian kernel $K_\sigma(\cdot, \cdot)$ and covariance matrix $\sigma^2 \boldsymbol{I}_n$, we can provide a smooth and representative approximation of the underlying data distribution.

**Assumption 2.1.** *Let $\boldsymbol{x}_1, \boldsymbol{x}_2, \ldots, \boldsymbol{x}_N \in \mathbb{R}^n$, where the $\boldsymbol{x}_i$'s are in ascending order if $n = 1$. We assume that the real data distribution has the following probability density function*

$$p_{data}(\boldsymbol{x}) = \frac{1}{N} \sum_{i=1}^{N} K_\sigma(\boldsymbol{x}, \boldsymbol{x}_i) := \frac{1}{N} \sum_{i=1}^{N} \frac{1}{(2\pi\sigma^2)^{n/2}} \cdot \exp\Big( -\frac{\|\boldsymbol{x} - \boldsymbol{x}_i\|_2^2}{2\sigma^2} \Big), \tag{3}$$

*Optionally, we may assume a separation condition parameter $\Delta > 0$, such that $\min_{1 \leq i < j \leq N} \|\boldsymbol{x}_i - \boldsymbol{x}_j\|_2 \geq \Delta$.*

We make the following assumption on the noise prior $p_z(\boldsymbol{z})$ for reasons in appendix B.

**Assumption 2.2.** *Let $n$ be the dimension of real samples. We assume that the noise prior $p_z \sim \mathcal{N}(\boldsymbol{0}, \boldsymbol{I}_n)$ is an $n$-dimensional standard Gaussian distribution.*

# 3 The Fitting Phase: Gradient Dynamics

The fitting phase of GAN training refers to the process where generated particles align with the modes of the real data distribution. This alignment is driven by both discriminator gradients and generator gradients, each playing a distinct yet complementary role: (i) discriminator gradients guide the generated particles by providing directions toward the nearest modes, and (ii) generator gradients, specifically steepness, determines the generator's ability to separate modes and reduce mode mixture. Together, these gradients ensure that generated particles converge around real modes, even though some particles may still remain outside or within the inter-modal regions, reflecting mode mixture. In this section, we analyze the role of discriminator gradients in section 3.1 and the impact of generator steepness in section 3.2.

## 3.1 Evolution of Discriminator Gradients

In this subsection, we analyze and visualize how discriminator gradients guide generated particles towards modes *during the fitting phase*. Importantly, we assume an optimal discriminator, which ensures exact update directions for particle movement. This assumption applies specifically to the fitting phase and does not hold in the collapse phase, where the discriminator may lose its optimality (see section 4.1). While prior work has established a solid theoretical foundation by showing that an optimal discriminator induces a Wasserstein gradient flow to align $p_g$ with $p_{\text{data}}$ (Franceschi et al., 2023; Yi et al., 2023), we examine four illustrative scenarios in table 2 to investigate how this process depends on the configurations of $p_g$ and $p_{\text{data}}$[2].

Table 2: Descriptions and configurations of $p_{\text{data}}$ and $p_g$ in the four cases.

|  | Description | $p_{\text{data}}$ | $p_g$ |
|---|---|---|---|
| **Case 1** | Initialization with concentrated particles | $\mathcal{N}([\pm1, \pm1], 0.1\boldsymbol{I}_2)$ | $\mathcal{N}([0, 0], 0.2\boldsymbol{I}_2)$ |
| **Case 2** | Particles covering all modes | $\mathcal{N}([\pm1, \pm1], 0.1\boldsymbol{I}_2)$ | $\mathcal{U}([-2, 2] \times [-2, 2])$ |
| **Case 3** | Particles concentrated near a single mode | $\mathcal{N}([\pm1, \pm1], 0.1\boldsymbol{I}_2)$ | $\mathcal{N}([1, 1], \boldsymbol{I}_2)$ |
| **Case 4** | Globally separated modes | $\mathcal{N}([\pm3, \pm3], 0.1\boldsymbol{I}_2)$ | $\mathcal{N}([3, 3], 3\boldsymbol{I}_2)$ |

---

[2] For simplicity, the 3-dimensional Gaussian mixture dataset in fig. 1 is projected onto the $xy$-plane, and the covariance of each Gaussian component is set to $0.1\boldsymbol{I}$.

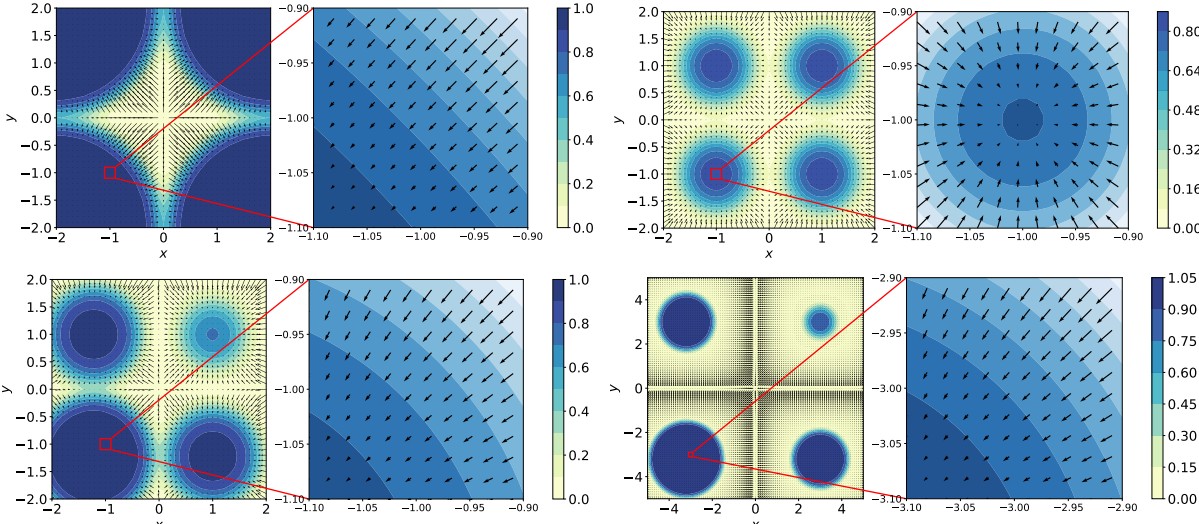

Figure 2: The vector field $\nabla d_*(\boldsymbol{x})/d_*(\boldsymbol{x})$ with zoomed-in views around the bottom left mode. Colorbars in the left subfigures show discriminator values (i.e., $d_*(\boldsymbol{x})$). **Top left**: Case 1. **Top right**: Case 2. **Bottom left**: Case 3. **Bottom right**: Case 4. These subfigures highlight how discriminator gradients guide generated particles under different configurations.

Case 1 (top left subfigure of fig. 2) shows that, the update vector field $\nabla d_*(\boldsymbol{x})/d_*(\boldsymbol{x})$ pulls particles toward the direction of the nearest modes, with vector lengths proportional to their distances from the modes. This ensures rapid initial movement of particles towards the modes at initialization. For case 2 (top right subfigure of fig. 2), particles near the mode centers exhibit minimal movement due to weak gradient forces, while those farther away are guided towards the nearest mode by stronger discriminator gradients. This results in a progressive sharpening of $p_g$, as particles accumulate near the modes. For case 3 (bottom left subfigure of fig. 2), the discriminator values (indicated by the colorbar) differ significantly across modes: the covered mode (which centers at $[1,1]$) has the lowest value, while the farthest mode (which centers at $[-1,-1]$) has the highest. The vector field's intensity peaks near unoccupied modes and diminishes near crowded ones, This dynamic ensures redistribution of particles to balance coverage across modes. For case 4 (bottom right subfigure in fig. 2), the gradient intensity weakens near all modes, making particle movement less effective. Proper initialization and balanced training are critical in this setting to avoid stagnation[3].

**The role of discriminator gradients during the fitting phase.** Discriminator gradients play a crucial role in guiding generated particles toward the nearest modes during the fitting phase. This explains why particles, initially clustered (the first subfigure of fig. 1), spread out and move toward modes. For locally clustered modes (Cases 1–3), the gradients effectively align particles with the modes. For globally separated modes (Case 4), the gradients weaken, and successful convergence depends on careful initialization and balanced training.

## 3.2 Evolution of Steepness

While discriminator gradients guide particles toward modes, mode mixture is still commonly observed in practice. To understand this phenomenon, we analyze the role of generator gradients, characterized by the steepness of the generator function. Steepness measures how sharply the generator transforms adjacent points in the latent space into distinct points in the data space. In this subsection, we first establish that the optimal generator must exhibit significant steepness to transform a standard Gaussian distribution into $p_{\text{data}}$ (both in 1-dimensional and $n$-dimensional cases). Conversely, when the steepness is insufficient, the

---

[3] Please refer to appendix C for the corresponding theoretical results.

transformation of the standard Gaussian distribution via the generator does not fully match $p_{\text{data}}$. This discrepancy causes mode mixture, where generated samples fail to align perfectly with the real modes.

**Steepness of measure-preserving maps.**    We begin by analyzing the steepness of the optimal generator function $g$ that satisfies $g_\# p_z = p_{\text{data}}$. In the 1-dimensional case, any measure-preserving map $g$ can be expressed as $g = \Psi^{-1} \circ h \circ \Phi$, where $\Phi$ and $\Psi$ are the cumulative distribution functions (CDFs) of $\mathcal{N}(0,1)$ and $p_{\text{data}}$, respectively, and $h$ is a measure-preserving map of the uniform distribution $\mathcal{U}(0,1)$. Among the infinitely many possible choices of $h$, the identity map holds particular significance. In this case, the corresponding $g$ represents the optimal transport map from $p_z$ to $p_{\text{data}}$ under the Wasserstein distance with strictly convex cost functions (Santambrogio, 2015), including the widely-used 2-Wasserstein distance as a specific example[4].

**Theorem 3.1** (Steepness of 1-dimensional measure-preserving maps). *Assume that the real data distribution $p_{data}(x)$ satisfies assumption 2.1 with $n = 1$ and separation condition parameter $\Delta = 6\sigma$. Let $\Phi(x)$ and $\Psi(x)$ denote the cumulative distribution functions (CDFs) of $\mathcal{N}(0,1)$ and $p_{data}(x)$, respectively. Define $g(x) := \Psi^{-1}(\Phi(x))$. Then, there exists a point $x_* \in \mathbb{R}$ such that the steepness of $g$ at $x_*$ satisfies:*

$$\mathcal{S}_g(x_*) \geq \min_{1 \leq i \leq N-1} \sigma \cdot \exp\left(\frac{(x_{i+1} - x_i)^2}{8\sigma^2}\right) \cdot \exp(-q^2), \tag{4}$$

*where $q$ is the $(1 - 1/N)$th quantile of the standard Gaussian distribution.*

As indicated by theorem 3.1, the steepness $\mathcal{S}_g$ grows exponentially with the square of the distance between neighboring modes. This means that for distributions where the modes are further apart, the generator function must exhibit a much larger steepness to accurately map samples between modes. Conversely, $\mathcal{S}_g$ is inversely related to the variance $\sigma^2$ of each mode. A smaller variance requires the generator to transition more sharply between modes, resulting in a steeper function. This property extends to higher dimensions, as demonstrated in theorem 3.2, which provides an explicit lower bound for $\mathcal{S}_g$.

**Theorem 3.2** (Steepness of $n$-dimensional measure preserving maps). *Assume that the real data distribution $p_{data}(\boldsymbol{x})$ satisfies assumption 2.1, and that the noise prior $p_z(\boldsymbol{z})$ is the truncated Gaussian $\mathcal{N}_r(\boldsymbol{0}, \boldsymbol{I}_n)$ defined on the $n$-dimensional ball $\mathcal{B}_r(\boldsymbol{0})$. Without loss of generality, suppose $\boldsymbol{x}_i \neq \boldsymbol{0}$ for all $1 \leq i \leq N$. Let $g \colon \mathcal{B}_r(\boldsymbol{0}) \to \mathbb{R}^n$ be a continuously differentiable, piecewise injective function satisfying $g_\# p_z = p_{data}$. Then, there exists a point $\boldsymbol{x}_* \in \mathbb{R}^n$ such that the steepness $\mathcal{S}_g(\boldsymbol{x}_*)$ satisfies $\mathcal{S}_g(\boldsymbol{x}_*) \geq M$, where*

$$M = \delta \cdot \sigma \cdot \sqrt{2\pi} \cdot \max_{\lambda \in [0,2]} \min_{1 \leq i \leq N} \exp\left(\frac{\|\lambda \bar{\boldsymbol{x}} - \boldsymbol{x}_i\|_2^2}{2n\sigma^2}\right). \tag{5}$$

*Here, $\bar{\boldsymbol{x}} = \sum_{i=1}^{N} \boldsymbol{x}_i / N$ is the mean of the mode centers, and $\delta = \exp(-r^2/2)/\sqrt{2\pi}$ accounts for the truncation of the Gaussian distribution.*

Similar to the 1-dimensional case, the bound exhibits exponential growth with increasing distances $\|\lambda \bar{\boldsymbol{x}} - \boldsymbol{x}_i\|_2$ and with decreasing variance $\sigma^2$. Consequently, when the modes are widely separated or when the standard deviation $\sigma$ is small, $\mathcal{S}_g$ becomes significantly large. This observation may provide insight into the common practice of normalizing or rescaling image data during preprocessing, as these steps can reduce the steepness required for the generator to map the noise distribution to the real data distribution, which potentially influence the training dynamics.

**Evolution of steepness.**    We have established that in order to map a standard Gaussian distribution to $p_{\text{data}}$, the generator function must exhibit large steepness. A natural question arises: How does the steepness evolve during training? In the next theorem 3.3, we analyze the setting where $p_{\text{data}}$ is a symmetric mixture of Gaussians and derive an evolution equation in continuous time for the steepness of the generator function $g_t$ at $z = 0$, i.e., $g_t'(0)$ (which follows from the observation in fig. 3 that the generator function reaches its maximum steepness at $z = 0$). This equation provides two key implications: (i) when the third-order derivative of $g_t$ at 0, i.e., $g_t^{(3)}(0)$, is small compared to $g_t'(0)$ (for example, when $g_t$ can be well-approximated

---

[4] For a visualization of generator functions $g$ corresponding to different $p_{\text{data}}$, please refer to appendix I.

by a linear function near $x = 0$, as fig. 3 depicts), the steepness $g_t'(0)$ monotonically increases; and (ii) at an early stage of training, the dominant terms in the numerator and denominator of the right-hand side of the evolution equation are $(\mu^2 - \sigma^2)/\sigma^4 \cdot (g_t'(0))^3$ and $(g_t'(0))^2$, respectively. As a result, $g_t'(0)$ initially grows exponentially with rate $(\mu^2 - \sigma^2)/\sigma^4$. This suggests that when the modes are well separated ($\mu \gg \sigma$), the generator rapidly increases its steepness to match the target distribution. Theorem 3.3 uses the gradient-flow framework, which is a common idealization in theoretical analyses, as it allows us to capture the continuous-time dynamics underlying the update process. In practice, this corresponds to settings where the generator learning rate ($s$ in algorithm 1) is sufficiently small and the generator is smooth, so that its evolution can be well-approximated by a continuous-time trajectory. Although this gradient-flow assumption is an abstraction from the discrete and stochastic updates used in practical implementations, it provides clear insights into the generator's behavior, most notably the rapid increase in steepness during early training, without imposing overly restrictive conditions on the generator. Our analysis indicates that similar trends are observed in experiments even when the dynamics deviate from the idealized gradient-flow limit. In particular, while strict monotonicity may not always hold in practice due to the complexities of real-world settings, our experiments (see section 6) confirm that steepness increases sharply during early training, enabling the generator to better capture different modes and reducing the severity of mode collapse.

**Theorem 3.3** (Evolution of steepness under a symmetric mixture of Gaussians). *Suppose $p_{data}$ is a symmetric mixture of Gaussians*

$$p_{data} \sim 0.5\mathcal{N}(-\mu, \sigma^2) + 0.5\mathcal{N}(\mu, \sigma^2) \tag{6}$$

*where $\mu \gg \sigma$, i.e., the modes are well separated, and that the discriminator is optimal, i.e., the discriminator consistently provides the precise moving direction for the particle. Let $g_t \colon \mathbb{R} \to \mathbb{R}$ be a one-dimensional generator evolving in continuous time $t \geq 0$ according to the gradient-flow limit of the update*

$$g_{t+\Delta t}(z) = g_t(z) + \Delta t \cdot \frac{d_t'(g_t(z))}{d_t(g_t(z))}, \tag{7}$$

*as in algorithm 1, where*

$$d_t(x) = \frac{p_{data}(x)}{p_{data}(x) + p_{g_t}(x)}, \tag{8}$$

*and $p_{g_t}$ is the push-forward of the standard normal $\mathcal{N}(0,1)$ under $g_t$. Assume that $g_t$ is continuously differentiable for all $t \geq 0$ with $g_0$ being an odd function and $g_0'(0) > 0$. Then the steepness of $g$ at $z = 0$, namely $g_t'(0)$, satisfies the ODE*

$$\frac{\mathrm{d}}{\mathrm{d}t} g_t'(0) = \frac{(\mu^2 - \sigma^2)/\sigma^4 \cdot (g_t'(0))^3 + g_t'(0) + g_t^{(3)}(0)}{\exp(-\mu^2/(2\sigma^2))/\sigma \cdot (g_t'(0))^3 + (g_t'(0))^2}. \tag{9}$$

We remark that while theorem 3.3 focuses on the two-Gaussian case, the underlying principle theoretically extends to distributions with more modes, though the corresponding evolution equations become significantly more complex and warrant a separate discussion. In contrast, for the simpler case of a single-mode distribution, the steepness admits a closed-form analytical expression. We provide a detailed analysis of this setting in appendix C.

**Quantitative relationship between steepness and mode mixture severity.** Next, we present quantitative results that illustrate how the steepness of the generator impacts the severity of mode mixture, as detailed in theorem 3.4. The theorem implies the following: (i) mode mixture is inevitable when steepness is insufficient: The probability that particles fall into the regions between adjacent modes (i.e., the mode mixture regions) depends inversely on the steepness $k$ of the generator. Specifically, for small $k$, these probabilities increase significantly, indicating that a less steep generator leads to more severe mode mixture. (ii) sufficient steepness minimizes mode mixture: As $k$ increases, the probability of particles falling into mode mixture regions decreases. This reflects the ability of a steep generator to tightly map latent samples to real modes, thereby aligning generated particles with the real data distribution and reducing overlap between modes.

Figure 3 visually supports these implications. In the left subfigure, different generator functions with varying steepness are depicted, showcasing their ability to align latent and real distributions. The right subfigure

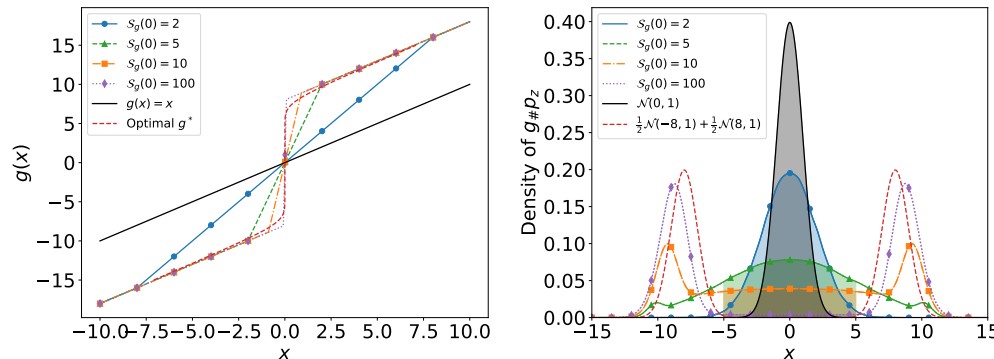

Figure 3: **Left**: Generator functions $g$ with varying steepness at $x = 0$. **Right**: The density plot of $p_g = g_\# p_z$, with $p_{\text{data}} \sim 0.5\mathcal{N}(-8, 1) + 0.5\mathcal{N}(8, 1)$. The shaded areas represent the severity of mode mixture. Generator functions with larger steepness exhibit less severe mode mixture. Quantitative results are detailed in theorem 3.4.

highlights the severity of mode mixture for the case $N = 2$ with $x_1 = -x_2 = 8$, where shaded areas represent the mode mixture regions. These visualizations confirm that a larger steepness reduces the severity of mode mixture, consistent with the theoretical findings in theorem 3.4. Practically, this underscores the importance of designing and training generators with sufficient steepness to alleviate mode mixture.

**Theorem 3.4** (Relationship between steepness and mode mixture severity). *Assume that the real data distribution $p_{data}(x)$ satisfies assumption 2.1 with $n = 1$ and separation condition parameter $\Delta = 6\sigma$. Furthermore, assume that the generator function $g$ is increasing and satisfies $\sup_{x \in \mathbb{R}} \mathcal{S}_g(x) \leq k$. Additionally, assume that*

$$g^{-1}\left(\frac{x_i + x_{i+1}}{2}\right) = \Phi^{-1}\left(\Psi\left(\frac{x_i + x_{i+1}}{2}\right)\right), \tag{10}$$

*where $\Phi(x)$ denotes the cumulative distribution function (CDF) of the standard normal distribution $\mathcal{N}(0, 1)$, and $\Psi(x)$ is the CDF of the distribution $p_{data}(x)$. Then, the probability that the particles fall into the interval*

$$\bigcup_{i=1}^{N-1} [x_i + 3\sigma, x_{i+1} - 3\sigma], \tag{11}$$

*which indicates mode mixture, is at least*

$$\sum_{i=1}^{N-1}\left(\Phi\left(\Phi^{-1}\left(\Psi\left(\frac{x_i + x_{i+1}}{2}\right)\right) + \frac{x_{i+1} - x_i - 3\sigma}{2k}\right) - \Phi\left(\Phi^{-1}\left(\Psi\left(\frac{x_i + x_{i+1}}{2}\right)\right) - \frac{x_{i+1} - x_i - 3\sigma}{2k}\right)\right). \tag{12}$$

**The role of steepness during the fitting phase.** To push forward a standard Gaussian to $p_{\text{data}}$, the generator's steepness exhibit exponential growth with increasing mode separation and inverse proportionality to mode variance. Under certain assumptions, steepness increases during training, aligning generated particles with real modes. Furthermore, mode mixture severity inversely correlates with steepness, such that higher steepness reduces overlap between modes. These findings collectively explain why mode mixture severity diminishes during the fitting phase, though it often cannot be entirely eliminated.

## 4 The Collapse Phase: Gradient Dynamics and Detection

In this section, we examine the collapse phase, where the diversity of generated samples deteriorates as they concentrate around fewer modes. This phase emerges at the end of the fitting phase, when generated samples closely approximate the real data. We investigate the underlying mechanisms of collapse, highlighting the role of discriminator gradients in driving particle dynamics in section 4.1 and its relationship to generator

steepness in section 4.2. Building on these insights, we introduce a practical early stopping algorithm in section 4.3 to stop GAN training at the critical transition from fitting to collapse, thereby preserving diversity.

## 4.1 Collapse Induced by Discriminator Gradients

In this subsection, we analyze the role of the discriminator gradient $\|\nabla d(\boldsymbol{x})/d(\boldsymbol{x})\|_2$ in the collapse phase. Collapse occurs at the end of the fitting phase, where generated samples closely approximate real data. Unlike the optimal discriminator in Vanilla GAN, which assigns uniform values of 0.5 to both real and generated samples at convergence, the optimal discriminator in NSGAN exhibits a more nuanced behavior. It assigns values near 0.5 at the central regions of the modes, values close to 0 in regions with scarce real data, and gradually transitions between these extremes.

**A locally linear approximation of the discriminator.** This behavior arises from two key mechanisms. First, recall that the optimal discriminator in NSGAN can be expressed as $d_*(\boldsymbol{x}) = p_{\text{data}}(\boldsymbol{x})/\big(p_{\text{data}}(\boldsymbol{x}) + p_g(\boldsymbol{x})\big)$ (Goodfellow et al., 2014). At the central regions of a mode, as discussed in section 3.1, empirical observations suggest the following dynamics: particles near the mode are attracted to it (Case 2). Conversely, an overaccumulation of particles at a mode triggers redistribution mechanisms (Case 3), which drive particles toward alternative modes to ensure balanced coverage of the data distribution. These dynamics collectively lead to the assumption that $p_g(\boldsymbol{x}) \approx p_{\text{data}}(\boldsymbol{x})$, which in turn implies $d_*(\boldsymbol{x}) \approx 0.5$. In regions far from the modes, $p_{\text{data}}(\boldsymbol{x}) \approx 0$. Due to the smoothing effect of $p_g(\boldsymbol{x})$, which spreads probability mass beyond the support of $p_{\text{data}}$, $p_g(\boldsymbol{x})$ remains finite. Consequently, $d_*(\boldsymbol{x})$ approaches 0. Second, the smoothing effect of $p_g(\boldsymbol{x})$ plays a critical role in shaping the transition between these extremes. Unlike $p_{\text{data}}$, which is sharply concentrated within the modes, $p_g(\boldsymbol{x})$ spreads probability mass more broadly, partly due to the mode mixture effects. Please refer to appendix G for empirical evidence on a toy example and real datasets. To better characterize the behavior of particle movement, we adopt a locally linear approximation of the discriminator, detailed in assumption 4.1.

**Assumption 4.1** (A locally linear approximation of the discriminator). *Assume that the real data distribution $p_{data}(x)$ satisfies assumption 2.1 with separation condition parameter $\Delta = 8\sigma$. We assume that at the end of the fitting phase where generated samples closely resemble real samples, the discriminator $d(\boldsymbol{x})$ is of the form*

$$d(\boldsymbol{x}) = \begin{cases} \dfrac{1}{2} - \dfrac{1}{8\sigma} \cdot \|\boldsymbol{x} - \boldsymbol{x}_i\|_2, & \boldsymbol{x} \in B_{4\sigma}(\boldsymbol{x}_i), \\ 0, & otherwise. \end{cases} \tag{13}$$

**Discriminator gradients near mode boundaries.** We compute $\|\nabla d(\boldsymbol{x})/d(\boldsymbol{x})\|_2$ to analyze gradient behavior near the mode boundaries. For a point $\tilde{\boldsymbol{x}}$ located $r$ away from $\boldsymbol{x}_i$, i.e., $\|\tilde{\boldsymbol{x}} - \boldsymbol{x}_i\|_2 = r$, we derive $\|\nabla d(\tilde{\boldsymbol{x}})/d(\tilde{\boldsymbol{x}})\|_2 = 1/(4\sigma - r)$. As $r$ approaches $4\sigma$, the sharp increase in $\|\nabla d(\boldsymbol{x})/d(\boldsymbol{x})\|_2$ indicates that particles near these regions experience disproportionately large updates. This disrupts the equilibrium established during the fitting phase, pushing particles away from the boundaries. Please refer to appendix G for a visualization of the discriminator gradient field in this scenario. As a result, generated samples begin to concentrate around fewer modes, reducing diversity and triggering mode collapse. Monitoring $\|\nabla d(\boldsymbol{x})/d(\boldsymbol{x})\|_2$ provides a clear signal of this transition, motivating the early stopping algorithm proposed in section 4.3 to prevent mode collapse and preserve sample diversity.

**The role of discriminator gradients during the collapse phase.** Near mode boundaries, discriminator gradients $\|\nabla d(\boldsymbol{x})/d(\boldsymbol{x})\|_2$ increase sharply, causing large particle updates that disrupt balance and push particles away from the boundaries. This behavior results in generated samples concentrating around fewer modes, thereby reducing diversity and triggering mode collapse.

## 4.2 A Local Analysis of Steepness

The steepness of the generator is a crucial metric for understanding its mapping behavior, particularly during the collapse phase of GAN training. A significant drop in steepness often signals mode collapse,

where generated samples lose diversity and concentrate around fewer modes. By studying steepness, we gain insights into how updates to the generator affect its ability to maintain diverse outputs. This motivates the need for a rigorous analysis of steepness dynamics and its connection to discriminator gradients, which is the focus of this subsection.

The discriminator gradients and the generator gradients are intrinsically connected through the particle update rule described in algorithm 1. This relationship, which governs how updates to particles propagate through the generator, is formally captured in theorem 4.1.

**Theorem 4.1** (Relationship between discriminator gradients and generator gradients)**.** *Following the notations in algorithm 1, assume that after the update step, the generator is optimal in the sense that $g_{\theta'}(\boldsymbol{z}_i) = \hat{\boldsymbol{Z}}_i$. Further assume there are infinitely many particles and that the step size $s > 0$ is sufficiently small. Then, the Jacobian $J_{g_{\theta'}}(\boldsymbol{z})$ of the updated generator $g_{\theta'}$ satisfies*

$$J_{g_{\theta'}}(\boldsymbol{z}) = J_{g_\theta}(\boldsymbol{z}) + s \cdot \nabla_{\boldsymbol{x}}\left(\frac{\nabla d_\omega}{d_\omega}\right)\big(g_\theta(\boldsymbol{z})\big) \cdot J_{g_\theta}(\boldsymbol{z}), \tag{14}$$

*where $\nabla_{\boldsymbol{x}}(\nabla d_\omega / d_\omega)(\boldsymbol{x})$ is the Jacobian of the vector field $\nabla d_\omega / d_\omega$ evaluated at $\boldsymbol{x}$.*

Building on this relationship, we focus on a local analysis of the generator's steepness near the collapsing mode. Specifically, we analyze how the steepness evolves after a single update during the collapse phase in theorem 4.2.

**Theorem 4.2** (Steepness drops in the collapse phase)**.** *Assume that the real data distribution $p_{data}(x)$ satisfies assumption 2.1 with separation condition parameter $\Delta = 8\sigma$. Suppose the current generator is $g_\theta$, and the current discriminator $d(\boldsymbol{x})$ satisfies the linear model assumption 4.1 near a certain mode $\boldsymbol{x}_i$, specifically $d(\boldsymbol{x}) = 1/2 - \|\boldsymbol{x} - \boldsymbol{x}_i\|_2/(8\sigma)$ for all $\boldsymbol{x} \in B_{4\sigma}(\boldsymbol{x}_i)$. Under the same conditions as in theorem 4.1, the steepness of the updated generator $g_{\theta'}$ satisfies*

$$\mathcal{S}_{g_{\theta'}}(\boldsymbol{z}) \leq \left(1 - \frac{s}{(4\sigma - r)^2}\right) \cdot \mathcal{S}_{g_\theta}(\boldsymbol{z}), \tag{15}$$

*for all latent vectors $\boldsymbol{z}$ such that $g_\theta(\boldsymbol{z}) \in B_{2\sigma}(\boldsymbol{x}_i)$, where $r = \|g_\theta(\boldsymbol{z}) - \boldsymbol{x}_i\|_2$, provided that the step size $s < (4\sigma - r)^2$ is sufficiently small.*

Theorem 4.2 reveal that steepness decreases significantly within the mode's surrounding neighborhood, reducing the generator's ability to separate latent points and maintain output diversity. This provides a clear signal for detecting the onset of mode collapse. Since detecting the distance to the mode center is often impractical, the proportional drop in steepness offers a viable alternative for identifying mode collapse. By focusing on the relative change in steepness, we simplify detection and reduce dependence on precise spatial measurements, as proposed in section 4.3.

**The role of steepness during the collapse phase.** Steepness decreases near collapsing modes, signaling reduced capacity to separate latent points. Monitoring proportional drops in steepness provides a practical method for detecting mode collapse.

## 4.3 The Early Stopping Algorithm

Based on the theoretical results, we propose an early stopping algorithm to stop GAN training before collapse occurs. This algorithm monitors two critical metrics: the discriminator gradient $\|\nabla d(\boldsymbol{x})/d(\boldsymbol{x})\|_2$, which signals large updates near mode boundaries, and the generator steepness $\mathcal{S}_g(\boldsymbol{x})$, which reflects the generator's ability to map latent vectors diversely. Training is terminated if either the discriminator gradient exceeds a predefined threshold or the generator steepness exhibits a significant proportional drop compared to its previous value. Please refer to algorithm 2 for the pseudocode.

The algorithm involves three key ingredients. (i) Two thresholds are introduced: $k_d/(2\sigma)$ for the discriminator gradient and $k_g$ for the generator steepness. The value of $\|\nabla d(\boldsymbol{x})/d(\boldsymbol{x})\|_2$ at $\boldsymbol{x}$ located $2\sigma$ away from certain mode accounts for the $1/(2\sigma)$, while $k_d$ is set proportional to the distance between adjacent modes. The

underlying rationale is that when $\|\nabla d(\boldsymbol{x})/d(\boldsymbol{x})\|_2$ is small relative to inter-mode distances, generated samples deviating from the modes can be re-attracted. However, as $\|\nabla d(\boldsymbol{x})/d(\boldsymbol{x})\|_2$ approaches inter-mode distances, particles gravitate toward alternate modes, risking collapse. The other threshold $k_g$ detects proportional drops in generator steepness, defined as $\Delta \mathcal{S}_g = (\mathcal{S}_g^{\text{current}} - \mathcal{S}_g^{\text{prev}})/\mathcal{S}_g^{\text{prev}}$. (ii) The $(1 - 1/m)$th quantile of $\|\nabla d(\boldsymbol{x})/d(\boldsymbol{x})\|_2$ is computed for each batch, where $m$ represents the number of modes. This choice presumes that once a specific mode begins to collapse, it signifies the start of the GAN transitioning into the collapse phase. (iii) A warm-up period of $N_w$ iterations prevents premature stopping during the fitting phase by ignoring initial metric fluctuations.

---

**Algorithm 2** Early Stopping of GANs (with Discriminator Gradient and Generator Steepness)

---

**Require:** A GAN model including a generator $g_\theta$ and a discriminator $d_\omega$, thresholds $k_d > 0$ and $k_g < 0$, the number of modes $m \geq 1$, the number of warm-up iterations $N_w$
 1: **for** each training iteration **do**
 2:     Train the discriminator $d_\omega$ and the generator $g_\theta$ as in algorithm 1.
 3:     Compute the $(1 - 1/m)$th quantile of $\|\nabla d_\omega/d_\omega\|_2$ for the current batch. Let this value be $q_d$.
 4:     Compute the mean steepness $\mathcal{S}_g^{\text{current}}$ across the batch and calculate the proportional drop compared to the previous iteration as $\Delta \mathcal{S}_g = (\mathcal{S}_g^{\text{current}} - \mathcal{S}_g^{\text{prev}})/\mathcal{S}_g^{\text{prev}}$.
 5:     **if** $(q_s > k_s/(2\sigma)$ **or** $\Delta \mathcal{S}_g < k_g)$ **and** current iteration $> N_w$ **then**
 6:         **break**
 7:     **end if**
 8:     Update $\mathcal{S}_g^{\text{prev}} = \mathcal{S}_g^{\text{current}}$.
 9: **end for**
10: **return** The best-performing model from earlier checkpoints.

---

We also discuss the feasibility of the number of modes $m$ and estimation of the generator's steepness. In practice, many datasets used in GAN training (e.g., MNIST, Fashion MNIST, CIFAR-10) have well-defined modes corresponding to distinct classes or clusters in the data distribution. For such datasets, the number of modes is typically known or can be reasonably estimated. In cases where the number of modes is unknown, clustering techniques (e.g., $k$-means or Gaussian Mixture Models) can provide a practical approximation of the mode count. While these methods may not always yield perfect accuracy, they offer a reasonable baseline for implementing algorithm 2 in more general scenarios. As for the steepness, it can be computed using automatic differentiation tools in modern deep learning frameworks. Although the computation may be resource-intensive, it is feasible for low-dimensional noise spaces or in settings with moderate computational budgets. For more efficient estimation, one could subsample the noise space or use stochastic approximation techniques to estimate steepness over a representative subset of noise vectors.

## 5 Related Work

In this section, we highlight two key aspects of related work: (i) the phenomenon of final-stage mode collapse in GAN training (Brock et al., 2019), and (ii) related concepts with generator steepness (Odena et al., 2018; Tanielian et al., 2020; Salmona et al., 2022). We position our work within these contexts and emphasize its contributions. For a more comprehensive review, please refer to appendix A.

**Final-stage mode collapse.** The phenomenon of mode collapse in the final training stages has been observed when scaling up GAN architectures, as documented in (Brock et al., 2019). Their work, which focuses on large-scale experiments using BigGAN architectures on high-resolution datasets, notes that "settings which were stable in previous works become unstable when applied at scale." Our study complements these findings by demonstrating that similar phenomena can occur in NSGAN at relatively smaller scales. Additionally, our analysis adopts a different perspective by focusing on the generator's overall steepness and the $L_2$-norm of the discriminator gradients $\nabla d(\boldsymbol{x})/d(\boldsymbol{x})$, providing a complementary angle to the layer-specific singular value analysis employed in BigGAN. Importantly, while Brock et al. (2019) primarily focuses on stabilizing large-scale GAN training to improve performance, our work emphasizes detecting mode collapse through quantitative metrics.

**Related concepts with steepness.** As for the notion of steepness, several related concepts have been examined, namely, the condition number of the generator's Jacobian (Odena et al., 2018) (i.e., the ratio of its largest to smallest singular value), and the global Lipschitz constant (Tanielian et al., 2020; Salmona et al., 2022) (which may be seen as the global supremum of steepness). Among these, Odena et al. (2018) introduced the condition number to assess the generator's conditioning and proposed Jacobian Clamping to stabilize training. While their work focuses on regularization techniques and overall performance, it does not explicitly address mode mixture or collapse. Conversely, our work uses steepness as a theoretical tool to quantify mode mixture and identify its decline as an indicator of mode collapse. Regarding the global Lipschitz constant, Tanielian et al. (2020) and Salmona et al. (2022) have provided valuable insights into its role in capturing multimodal distributions. Tanielian et al. (2020) derived an upper bound on precision for learning disconnected manifolds and proposed a Jacobian-based truncation method to reject off-manifold samples. Salmona et al. (2022) demonstrated that a high Lipschitz constant is necessary to capture well-separated modes and provided bounds for two-modal distributions. Our work complements these studies by deriving a lower bound for multi-modal distributions. In terms of technical details, our work differs in three key aspects: (i) we employ the density transformation formula as a central tool, directly linking the generator's push-forward density to the noise distribution, whereas Tanielian et al. (2020); Salmona et al. (2022) rely on the Gaussian isoperimetric inequality to analyze divergence measures; (ii) we analyze the evolution of steepness during training (theorem 3.3), combined with the quantitative relationship between steepness and mode mixture severity (theorem 3.4), we establish that mode mixture severity decreases over time. In contrast, Tanielian et al. (2020); Salmona et al. (2022) assume a static generator with a fixed Lipschitz constant, focusing on theoretical guarantees under this assumption; and (iii) we adopt a localized approach by leveraging steepness to analyze mode dynamics in specific regions of the data space. While Tanielian et al. (2020) follow a similar spirit in their rejection method for identifying and filtering out potentially mode mixture samples, our perspective allows us to observe how steepness declines near individual modes during collapse, offering complementary insights into the dynamics of mode collapse.

## 6 Experiments

In this section, we present the experimental results. All codes are provided in the supplementary material.

### 6.1 Verifying Fitting

We empirically verify the existence of the fitting phase in real-world datasets. Our experiments focus on MNIST and Fashion MNIST due to the clear separability of their modes. Detailed results, including those for Fashion MNIST, are provided in appendix G, with experimental settings and rationale detailed in appendix F.

**Methodology.** We train NSGAN on MNIST and analyze the generated images using a classification network $q(\boldsymbol{x})$. Here, $\boldsymbol{x}$ is an image tensor, and $q(\boldsymbol{x})$ outputs a 10-dimensional vector $(p_0, p_1, \ldots, p_9)$, where $p_i \in [0, 1]$ represents the likelihood of $\boldsymbol{x}$ being classified as digit $i$. For each batch, we count the pairs $(i, j)$ where both $p_i$ and $p_j$ exceed $10^{-2}$. Such occurrences, visualized in heatmaps in fig. 4, indicate that the corresponding image exhibits characteristics of both modes $i$ and $j$, which we interpret as mode mixture.

**Results.** At the beginning of training, the heatmap shows few nonzero entries, primarily due to the initial noise prior, which generates similar outputs across samples. As training progresses, more entries appear, reflecting the fitting phase, where generated samples spread to cover the space containing the modes. Off-diagonal entries, which indicate mode mixture, initially increase but then decrease in magnitude as the generator reduces overlap between modes. However, mode mixture persists even at the end of the fitting phase. These observations align with the theoretical analysis in section 3. This trend is closely linked to the behavior of the discriminator gradient and steepness, as shown in fig. 5. At the stage when more entries appear in the heatmap, the discriminator gradient's magnitude remains relatively small, while its steepness increases rapidly. This corresponds to the beginning of training, where the generator increases steepness to distribute samples across the space and capture more modes. As training continues and off-diagonal entries begin to decrease in magnitude, the steepness stabilizes, and the discriminator gradient starts to oscillate. The stabilization reflects the generator's ability to better separate modes, leading to a reduction in mode

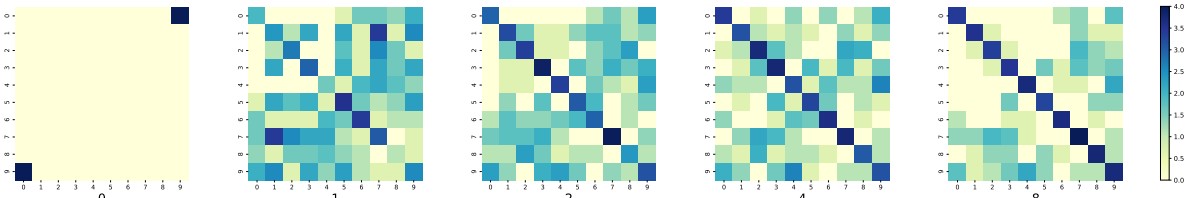

Figure 4: The logarithm of the occurrence of pairings $(i, j)$ plus 1 in a batch of size 256. Epochs from left to right: 0, 1, 2, 4, 8. At initialization, the noise prior results in few nonzero entries. As training progresses, more entries appear, indicating that generated samples spread across the mode space. Off-diagonal entries signal mode mixture, which decreases over time, validating the fitting. However, mode mixture persists even after fitting. Annotated heatmaps can be found in appendix G.

mixture. At the same time, the discriminator becomes increasingly effective at distinguishing samples and providing gradients that align with the assumptions in assumption 4.1.

## 6.2 Early Stopping

We present the results of applying early stopping (algorithm 2) to 3-dimensional Gaussian mixture, MNIST, Fashion MNIST, and CIFAR-10. Detailed experimental settings are provided in appendix F.

**Early stopping.** We train NSGAN on each dataset and record $\|\nabla d(\boldsymbol{x})/d(\boldsymbol{x})\|_2$ at each epoch until reaching the maximum specified epochs. The thresholds for early stopping are determined by $k_d/(2\sigma)$ for the discriminator gradient and $k_g = -0.5$ for the generator steepness, where $k_d$ represents the estimated distance between two modes and $\sigma$ is the estimated standard deviation of the data distribution. To evaluate the effectiveness of early stopping, we continue training beyond the stopping point to observe the sample quality both before and after this threshold is crossed. The experimental results are shown in fig. 5, which are shown at intervals of a few epochs before and after early stopping. These intervals reflect the common practice of periodically saving model checkpoints. However, the intervals were determined post-hoc by visually identifying the collapse point, so they may not be evenly spaced. Before the stopping point, the generated samples remain diverse and realistic. In the cases of MNIST and CIFAR-10, our algorithm effectively detects the epochs immediately preceding collapse. While the threshold may not always pinpoint the exact moment of collapse in other scenarios, it consistently ensures high-quality samples are retained before collapse occurs. Beyond this stopping point, however, the samples frequently collapse into a limited number of modes or oscillate between modes, leading to a significant loss in diversity and quality.

**Comparison with FID score and duality gaps.** In the evaluation of GAN performance, the metrics used can generally be classified into two categories: domain-specific and domain-agnostic. To comprehensively assess our proposed approach, we selected the FID score (Heusel et al., 2017) as a representative of domain-specific metrics, focusing on the quality of the generated images, and duality gaps (Grnarova et al., 2019; Sidheekh et al., 2021) to represent domain-agnostic metrics that evaluate the optimization process. We compare our proposed metric with the FID score across three key aspects. (i) In terms of applicability during training, the FID score is frequently used for retrospective evaluation, where generator checkpoints are saved periodically, and the FID score is calculated post hoc. This approach typically involves generating a large number of samples (e.g., 10k or more) and feeding them through a pretrained Inception network, making it computationally intensive. In contrast, our algorithm is designed to integrate directly into the training process, and is relatively computationally efficient (requiring batch-level gradient computations for both the generator and the discriminator, which is comparable to standard GAN training). Moreover, the use of early stopping with our metrics can reduce the number of checkpoint saving, providing a practical advantage in resource-intensive training scenarios. (ii) In terms of sensitivity to mode collapse, in fig. 6, we show that our metric, steepness, is closely aligned with FID in detecting mode collapse. Specifically, steepness exhibits a sharp decline during collapse phases, corresponding to a rapid increase in FID scores. Both

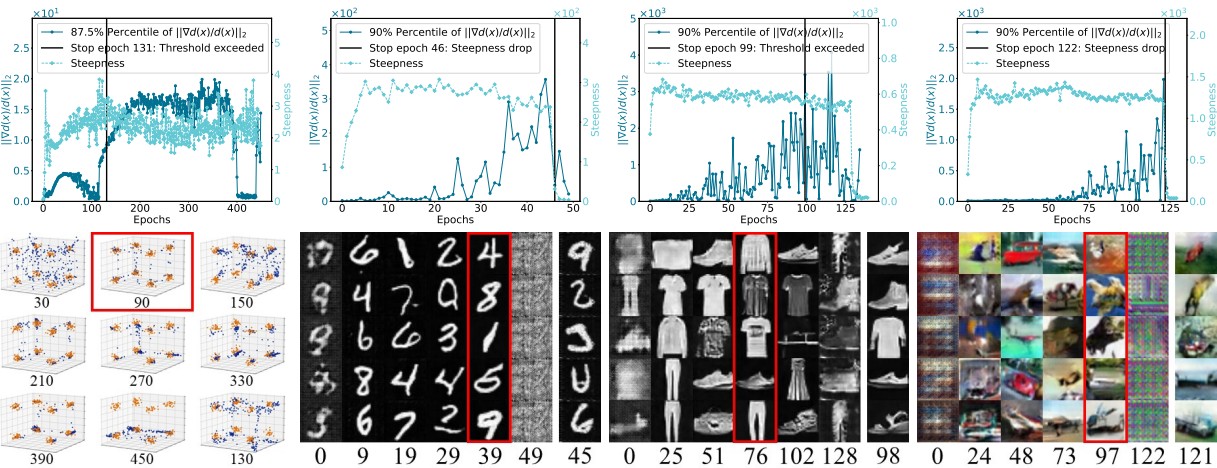

Figure 5: Experimental results of early stopping. Each column, from left to right, represents results for the following datasets: Gaussian Mixture, MNIST, Fashion MNIST, and CIFAR-10. In the first row, the blue circled lines indicate $\|\nabla d(\boldsymbol{x})/d(\boldsymbol{x})\|_2$, while the light blue diamond-shaped lines represent the steepness. The stopping epoch is indicated by the black vertical line. In the second row, generated images are shown at intervals of a few epochs before and after early stopping, reflecting the common practice of saving model checkpoints periodically. Images highlighted with red frames correspond to the most realistic samples among those saved before the stopping point. Additionally, the final samples generated just prior to the early stopping trigger are included: 130th epoch for Gaussian Mixture, 45th epoch for MNIST, 98th epoch for Fashion MNIST, and 121th epoch for CIFAR-10. For Gaussian Mixture and MNIST (white dots appear along the edges of the images), these generated samples show turbulence. For Fashion MNIST and CIFAR-10, the generated samples remain diverse and realistic before early stopping, without noticeable quality deterioration. Notably, in the MNIST and CIFAR-10 cases, our algorithm accurately identifies the epoch just before collapse. In other cases, while the threshold may not always precisely pinpoint the exact collapse point, it reliably ensures that samples remain of high quality prior to collapse. Beyond the stopping point, however, the samples often collapse to a few modes or oscillate between modes, significantly reducing diversity and quality.

metrics effectively signal this transition. (iii) In terms of training dynamics, our metric provides insights that complement the FID score. Notably, steepness increases during the early epochs of training, corresponding to the phase where FID decreases most rapidly. This behavior may reflect the transition from prior noise to the modes of the real data distribution. Steepness then stabilizes and oscillates, which corresponds to the FID score slowly decreasing. This reflects the particles moving closer to the modes and improving the sample quality gradually. Toward the later stages of training, steepness begins to decline sharply, signaling mode collapse, which coincides with a rapid increase in the FID score. In summary, while FID remains a widely used and effective domain-specific measure for assessing GAN performance, our metric provides a reliable and complementary perspective. It is computationally efficient, sensitive to collapse phases, and offers interpretability during the training process, making it a practical tool for real-time training monitoring and intervention. Additionally, in appendix G, we compare $\|\nabla d(\boldsymbol{x})/d(\boldsymbol{x})\|_2$ with the FID score, both metrics with the duality gaps and present the GAN training losses.

**Validating the early stopping metric.** As a by-product, our discussion in section 4 supports the established practice of adding noise to the discriminator to stabilize GAN training (Wieluch & Schwenker, 2019). This stabilization mitigates disproportionately large $\|\nabla d(\boldsymbol{x})/d(\boldsymbol{x})\|_2$ values near mode boundaries which contributes to mode collapse. Conversely, observing $\|\nabla d(\boldsymbol{x})/d(\boldsymbol{x})\|_2$ in this noised setting validates the effectiveness of our metric. Specifically, prior to the 54th epoch, the noise-free model generally exhibits larger values compared to the noised model. At the end of the 54th epoch, the noise-free model collapses, with the value tending toward zero. Meanwhile, the noised model maintains stable values, as shown in fig. 7. As a remark, the apparent concentration of density on the left side of the histogram at the 54th epoch is due

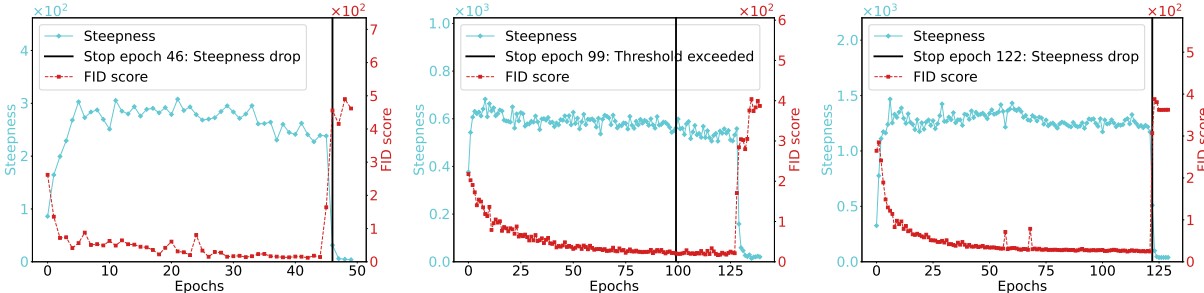

Figure 6: The tendency of steepness and FID score for MNIST, Fashion MNIST and CIFAR-10, from left to right. Light blue diamond-shaped for the steepness and red square-shaped for the FID score. A consistent pattern is observed: the steepness initially increases and stabilizes. Subsequently, whenever the steepness decreases significantly, the FID score nearly concurrently escalate to high values, signifying a notable deterioration in sample quality. Please refer to appendix G for comparison between $\|\nabla d(\boldsymbol{x})/d(\boldsymbol{x})\|_2$ and the FID score.

to the logarithmic scale of the $x$-axis, which compresses the right tail where outliers are located. Although these outliers are visually negligible, they correspond to large $\|\nabla d(\boldsymbol{x})/d(\boldsymbol{x})\|_2$ values and account for 10% of the total area beyond the 90th percentile. See appendix G for additional results.

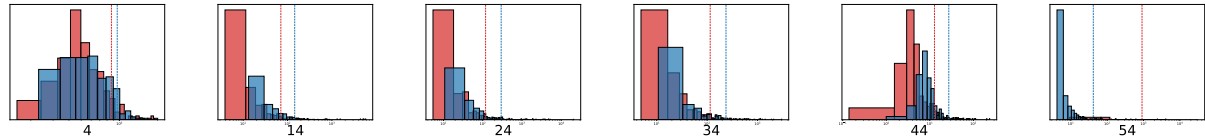

Figure 7: Histograms of the values of $\|\nabla d(\boldsymbol{x})/d(\boldsymbol{x})\|_2$ and their 90th percentile across epochs. Red for the model with noise and blue for the model without noise. The noise-free GAN collapses at the 54th epoch. Preceding that, the noised model nearly always exhibits lower $\|\nabla d(\boldsymbol{x})/d(\boldsymbol{x})\|_2$ values compared to its noise-free counterpart. Post that, this relationship reverses. Notably, in the noise-free model, $\|\nabla d(\boldsymbol{x})/d(\boldsymbol{x})\|_2$ tends towards zero, contributing to this observed divergence. See appendix G for additional results.

## 7 Conclusion

In this work, we propose a perspective that views GAN training as a two-phase progression from fitting to collapse, where mode mixture and mode collapse are treated as interconnected phenomena. We demonstrated that mode collapse can emerge in the later stages of a converging GAN and emphasized the importance of early stopping to retain sample diversity and quality. Using gradient dynamics, we analyzed how the discriminator gradient guides the movement of particles (generated samples) towards modes, while the generator gradient quantifies the severity of mode mixture by measuring how closely the generator maps nearby points in the latent space to distinct points in the output space. These insights allowed us to track the evolution of generated samples across the two phases. Our findings, validated through synthetic and real-world datasets, challenge conventional views on mode collapse and lay the groundwork for future research into improving GAN training stability and performance. For additional discussions, please refer to appendix J.

## Acknowledgements

We thank Luo Luo for his valuable suggestions. We also appreciate the anonymous reviewers for their constructive feedback, which significantly contributed to the revision process. We are grateful to Fernando Perez-Cruz and Michael U. Gutmann for handling our submissions and for the editorial guidance. This research was supported by the National Key R&D Program of China under grant 2020YFA0711902.

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

**Roadmap.** The appendix is organized as follows:

- Appendix A presents a comprehensive review of the literature, covering generative models, practical considerations and theoretical understandings of GANs, the relationship between GANs and particle models, and phased processes in diffusion models.

- Appendix B explains the rationale behind the choice of the latent dimension in assumption 2.2.

- Appendix C provides proofs for all the theorems, propositions, and additional theoretical results not included in the main text, which include

  - Equivalence of NSGAN with its particle model interpretation (appendix C.1)
  - Properties of particle update dynamics — the general result (appendix C.2)
  - Properties of particle update dynamics — the data-dependent results (appendix C.3)
  - Characterization of measuring-preserving maps (appendix C.4)
  - Steepness of measure-preserving map in 1-dimension (appendix C.5)
  - Steepness of measure-preserving maps in higher dimensions (appendix C.6)
  - Evolution of steepness (appendix C.7)
  - Quantitative results on how steepness impacts the severity of mode mixture (appendix C.8)
  - Local analysis of steepness at collapse (appendix C.9)

- Appendix D explores a class of suboptimal discriminators, complementing the theory of their optimal counterparts.

- Appendix E visualizes the distances between modes in datasets such as MNIST, Fashion MNIST, and CIFAR-10.

- Appendix F outlines the detailed settings for the experiments described in section 6.

- Appendix G presents additional experimental results, including the behavior of the discriminator at the collapse phase, verification of the fitting phase, a comparison with duality gaps, GAN training losses, and an evaluation of the effectiveness of the early stopping metric after applying techniques to mitigate mode collapse.

- Appendix H discusses how the analyses in this work can be extended to other divergence GANs.

- Appendix I provides visualizations of generator functions under different settings to offer intuition for section 3.2.

- Appendix J shares additional intuitions and implications.

## A  Additional Literature Review

In this section, we provide a detailed literature review.

**Generative models.** Learning the generative model based on large amounts of data is a fundamental task in machine learning and statistics. Popular techniques include Variational Autoencoders (Kingma & Welling, 2014; Chen et al., 2017; Razavi et al., 2019; Child, 2021; Simkus & Gutmann, 2024), Generative Adversarial Networks (Goodfellow et al., 2014; Radford et al., 2016; Arjovsky et al., 2017; Gulrajani et al., 2017; Nguyen et al., 2017; Ghosh et al., 2018; Lin et al., 2018; Brock et al., 2019; Karras et al., 2020), flow-based generative models (Dinh et al., 2017; Kingma & Dhariwal, 2018; Chen et al., 2019; Grathwohl et al., 2019), autoregressive models (Van den Oord et al., 2016; Van Den Oord et al., 2016), energy-based models (Xie et al., 2018; Gao et al., 2021), diffusion models (Ho et al., 2020; Song et al., 2021; Karras et al., 2022), and other variants (Srivastava et al., 2018; Sun et al., 2022). Among these models, GANs' ability for rapid sampling, unsupervised feature learning and broad applicability makes them the primary focus of this study.

**Practical considerations of GANs.** In the realm of GANs, mode collapse (Goodfellow, 2017) is arguably one of the major challenges which has received a lot of attention. It refers to the situation where the generator produces samples on only a few modes instead of the entire data distribution. The issue of mode collapse has been addressed mainly from three perspectives: modifying the network architecture, designing new objective functions and using normalization techniques. Regarding the network architecture, existing approaches involve increasing the number of generator (Ghosh et al., 2018) or discriminator (Nguyen et al., 2017), using joint architectures (Larsen et al., 2016). From the objective function side, various metrics such as the Wasserstein distance (Arjovsky et al., 2017), $f$-divergence (Nowozin et al., 2016), least squares distance (Mao et al., 2017), maximum mean discrepancy (Li et al., 2017) are employed. Normalization techniques such as batch normalization (Ioffe & Szegedy, 2015), layer normalization (Ba et al., 2016) and spectral normalization (Miyato et al., 2018) have also achieved superb empirical performance. Mode mixture (Lei et al., 2019) is another troublesome phenomenon in which the generated samples fall outside the real distribution and are thus unrealistic. Existing approaches include picking generated samples using a rejection sampling method (Tanielian et al., 2020), or generating samples with discontinuous optimal transport rather than deep neural networks (Lei et al., 2019; An et al., 2020; Gu et al., 2021).

**Theoretical understandings of GANs.** Another line of research approaches mode collapse and mode mixture by developing theoretical understandings for better analyzing and optimizing GAN training. These researches fall into two categories: landscape analysis and dynamic analysis. Landscape analysis is static because it examines the results of GAN training; it ignores the interaction between the discriminator and generator during training. For instance, Sun et al. (2020) analyzed the landscape of a family of GANs called separable-GAN. They proved that the landscape of separable-GAN has exponentially many bad basins, all of which are deemed as mode-collapse. No et al. (2021) demonstrated that Wasserstein GAN with an infinitely broad generator has no spurious stationary points by modeling both the generator and the discriminator using random feature theory. Lei et al. (2019) used results from optimal transport theory to account for mode mixture. Dynamic analysis, on the other hand, considers how the discriminator and generator interact. Franceschi et al. (2022) considered GANs from the perspective of Neural Tangent Kernel (NTK). Becker et al. (2022) suggested the "Isolated Points Model" to explain the causes of GANs' instability. Another dynamical way of modeling GANs is to regard it as a particle model (Huang & Zhang, 2023; Franceschi et al., 2023). This kind of modeling is used in conjunction with Fokker–Planck equation theories by Huang & Zhang (2023) to demonstrate the convergence of GANs to the global stationary point.

**Relationship between GANs and particle models.** There has been an emerging trend in recent years to conceptualize GANs as particle models. We present the interpretation of NSGAN as a particle model in algorithm 1, which is fundamentally grounded in the work of Yi et al. (2023). Their framework rethinks Divergence GANs from the perspective of differential equations, interpreting the evolution of generated samples as particle flows guided by vector fields derived from the discriminator's gradients. Huang & Zhang (2023) examined a similar interpretation of vanilla GAN, but did not specifically discuss NSGAN. Gao et al. (2019) used a variational gradient flow approach to analyze GANs, without placing much emphasis on the connection to particle models. Franceschi et al. (2023) unified GANs within the context of particle models and interpreted GANs as "interactive particle models."

**Phased processes in diffusion models.** Recently, analogous phase transition phenomena, akin to those elucidated in our paper, have been uncovered in diffusion models. For example, Biroli et al. (2024) showed that the generative process in diffusion models undergoes a "speciation" transition, revealing data structure from noise, followed by a "collapse" transition, converging dynamics to memorized data points, akin to condensation in a glass phase. Sclocchi et al. (2024) found that the backward diffusion process acting after a time $t$ is governed by a phase transition at some threshold time, where the probability of reconstructing high-level features suddenly drops and the reconstruction of low-level features evolves smoothly across the whole diffusion process. Li & Chen (2024) studied properties of critical windows that are are narrow time intervals in sampling during which particular features of the final image emerge.

# B    Choice of Latent Dimension

In this section, we provide the rationale behind our choice of the latent dimension in assumption 2.2. At the population level, Yi et al. (2023) demonstrated that NSGAN minimizes the $f$-divergence $D_f(p_{\text{data}}\|p_g)$ with

$$f(u) = -(u+1)\log\frac{u}{u+1} + u(1 - 2\log 2) - 1. \tag{16}$$

Let $\mu$ and $\nu$ be mutually singular measures on $\mathbb{R}^n$, Yang et al. (2022) proved that

$$D_f(\mu\|\nu) = f(0) + f^*(0) > 0, \tag{17}$$

where $f^*$ stands for the Fenchel conjugate of $f$. If the latent dimension is less than $n$, then $g_{\theta\#}p_z$ is supported on a low-dimensional manifold, so that $g_{\theta\#}p_z$ and $\nu$ will be mutually singular. Thus there is always a positive gap in $f$-divergence between $g_{\theta\#}p_z$ and $\nu$. In other words, $g_{\theta\#}p_z$ cannot approximate $\nu$ well even if the GAN model has been trained perfectly. To prevent such inherent misalignment, we assume that the latent dimension always equals $n$. Combined with the continuous data augmentation of real-world datasets, we assume that the noise prior $p_z(z)$ is an $n$-dimensional standard Gaussian distribution, denoted as $\mathcal{N}(\mathbf{0}, \boldsymbol{I}_n)$, where $n$ is the dimension of real samples.

# C    Proofs to Theorems

Here, we aggregate all the theorems presented in the paper and furnish proofs for some of them.

## C.1    Equivalence of NSGAN with Its Particle Model Interpretation

**Corollary C.1** ((Yi et al., 2023)). *The update of $g_\theta$ via applying the stop gradient operator to $\hat{Z}_i$ and descending the gradient*

$$\nabla_\theta \frac{1}{m} \sum_{i=1}^{m} \left\| g_\theta(z_i) - \hat{Z}_i \right\|_2^2 \tag{18}$$

*in algorithm 1 is equivalent to descending the gradient*

$$-\nabla_\theta \frac{1}{m} \sum_{i=1}^{m} \log\left(d_\omega(g_\theta(z_i))\right) \tag{19}$$

*in the original formulation of NSGAN.*

*Proof.* We prove by directly computing the gradient using the chain rule. In fact, we have

$$
\begin{aligned}
\nabla_\theta \frac{1}{m} \sum_{i=1}^{m} \left\| g_\theta(z_i) - \hat{Z}_i \right\|_2^2 &= \frac{2}{m} \sum_{i=1}^{m} \nabla_\theta g_\theta(z_i)^\top \cdot \left(g_\theta(z_i) - \hat{Z}_i\right) \\
&= -\frac{s}{m} \sum_{i=1}^{m} \nabla_\theta g_\theta(z_i)^\top \cdot \frac{\nabla d_\omega(\boldsymbol{Z}_i)}{d_\omega(\boldsymbol{Z}_i)} \\
&= -s \nabla_\theta \frac{1}{m} \sum_{i=1}^{m} \log\left(d_\omega(g_\theta(z_i))\right).
\end{aligned}
\tag{20}
$$

Note that in the first equation, we implicitly use the fact that $\nabla_\theta \hat{Z}_i = 0$ due to the assumption that the stop gradient operator is applied to $\hat{Z}_i$. $\qquad\square$

## C.2    Properties of Particle Update Dynamics — The General Result

**Theorem C.1.** *Assume that the discriminator is optimal, i.e., $d_*(\boldsymbol{x}) = p_{data}(\boldsymbol{x})/(p_{data}(\boldsymbol{x}) + p_g(\boldsymbol{x}))$. Denote $r(\boldsymbol{x}) = p_{data}(\boldsymbol{x})/p_g(\boldsymbol{x})$. At a point $\boldsymbol{x}$ where $r(\boldsymbol{x}) \approx 0$, $\boldsymbol{x}$ is updated following approximately $\nabla \log\left(r(\boldsymbol{x})\right)$. Conversely, when $r(\boldsymbol{x}) \gg 1$, $\boldsymbol{x}$ is updated following approximately $\nabla\left(-1/r(\boldsymbol{x})\right)$.*

*Proof.* We rewrite $\nabla d(\boldsymbol{x})/d(\boldsymbol{x})$ in terms of $r(\boldsymbol{x})$:

$$\begin{aligned}
\frac{\nabla d(\boldsymbol{x})}{d(\boldsymbol{x})} &= \frac{-p_{\text{data}}(\boldsymbol{x})\nabla p_g(\boldsymbol{x}) + p_g(\boldsymbol{x})\nabla p_{\text{data}}(\boldsymbol{x})}{(p_{\text{data}}(\boldsymbol{x}) + p_g(x))p_{\text{data}}(\boldsymbol{x})} \\
&= \nabla\Big(\frac{p_{\text{data}}(\boldsymbol{x})}{p_g(\boldsymbol{x})}\Big) \cdot \frac{p_g(\boldsymbol{x})^2}{p_{\text{data}}(\boldsymbol{x})(p_{\text{data}}(\boldsymbol{x}) + p_g(\boldsymbol{x}))} \\
&= \nabla r(\boldsymbol{x}) \cdot \frac{1}{r(\boldsymbol{x})(1 + r(\boldsymbol{x}))}.
\end{aligned} \tag{21}$$

When $r(\boldsymbol{x}) \approx 0$, we have

$$\frac{1}{r(\boldsymbol{x})(1 + r(\boldsymbol{x}))} \approx \frac{1}{r(\boldsymbol{x})}. \tag{22}$$

As a result,

$$\frac{\nabla d(\boldsymbol{x})}{d(\boldsymbol{x})} \approx \nabla \log r(\boldsymbol{x}). \tag{23}$$

When $r(\boldsymbol{x}) \gg 1$, we have

$$\frac{1}{r(\boldsymbol{x})(1 + r(\boldsymbol{x}))} \approx \frac{1}{r(\boldsymbol{x})^2}. \tag{24}$$

Consequently,

$$\frac{\nabla d(\boldsymbol{x})}{d(\boldsymbol{x})} \approx \nabla\Big(-\frac{1}{r(\boldsymbol{x})}\Big). \tag{25}$$

$\square$

We hereby outline the implications of this theorem. The value of $\log\big(r(\boldsymbol{x})\big)$ changes dramatically as $\boldsymbol{x}$ decreases from 1 to 0, leading to correspondingly large magnitudes of $\|\nabla \log\big(r(\boldsymbol{x})\big)\|_2$ when $r(\boldsymbol{x}) \approx 0$. This indicates that in the regions where $p_g(\boldsymbol{x})$ significantly exceeds $p_{\text{data}}(\boldsymbol{x})$, particles are propelled towards distant points. Conversely, $\nabla\big(-1/r(\boldsymbol{x})\big)$ changes more gradually with increasing $\boldsymbol{x}$, resulting in smaller magnitudes of $\|\nabla\big(-1/r(\boldsymbol{x})\big)\|_2$ when $r(\boldsymbol{x}) \gg 1$. In such regions where $p_g(\boldsymbol{x})$ is lower than $p_{\text{data}}(\boldsymbol{x})$, particles tend to remain relatively stationary. These align with our observations in section 3.

## C.3 Properties of Particle Update Dynamics — The Data-Dependent Results

**Proposition C.1.** *Assume that*

$$p_{data} \sim \frac{1}{4}\mathcal{N}([1,1], 0.1\boldsymbol{I}_2) + \frac{1}{4}\mathcal{N}([1,-1], 0.1\boldsymbol{I}_2) + \frac{1}{4}\mathcal{N}([-1,1], 0.1\boldsymbol{I}_2) + \frac{1}{4}\mathcal{N}([-1,-1], 0.1\boldsymbol{I}_2) \tag{26}$$

*and that $p_g \sim \mathcal{N}([0,0], 0.2\boldsymbol{I}_2)$. Let $\boldsymbol{x} = [x_1, x_2]$. Then the vector field that governs particles' update is given by*

$$\nabla r(\boldsymbol{x}) \cdot \frac{1}{r(\boldsymbol{x})(1 + r(\boldsymbol{x}))}, \tag{27}$$

*where*

$$r(\boldsymbol{x}) = \frac{1}{2} \sum_{(a,b)\in\{(\pm 1,\pm 1)\}} \exp\big(-2.5\big((x_1 - 2a)^2 + (x_2 - 2b)^2\big) + 5a^2 + 5b^2\big) \tag{28}$$

*and*

$$\nabla r(\boldsymbol{x}) = -\frac{5}{2} \sum_{(a,b)\in\{(\pm 1,\pm 1)\}} \exp\big(-2.5\big((x_1 - 2a)^2 + (x_2 - 2b)^2\big) + 5a^2 + 5b^2\big) \begin{bmatrix} x_1 - 2a \\ x_2 - 2b \end{bmatrix}. \tag{29}$$

*Proof.* For each Gaussian distribution, the density function is

$$\mathcal{N}(\boldsymbol{\mu}, \boldsymbol{\Sigma})(\boldsymbol{x}) = \frac{1}{2\pi\sqrt{\det(\boldsymbol{\Sigma})}} \exp\Big(-\frac{1}{2}(\boldsymbol{x} - \boldsymbol{\mu})^\top \boldsymbol{\Sigma}^{-1}(\boldsymbol{x} - \boldsymbol{\mu})\Big). \tag{30}$$

Here, $\boldsymbol{\mu} \in \{[1,1], [1,-1], [-1,1], [-1,-1]\}$, and $\boldsymbol{\Sigma} = 0.1\boldsymbol{I}_2$. Therefore,

$$
\begin{aligned}
\mathcal{N}([a,b], 0.1\boldsymbol{I}_2)(\boldsymbol{x}) &= \frac{1}{2\pi \cdot 0.1} \cdot \exp\Big(-\frac{1}{2 \cdot 0.1}\big((x_1 - a)^2 + (x_2 - b)^2\big)\Big) \\
&= \frac{1}{0.2\pi} \cdot \exp\big(-5\big((x_1 - a)^2 + (x_2 - b)^2\big)\big).
\end{aligned}
\tag{31}
$$

Thus,

$$
p_{\text{data}}(\boldsymbol{x}) = \frac{1}{0.8\pi} \sum_{(a,b)\in\{(\pm 1, \pm 1)\}} \exp\big(-5\big((x_1 - a)^2 + (x_2 - b)^2\big)\big).
\tag{32}
$$

For $p_g(\boldsymbol{x})$ which is normally distributed with mean $[0,0]$ and covariance $0.2\boldsymbol{I}_2$, we have

$$
p_g(\boldsymbol{x}) = \frac{1}{0.4\pi} \cdot \exp\big(-2.5(x_1^2 + x_2^2)\big).
\tag{33}
$$

Combining the above results, we have

$$
r(\boldsymbol{x}) = \frac{1}{2} \sum_{(a,b)\in\{(\pm 1, \pm 1)\}} \exp\big(-2.5\big((x_1 - 2a)^2 + (x_2 - 2b)^2\big) + 5a^2 + 5b^2\big).
\tag{34}
$$

Next, we compute $\nabla r(\boldsymbol{x})$:

$$
\nabla r(\boldsymbol{x}) = \frac{1}{2} \sum_{(a,b)\in\{(\pm 1, \pm 1)\}} \nabla \exp\big(-2.5\big((x_1 - 2a)^2 + (x_2 - 2b)^2\big) + 5a^2 + 5b^2\big).
\tag{35}
$$

For each term $\exp\big(-2.5\big((x_1 - 2a)^2 + (x_2 - 2b)^2\big) + 5a^2 + 5b^2\big)$, its gradient is:

$$
\begin{aligned}
&\nabla \exp\big(-2.5\big((x_1 - 2a)^2 + (x_2 - 2b)^2\big) + 5a^2 + 5b^2\big) \\
&= -5 \exp\big(-2.5\big((x_1 - 2a)^2 + (x_2 - 2b)^2\big) + 5a^2 + 5b^2\big) \begin{bmatrix} x_1 - 2a \\ x_2 - 2b \end{bmatrix}.
\end{aligned}
\tag{36}
$$

Thus,

$$
\nabla r(\boldsymbol{x}) = -\frac{5}{2} \sum_{(a,b)\in\{(\pm 1, \pm 1)\}} \exp\big(-2.5\big((x_1 - 2a)^2 + (x_2 - 2b)^2\big) + 5a^2 + 5b^2\big) \begin{bmatrix} x_1 - 2a \\ x_2 - 2b \end{bmatrix}.
\tag{37}
$$

Putting the expressions of $r(\boldsymbol{x})$ and $\nabla r(\boldsymbol{x})$ together, we will have

$$
\nabla r(\boldsymbol{x}) \cdot \frac{1}{r(\boldsymbol{x})(1 + r(\boldsymbol{x}))}.
\tag{38}
$$

When we take a closer look at the numerator $\nabla r(\boldsymbol{x})$, we observe that it is a weighted sum of the vectors originating from $\boldsymbol{x}$ and pointing towards two times the centers of the four modes, which are $(2,2)$, $(2,-2)$, $(-2,2)$, and $(-2,-2)$. Due to the exponential decay property of the exponential function, the influence of these vectors diminishes rapidly with distance. Consequently, the vector field is predominantly influenced by the mode in the same quadrant as $\boldsymbol{x}$. Specifically, if we assume without loss of generality that $\boldsymbol{x}$ lies in the first quadrant, the vector field will be approximately $[2 - x_1, 2 - x_2]^\top$, up to a scaling factor. $\qquad\square$

**Proposition C.2.** *Assume that*

$$
p_{data} \sim \frac{1}{4}\mathcal{N}([1,1], 0.1\boldsymbol{I}_2) + \frac{1}{4}\mathcal{N}([1,-1], 0.1\boldsymbol{I}_2) + \frac{1}{4}\mathcal{N}([-1,1], 0.1\boldsymbol{I}_2) + \frac{1}{4}\mathcal{N}([-1,-1], 0.1\boldsymbol{I}_2)
\tag{39}
$$

*and that $p_g \sim \mathcal{U}\big([-2,2] \times [-2,2]\big)$. Let $\boldsymbol{x} = [x_1, x_2]$. Then the vector field that governs particles' update is given by*

$$
\nabla r(\boldsymbol{x}) \cdot \frac{1}{r(\boldsymbol{x})(1 + r(\boldsymbol{x}))},
\tag{40}
$$

*where*

$$r(\boldsymbol{x}) = \frac{20}{\pi} \sum_{(a,b) \in \{(\pm 1, \pm 1)\}} \exp\big(-5\big((x_1 - a)^2 + (x_2 - b)^2\big)\big) \cdot \mathbf{1}_{\boldsymbol{x} \in [-2,2] \times [-2,2]} \tag{41}$$

*and*

$$\nabla r(\boldsymbol{x}) = -\frac{200}{\pi} \sum_{(a,b) \in \{(\pm 1, \pm 1)\}} \exp\big(-5\big((x_1 - a)^2 + (x_2 - b)^2\big)\big) \begin{bmatrix} x_1 - a \\ x_2 - b \end{bmatrix} \cdot \mathbf{1}_{\boldsymbol{x} \in [-2,2] \times [-2,2]}. \tag{42}$$

*Proof.* For each Gaussian distribution, the density function is

$$\mathcal{N}(\boldsymbol{\mu}, \boldsymbol{\Sigma})(\boldsymbol{x}) = \frac{1}{2\pi \sqrt{\det(\boldsymbol{\Sigma})}} \exp\Big(-\frac{1}{2}(\boldsymbol{x} - \boldsymbol{\mu})^\top \boldsymbol{\Sigma}^{-1}(\boldsymbol{x} - \boldsymbol{\mu})\Big). \tag{43}$$

Here, $\boldsymbol{\mu} \in \{[1,1], [1,-1], [-1,1], [-1,-1]\}$, and $\boldsymbol{\Sigma} = 0.1\boldsymbol{I}_2$. Therefore,

$$\begin{aligned} \mathcal{N}([a,b], 0.1\boldsymbol{I}_2)(\boldsymbol{x}) &= \frac{1}{2\pi \cdot 0.1} \cdot \exp\Big(-\frac{1}{2 \cdot 0.1}\big((x_1 - a)^2 + (x_2 - b)^2\big)\Big) \\ &= \frac{1}{0.2\pi} \cdot \exp\big(-5\big((x_1 - a)^2 + (x_2 - b)^2\big)\big). \end{aligned} \tag{44}$$

Thus,

$$p_{\text{data}}(\boldsymbol{x}) = \frac{1}{0.8\pi} \sum_{(a,b) \in \{(\pm 1, \pm 1)\}} \exp\big(-5\big((x_1 - a)^2 + (x_2 - b)^2\big)\big) \tag{45}$$

For $p_g(\boldsymbol{x})$ which is uniformly distributed, we have

$$p_g(\boldsymbol{x}) = \frac{1}{16} \cdot \mathbf{1}_{\boldsymbol{x} \in [-2,2] \times [-2,2]}. \tag{46}$$

Combining the above results,

$$r(\boldsymbol{x}) = \frac{20}{\pi} \sum_{(a,b) \in \{(\pm 1, \pm 1)\}} \exp\big(-5\big((x_1 - a)^2 + (x_2 - b)^2\big)\big) \cdot \mathbf{1}_{\boldsymbol{x} \in [-2,2] \times [-2,2]}. \tag{47}$$

Now, we compute $\nabla r(\boldsymbol{x})$:

$$\nabla r(\boldsymbol{x}) = \frac{20}{\pi} \sum_{(a,b) \in \{(\pm 1, \pm 1)\}} \nabla \exp\big(-5\big((x_1 - a)^2 + (x_2 - b)^2\big)\big) \mathbf{1}_{\boldsymbol{x} \in [-2,2] \times [-2,2]}. \tag{48}$$

For each term $\exp\big(-5\big((x_1 - a)^2 + (x_2 - b)^2\big)\big)$, its gradient is:

$$\nabla \exp\big(-5\big((x_1 - a)^2 + (x_2 - b)^2\big)\big) = -10 \exp\big(-5\big((x_1 - a)^2 + (x_2 - b)^2\big)\big) \begin{bmatrix} x_1 - a \\ x_2 - b \end{bmatrix}. \tag{49}$$

Thus,

$$\nabla r(\boldsymbol{x}) = -\frac{200}{\pi} \sum_{(a,b) \in \{(\pm 1, \pm 1)\}} \exp\big(-5\big((x_1 - a)^2 + (x_2 - b)^2\big)\big) \begin{bmatrix} x_1 - a \\ x_2 - b \end{bmatrix} \cdot \mathbf{1}_{\boldsymbol{x} \in [-2,2] \times [-2,2]}. \tag{50}$$

Putting the expressions of $r(\boldsymbol{x})$ and $\nabla r(\boldsymbol{x})$ together, we will have

$$\nabla r(\boldsymbol{x}) \cdot \frac{1}{r(\boldsymbol{x})(1 + r(\boldsymbol{x}))}. \tag{51}$$

When we take a closer look at the numerator $\nabla r(\boldsymbol{x})$, we observe that it is a weighted sum of the vectors originating from $\boldsymbol{x}$ and pointing towards the centers of the four modes, which are $(1,1)$, $(1,-1)$, $(-1,1)$, and $(-1,-1)$. Due to the exponential decay property of the exponential function, the influence of these vectors diminishes rapidly with distance. Consequently, the vector field is predominantly influenced by the mode in the same quadrant as $\boldsymbol{x}$. Specifically, if we assume without loss of generality that $\boldsymbol{x}$ lies in the first quadrant, the vector field will be approximately $[1 - x_1, 1 - x_2]^\top$, up to a scaling factor. □

**Proposition C.3.** *Assume that*

$$p_{data} \sim \frac{1}{4}\mathcal{N}([1,1], 0.1\boldsymbol{I}_2) + \frac{1}{4}\mathcal{N}([1,-1], 0.1\boldsymbol{I}_2) + \frac{1}{4}\mathcal{N}([-1,1], 0.1\boldsymbol{I}_2) + \frac{1}{4}\mathcal{N}([-1,-1], 0.1\boldsymbol{I}_2) \tag{52}$$

*and that* $p_g \sim \mathcal{N}([1,1], \boldsymbol{I}_2)$. *Let* $\boldsymbol{x} = [x_1, x_2]$. *Then the vector field that governs particles' update is given by*

$$\nabla r(\boldsymbol{x}) \cdot \frac{1}{r(\boldsymbol{x})(1 + r(\boldsymbol{x}))}, \tag{53}$$

*where*

$$r(\boldsymbol{x}) = \frac{5}{2} \sum_{(a,b)\in\{(\pm 1,\pm 1)\}} \exp\left(-\frac{9}{2}\left(\left(x_1 - \frac{10a-1}{9}\right)^2 + \left(x_2 - \frac{10b-1}{9}\right)^2\right) + \frac{5}{9}(a-1)^2 + \frac{5}{9}(b-1)^2\right) \tag{54}$$

*and*

$$\nabla r(\boldsymbol{x}) = -\frac{45}{2} \cdot$$

$$\sum_{(a,b)\in\{(\pm 1,\pm 1)\}} \exp\left(-\frac{9}{2}\left(\left(x_1 - \frac{10a-1}{9}\right)^2 + \left(x_2 - \frac{10b-1}{9}\right)^2\right) + \frac{5}{9}(a-1)^2 + \frac{5}{9}(b-1)^2\right) \begin{bmatrix} x_1 - \frac{10a-1}{9} \\ x_2 - \frac{10b-1}{9} \end{bmatrix}. \tag{55}$$

*Proof.* For each Gaussian distribution, the density function is

$$\mathcal{N}(\boldsymbol{\mu}, \boldsymbol{\Sigma})(\boldsymbol{x}) = \frac{1}{2\pi\sqrt{\det(\boldsymbol{\Sigma})}} \exp\left(-\frac{1}{2}(\boldsymbol{x} - \boldsymbol{\mu})^\top \boldsymbol{\Sigma}^{-1}(\boldsymbol{x} - \boldsymbol{\mu})\right). \tag{56}$$

Here, $\boldsymbol{\mu} \in \{[1,1], [1,-1], [-1,1], [-1,-1]\}$, and $\boldsymbol{\Sigma} = 0.1\boldsymbol{I}_2$. Therefore,

$$\begin{aligned} \mathcal{N}([a,b], 0.1\boldsymbol{I}_2)(\boldsymbol{x}) &= \frac{1}{2\pi \cdot 0.1} \cdot \exp\left(-\frac{1}{2 \cdot 0.1}\left((x_1 - a)^2 + (x_2 - b)^2\right)\right) \\ &= \frac{1}{0.2\pi} \cdot \exp\left(-5\left((x_1 - a)^2 + (x_2 - b)^2\right)\right). \end{aligned} \tag{57}$$

Thus,

$$p_{\text{data}}(\boldsymbol{x}) = \frac{1}{0.8\pi} \sum_{(a,b)\in\{(\pm 1,\pm 1)\}} \exp\left(-5\left((x_1 - a)^2 + (x_2 - b)^2\right)\right). \tag{58}$$

For $p_g(\boldsymbol{x})$ which is normally distributed with mean $[1,1]$ and covariance $\boldsymbol{I}_2$, we have

$$p_g(\boldsymbol{x}) = \frac{1}{2\pi} \cdot \exp\left(-0.5\left((x_1 - 1)^2 + (x_2 - 1)^2\right)\right). \tag{59}$$

Combining the above results, we have

$$r(\boldsymbol{x}) = \frac{5}{2} \sum_{(a,b)\in\{(\pm 1,\pm 1)\}} \exp\left(-\frac{9}{2}\left(\left(x_1 - \frac{10a-1}{9}\right)^2 + \left(x_2 - \frac{10b-1}{9}\right)^2\right) + \frac{5}{9}(a-1)^2 + \frac{5}{9}(b-1)^2\right). \tag{60}$$

Next, we compute $\nabla r(\boldsymbol{x})$:

$$\nabla r(\boldsymbol{x}) = \frac{5}{2} \sum_{(a,b)\in\{(\pm 1,\pm 1)\}} \nabla \exp\left(-\frac{9}{2}\left(\left(x_1 - \frac{10a-1}{9}\right)^2 + \left(x_2 - \frac{10b-1}{9}\right)^2\right) + \frac{5}{9}(a-1)^2 + \frac{5}{9}(b-1)^2\right). \tag{61}$$

For each term on the right-hand side, its gradient is:

$$\begin{aligned} &\nabla \exp\left(-\frac{9}{2}\left(\left(x_1 - \frac{10a-1}{9}\right)^2 + \left(x_2 - \frac{10b-1}{9}\right)^2\right) + \frac{5}{9}(a-1)^2 + \frac{5}{9}(b-1)^2\right) \\ &= -9 \cdot \exp\left(-\frac{9}{2}\left(\left(x_1 - \frac{10a-1}{9}\right)^2 + \left(x_2 - \frac{10b-1}{9}\right)^2\right) + \frac{5}{9}(a-1)^2 + \frac{5}{9}(b-1)^2\right) \begin{bmatrix} x_1 - (10a-1)/9 \\ x_2 - (10b-1)/9 \end{bmatrix}. \end{aligned} \tag{62}$$

Thus,

$$\nabla r(\boldsymbol{x}) = -\frac{45}{2} \cdot$$

$$\sum_{(a,b) \in \{(\pm 1, \pm 1)\}} \exp\left(-\frac{9}{2}\left(\left(x_1 - \frac{10a-1}{9}\right)^2 + \left(x_2 - \frac{10b-1}{9}\right)^2\right) + \frac{5}{9}(a-1)^2 + \frac{5}{9}(b-1)^2\right) \begin{bmatrix} x_1 - \dfrac{10a-1}{9} \\ x_2 - \dfrac{10b-1}{9} \end{bmatrix}. \tag{63}$$

Putting the expressions of $r(\boldsymbol{x})$ and $\nabla r(\boldsymbol{x})$ together, we will have

$$\nabla r(\boldsymbol{x}) \cdot \frac{1}{r(\boldsymbol{x})(1 + r(\boldsymbol{x}))}. \tag{64}$$

When we take a closer look at the numerator $\nabla r(\boldsymbol{x})$, we observe that it is a weighted sum of the vectors originating from $\boldsymbol{x}$ and pointing towards $(1,1)$, $(-11/9, 1)$, $(1, -11/9)$, and $(-11/9, -11/9)$, respectively. Due to the exponential decay property of the exponential function, the influence of these vectors diminishes rapidly with distance. Consequently, the vector field is predominantly influenced by the mode in the same quadrant as $\boldsymbol{x}$. Specifically, if we assume without loss of generality that $\boldsymbol{x}$ lies in the first quadrant, the vector field will be approximately $[1 - x_1, 1 - x_2]^\top$, up to a scaling factor. $\qquad\square$

**Proposition C.4.** *Assume that*

$$p_{data} \sim \frac{1}{4}\mathcal{N}([3,3], 0.1\boldsymbol{I}_2) + \frac{1}{4}\mathcal{N}([3,-3], 0.1\boldsymbol{I}_2) + \frac{1}{4}\mathcal{N}([-3,3], 0.1\boldsymbol{I}_2) + \frac{1}{4}\mathcal{N}([-3,-3], 0.1\boldsymbol{I}_2) \tag{65}$$

*and that $p_g \sim \mathcal{N}([3,3], 3\boldsymbol{I}_2)$. Let $\boldsymbol{x} = [x_1, x_2]$. Then the vector field that governs particles' update is given by*

$$\nabla r(\boldsymbol{x}) \cdot \frac{1}{r(\boldsymbol{x})(1 + r(\boldsymbol{x}))}, \tag{66}$$

*where*

$$r(\boldsymbol{x}) = \frac{15}{4} \sum_{(a,b) \in \{(\pm 3, \pm 3)\}} \exp\left(-\frac{29}{6}\left(\left(x_1 - \frac{30a-3}{29}\right)^2 + \left(x_2 - \frac{30b-3}{29}\right)^2\right) + \frac{5}{29}(a-3)^2 + \frac{5}{29}(b-3)^2\right) \tag{67}$$

*and*

$$\nabla r(\boldsymbol{x}) = -\frac{145}{4} \cdot$$

$$\sum_{(a,b) \in \{(\pm 3, \pm 3)\}} \exp\left(-\frac{29}{6}\left(\left(x_1 - \frac{30a-3}{29}\right)^2 + \left(x_2 - \frac{30b-3}{29}\right)^2\right) + \frac{5}{29}(a-3)^2 + \frac{5}{29}(b-3)^2\right) \begin{bmatrix} x_1 - \dfrac{30a-3}{29} \\ x_2 - \dfrac{30b-3}{29} \end{bmatrix}. \tag{68}$$

*Proof.* For each Gaussian distribution, the density function is

$$\mathcal{N}(\boldsymbol{\mu}, \boldsymbol{\Sigma})(\boldsymbol{x}) = \frac{1}{2\pi\sqrt{\det(\boldsymbol{\Sigma})}} \exp\left(-\frac{1}{2}(\boldsymbol{x} - \boldsymbol{\mu})^\top \boldsymbol{\Sigma}^{-1}(\boldsymbol{x} - \boldsymbol{\mu})\right). \tag{69}$$

Here, $\boldsymbol{\mu} \in \{[3,3], [3,-3], [-3,3], [-3,-3]\}$, and $\boldsymbol{\Sigma} = 0.1\boldsymbol{I}_2$. Therefore,

$$\begin{aligned}
\mathcal{N}([a,b], 0.1\boldsymbol{I}_2)(\boldsymbol{x}) &= \frac{1}{2\pi \cdot 0.1} \cdot \exp\left(-\frac{1}{2 \cdot 0.1}\left((x_1 - a)^2 + (x_2 - b)^2\right)\right) \\
&= \frac{1}{0.2\pi} \cdot \exp\left(-5\left((x_1 - a)^2 + (x_2 - b)^2\right)\right).
\end{aligned} \tag{70}$$

Thus,

$$p_{\text{data}}(\boldsymbol{x}) = \frac{1}{0.8\pi} \sum_{(a,b) \in \{(\pm 3, \pm 3)\}} \exp\left(-5\left((x_1 - a)^2 + (x_2 - b)^2\right)\right). \tag{71}$$

For $p_g(\boldsymbol{x})$ which is normally distributed with mean $[3,3]$ and covariance $3\boldsymbol{I}_2$, we have

$$p_g(\boldsymbol{x}) = \frac{1}{6\pi} \cdot \exp\left(-\frac{1}{6}\left((x_1 - 3)^2 + (x_2 - 3)^2\right)\right). \tag{72}$$

Combining the above results, we have

$$r(\boldsymbol{x}) = \frac{15}{4} \sum_{(a,b)\in\{(\pm 3,\pm 3)\}} \exp\left(-\frac{29}{6}\left(\left(x_1 - \frac{30a-3}{29}\right)^2 + \left(x_2 - \frac{30b-3}{29}\right)^2\right) + \frac{5}{29}(a-3)^2 + \frac{5}{29}(b-3)^2\right). \tag{73}$$

Next, we compute $\nabla r(\boldsymbol{x})$:

$$\frac{15}{4} \sum_{(a,b)\in\{(\pm 3,\pm 3)\}} \nabla \exp\left(-\frac{29}{6}\left(\left(x_1 - \frac{30a-3}{29}\right)^2 + \left(x_2 - \frac{30b-3}{29}\right)^2\right) + \frac{5}{29}(a-3)^2 + \frac{5}{29}(b-3)^2\right). \tag{74}$$

For each term on the right-hand side, its gradient is:

$$\exp\left(-\frac{29}{6}\left(\left(x_1 - \frac{30a-3}{29}\right)^2 + \left(x_2 - \frac{30b-3}{29}\right)^2\right) + \frac{5}{29}(a-3)^2 + \frac{5}{29}(b-3)^2\right) =$$

$$-\frac{29}{3}\exp\left(-\frac{29}{6}\left(\left(x_1 - \frac{30a-3}{29}\right)^2 + \left(x_2 - \frac{30b-3}{29}\right)^2\right) + \frac{5}{29}(a-3)^2 + \frac{5}{29}(b-3)^2\right)\begin{bmatrix} x_1 - (30a-3)/29 \\ x_2 - (30b-3)/29 \end{bmatrix}. \tag{75}$$

Thus,

$$\nabla r(\boldsymbol{x}) = -\frac{145}{4}\cdot$$

$$\sum_{(a,b)\in\{(\pm 3,\pm 3)\}} \exp\left(-\frac{29}{6}\left(\left(x_1 - \frac{30a-3}{29}\right)^2 + \left(x_2 - \frac{30b-3}{29}\right)^2\right) + \frac{5}{29}(a-3)^2 + \frac{5}{29}(b-3)^2\right)\begin{bmatrix} x_1 - \dfrac{30a-3}{29} \\ x_2 - \dfrac{30b-3}{29} \end{bmatrix}. \tag{76}$$

Putting the expressions of $r(\boldsymbol{x})$ and $\nabla r(\boldsymbol{x})$ together, we will have

$$\nabla r(\boldsymbol{x}) \cdot \frac{1}{r(\boldsymbol{x})(1 + r(\boldsymbol{x}))}. \tag{77}$$

When we take a closer look at the numerator $\nabla r(\boldsymbol{x})$, we observe that it is a weighted sum of the vectors originating from $\boldsymbol{x}$ and pointing towards $(27/29, 27/29)$, $(-33/29, 27/29)$, $(27/29, -33/29)$, and $(-33/29, -33/29)$, respectively. Due to the exponential decay property of the exponential function, the influence of these vectors diminishes rapidly with distance. Consequently, the vector field is predominantly influenced by the mode in the same quadrant as $\boldsymbol{x}$. Specifically, if we assume without loss of generality that $\boldsymbol{x}$ lies in the first quadrant, the vector field will be approximately $[27/29 - x_1, 27/29 - x_2]^\top$, up to a scaling factor. Regarding the term $1 + r(\boldsymbol{x})$ in the denominator, we observe that its magnitude is large when $\boldsymbol{x}$ is far from the coordinates $x_1 = 0$, $x_2 = 0$, and the centers of the modes. This increased magnitude compared to the scenario in proposition C.3 explains the overall weakening of the attraction intensity near all the modes. $\qquad\square$

## C.4 Characterization of Measuring-Preserving Maps

**Lemma C.1** ((Durrett, 2019))**.** *Let $\boldsymbol{X}$ be a random variable taking values on $\mathbb{R}$ and let $F_{\boldsymbol{X}}(x)$ be its CDF. Then*

$$F_{\boldsymbol{X}}^{-1}\big(\mathcal{U}(0,1)\big) \sim \boldsymbol{X} \tag{78}$$

*and*

$$F_X(\boldsymbol{X}) \sim \mathcal{U}(0,1), \tag{79}$$

*where $\mathcal{U}(0,1)$ denotes the uniform distribution on $(0,1)$.*

**Theorem C.2.** *Let $\Phi(x)$ denotes the cumulative distribution function (CDF) of $\mathcal{N}(0,1)$ and let $\Psi(x)$ be that of $p_{data}(x)$. If $g$ satisfies $g_{\#}p_z = p_{data}$, then $g = \Psi^{-1} \circ h \circ \Phi$, where $h$ is a measure-preserving map of $\mathcal{U}(0,1)$, i.e., the uniform distribution on $(0,1)$.*

*Proof.* We only need to show that $\Psi \circ g \circ \Phi^{-1}$ is a measure-preserving map of $\mathcal{U}(0,1)$. In fact, by lemma C.1, we have

$$(\Psi \circ g \circ \Phi^{-1})_{\#}\mathcal{U}(0,1) = (\Psi \circ g)_{\#}p_z = \Psi_{\#}p_{\text{data}} = \mathcal{U}(0,1). \tag{80}$$

$\square$

### C.5 Steepness of Measure-Preserving Map in $1$-Dimension

**Theorem C.3.** *Assume that the real data distribution $p_{data}(x)$ satisfies assumption 2.1 with $n = 1$ and separation condition $\Delta = 6\sigma$. Let $\Phi(x)$ and $\Psi(x)$ denote the cumulative distribution functions (CDFs) of $\mathcal{N}(0,1)$ and $p_{data}(x)$, respectively. Define $g(x) := \Psi^{-1}(\Phi(x))$. Then, there exists a point $x_* \in \mathbb{R}$ such that the steepness of $g$ at $x_*$ satisfies:*

$$\mathcal{S}_g(x_*) \geq \min_{1 \leq i \leq N-1} \sigma \cdot \exp\left(\frac{(x_{i+1} - x_i)^2}{8\sigma^2}\right) \cdot \exp(-q^2), \tag{81}$$

*where $q$ is the $(1 - 1/N)$th quantile of the standard Gaussian distribution.*

*Proof.* Instead of computing the derivative of $g$, we compute that of $g^{-1}$. By the formula for the derivative of inverse functions, we have that for any $y \in \mathbb{R}$,

$$
\begin{aligned}
(g^{-1})'(y) &= \frac{\Psi'(y)}{\Phi'\left(\Phi^{-1}(\Psi(y))\right)} \\
&= \frac{1}{N\sigma} \sum_{i=1}^{N} \exp\left(-\frac{(y - x_i)^2}{2\sigma^2}\right) \cdot \exp\left(\frac{(\Phi^{-1}(\Psi(y)))^2}{2}\right) \\
&\leq \max_{1 \leq i \leq N-1} \frac{1}{N\sigma} \cdot N \cdot \exp\left(-\frac{(x_{i+1} - x_i)^2}{8\sigma^2}\right) \cdot \exp\left(\frac{(\Phi^{-1}(\Psi((x_i + x_{i+1})/2)))^2}{2}\right) \\
&\leq \max_{1 \leq i \leq N-1} \frac{1}{\sigma} \cdot \exp\left(-\frac{(x_{i+1} - x_i)^2}{8\sigma^2}\right) \cdot \exp(q^2).
\end{aligned}
\tag{82}
$$

where $q$ is the $(1 - 1/N)$th quantile of the standard Gaussian distribution. Again, by the formula for the derivative of inverse functions, there exists $x_* \in \mathbb{R}$ such that

$$\mathcal{S}_g(x_*) \geq \min_{1 \leq i \leq N-1} \sigma \cdot \exp\left(\frac{(x_{i+1} - x_i)^2}{8\sigma^2}\right) \cdot \exp(-q^2). \tag{83}$$

$\square$

### C.6 Steepness of Measure-Preserving Maps in Higher Dimensions

The standard result in (Durrett, 2019) specifically addresses the case of lemma C.2 where $K = 1$. And it can be straightforwardly extended to encompass any $K$.

**Lemma C.2** ((Durrett, 2019)). *Let $\boldsymbol{X} \sim \rho(\boldsymbol{x})\mathrm{d}\boldsymbol{x}$ be a $n$-dimensional random vector. Let $\mathcal{D} \subset \mathbb{R}^n$ satisfy $\mathbb{P}(\boldsymbol{X} \in \mathcal{D}) = 1$. Assume that the map*

$$\varphi: \mathcal{D} = \biguplus_{k=1}^{K} \mathcal{D}_i \to \mathbb{R}^n \tag{84}$$

*satisfies the following requirements: for each $1 \leq k \leq K$, $\varphi := \varphi|_{\mathcal{D}_k}$ is injective and its inverse function is continuously differentiable. Then the probability density function of $\boldsymbol{Y} = \varphi(\boldsymbol{X})$ is*

$$\rho_{\boldsymbol{Y}}(\boldsymbol{y}) = \sum_{k=1}^{K} \rho_{\boldsymbol{X}}\left(\varphi^{-1}(\boldsymbol{y})\right) \cdot \left|\det\left(J_{\varphi_k^{-1}(\boldsymbol{y})}\right)\right| \cdot \mathbf{1}_{\varphi(\mathcal{D}_k)}(\boldsymbol{y}). \tag{85}$$

*Equivalently, for any $\boldsymbol{x} \in \mathcal{D}$,*

$$\rho_{\boldsymbol{Y}}(\varphi(\boldsymbol{x})) = \sum_{k=1}^{K} \rho_{\boldsymbol{X}}(\boldsymbol{x}) \cdot |\det(J_\varphi(\boldsymbol{x}))|^{-1} \cdot \mathbf{1}_{\varphi(\mathcal{D}_k)}(\varphi(\boldsymbol{x})). \tag{86}$$

**Theorem C.4.** *Assume that the real data distribution $p_{data}(\boldsymbol{x})$ satisfies assumption 2.1, and that the noise prior $p_z(\boldsymbol{z})$ is the truncated Gaussian $\mathcal{N}_r(\boldsymbol{0}, \boldsymbol{I}_n)$ defined on the $n$-dimensional ball $\mathcal{B}_r(\boldsymbol{0})$. Without loss of generality, suppose $\boldsymbol{x}_i \neq \boldsymbol{0}$ for all $1 \leq i \leq N$. Let $g: \mathcal{B}_r(\boldsymbol{0}) \to \mathbb{R}^n$ be a continuously differentiable, piecewise injective function satisfying $g_\# p_z = p_{data}$. Then, there exists a point $\boldsymbol{x}_* \in \mathbb{R}^n$ such that the steepness $\mathcal{S}_g(\boldsymbol{x}_*)$ satisfies $\mathcal{S}_g(\boldsymbol{x}_*) \geq M$, where*

$$M = \delta \cdot \sigma \cdot \sqrt{2\pi} \cdot \max_{\lambda \in [0,2]} \min_{1 \leq i \leq N} \exp\left(\frac{\|\lambda \bar{\boldsymbol{x}} - \boldsymbol{x}_i\|_2^2}{2n\sigma^2}\right). \tag{87}$$

*Here, $\bar{\boldsymbol{x}} = \sum_{i=1}^{N} \boldsymbol{x}_i / N$ is the mean of the mode centers, and $\delta = \exp(-r^2/2)/\sqrt{2\pi}$ accounts for the truncation of the Gaussian distribution.*

*Proof.* Let $\mathcal{D}_k$ $(1 \leq k \leq K)$ be a partition of $\mathcal{B}_r(\boldsymbol{0})$ such that for each $1 \leq k \leq K$, $g|_{\mathcal{D}_k}$ is injective. We regard $g$ as the composition of two functions $g := g_2 \circ g_1$. Here, $g_1: \mathcal{B}_r(\boldsymbol{0}) \to (0,1)^n$ satisfies

$$g_1(\boldsymbol{x}) = g_1(x_1, x_2, \ldots, x_n) = (\Phi_r(x_1), \Phi_r(x_2), \ldots, \Phi_r(x_n)), \tag{88}$$

where $\Phi_r(\cdot)$ is the cumulative density function of the 1-dimensional standard Gaussian distribution truncated in $(-r, r)$. It is straightforward to show that the derivative of $\Phi_r$ has a positive lower bound, say,

$$\delta := \frac{1}{\sqrt{2\pi}} \exp\left(-\frac{r^2}{2}\right). \tag{89}$$

Thus $|\det J_{g_1}(\boldsymbol{x})| \geq \delta^n$ for any $\boldsymbol{x} \in \mathcal{B}_r(\boldsymbol{0})$.

By lemma C.1, $g_{1\#} p_z = \pi$, where $\pi$ is the uniform distribution on $(0,1)^n$. In the rest of the proof, we direct our focus to $g_2: (0,1)^n \to \mathbb{R}^n$, which satisfies $g_{2\#}\pi = p_{data}(x)$. Because $g_2 = g \circ g_1^{-1}$ and $g$ is injective on $\mathcal{D}_i$ $(1 \leq i \leq N)$, we conclude that $g_2$ is injective on $g_1(\mathcal{D}_k)$ $(1 \leq k \leq K)$. By applying lemma C.2 to $g_2$ and $g_1(\mathcal{D}_k)$ $(1 \leq k \leq K)$, we deduce that for $\boldsymbol{y} \in (0,1)^n$,

$$p_{data}(g_2(\boldsymbol{y})) = \sum_{k=1}^{K} \frac{1}{|\det(J_{g_2}(\boldsymbol{y}))|} \cdot \mathbf{1}_{g_2(g_1(\mathcal{D}_k))}(g_2(\boldsymbol{y})) \geq \sum_{k=1}^{K} \frac{1}{|\det(J_{g_2}(\boldsymbol{y}))|} \cdot \mathbf{1}_{g_1(\mathcal{D}_k)}(\boldsymbol{y}) = \frac{1}{|\det(J_{g_2}(\boldsymbol{y}))|}. \tag{90}$$

Let $\mathcal{B}_R(\boldsymbol{0})$ be the $n$-dimensional open ball centered at the origin with radius $R = 2 \cdot \max_{1 \leq i \leq N} \|\boldsymbol{x}_i\|_2$. We consider the point $\boldsymbol{y}_0$ satisfying

$$g_2(\boldsymbol{y}_0) = \arg \max_{\boldsymbol{x} \in \mathcal{B}_R(\boldsymbol{0})} \min_{1 \leq i \leq N} \|\boldsymbol{x} - \boldsymbol{x}_i\|_2. \tag{91}$$

If there are many of them, we randomly pick one. Let $\bar{\boldsymbol{x}} = \sum_{i=1}^{N} \boldsymbol{x}_i / N$. For this $\boldsymbol{y}_0$, we have

$$p_{data}(g_2(\boldsymbol{y}_0)) \leq \hat{f}(\lambda \bar{\boldsymbol{x}}) = \frac{1}{N} \sum_{i=1}^{N} \frac{1}{(2\pi\sigma^2)^{n/2}} \cdot \exp\left(-\frac{\|\lambda \bar{\boldsymbol{x}} - \boldsymbol{x}_i\|_2^2}{2\sigma^2}\right) \tag{92}$$

for any $\lambda \in [0, 2]$.

Hence

$$|\det(J_{g_2}(\boldsymbol{y}_0))| \geq p_{data}(g_2(\boldsymbol{y}_0))^{-1} \geq (2\pi\sigma^2)^{n/2} \cdot \min_{1 \leq i \leq N} \exp\left(\frac{\|\lambda \bar{\boldsymbol{x}} - \boldsymbol{x}_i\|_2^2}{2\sigma^2}\right). \tag{93}$$

Recall that we have $|\det(J_{g_1}(g_2(\boldsymbol{y}_0)))| \geq \delta^n$, where $\delta = \frac{1}{\sqrt{2\pi}} \exp\left(-\frac{r^2}{2}\right)$.

Combine the above results and we have

$$| \det(J_g(\boldsymbol{y}_0))| = | \det(J_{g_2}(\boldsymbol{y}_0)) \det(J_{g_1}(g_2(\boldsymbol{y}_0)))| \geq (\sqrt{2\pi}\sigma\delta)^n \cdot \min_{1 \leq i \leq N} \exp\Big(\frac{\|\lambda\bar{\boldsymbol{x}} - \boldsymbol{x}_i\|_2^2}{2\sigma^2}\Big). \tag{94}$$

If $\mathcal{S}_g(\boldsymbol{y}_0) < M$, then by the property that the determinant of a matrix is bounded above by the $n$th power of its spectral norm, we have

$$| \det(J_g(\boldsymbol{y}_0))| < M^n. \tag{95}$$

However, substituting the expression for $M$ into this inequality leads to a contradiction with the previously derived bounds. Therefore, we conclude that the assumption $\mathcal{S}_g(\boldsymbol{y}_0) < M$ is invalid. Let $\boldsymbol{x}_* = \boldsymbol{y}_0$, which completes the proof. $\qquad\square$

We remark that by choosing $\lambda = 1$, the lower bound becomes

$$M = \delta \cdot \sigma \cdot \sqrt{2\pi} \cdot \min_{1 \leq i \leq N} \exp\Big(\frac{\|\bar{\boldsymbol{x}} - \boldsymbol{x}_i\|_2^2}{2n\sigma^2}\Big), \tag{96}$$

which provides a useful baseline as it directly relates the bound to the distance between the mean of all modes, $\bar{\boldsymbol{x}}$, and individual modes, offering an interpretable measure of steepness.

### C.7 Evolution of Steepness

In theorem C.5, we first analyze a simplified setting where $p_{\text{data}}$ composed of a single mode with large variance, and derive a recurrence relation for the steepness $k_t$ of the generator $g$ at $\boldsymbol{x} = 0$. This relation provides two key implications: (i) the steepness $k_t$ increases monotonically over time, ensuring that the generator progressively adapts to map the latent distribution $p_z$ to the real data distribution $p_{\text{data}}$; and (ii) as $t \to \infty$, $k_t$ converges to $k_*$[5], the steepness of the optimal generator, indicating that the generator eventually reaches the steepness required for effective mode alignment.

**Theorem C.5.** *Assume that $p_{data} \sim \mathcal{N}(\boldsymbol{0}, k_*^2 \boldsymbol{I}_n)$ and that the discriminator is optimal, i.e., the discriminator consistently provides the precise moving direction for the particle. Then $k_t$, the steepness of $g$ at $\boldsymbol{x} = 0$ at discrete time step $t$ satisfies*

$$k_{t+1} = k_t + s\Big(\frac{1}{k_t^2} - \frac{1}{k_*^2}\Big) \cdot \frac{1}{1 + \frac{k_t\varphi(k_t\boldsymbol{x}_0/k_*)}{k_*\varphi(\boldsymbol{x}_0)}}, \tag{97}$$

*where $0 \leq t \leq T$, and $T$ is the maximum time. Here, $\varphi$ is the probability density function of $\mathcal{N}(\boldsymbol{0}, \boldsymbol{I}_n)$.*

*Proof.* Let $\varphi(\boldsymbol{x})$ be the probability density function of the $n$-dimensional standard Gaussian distribution

$$\varphi(\boldsymbol{x}) = \frac{1}{(2\pi)^{n/2}} \cdot \exp\Big(-\frac{1}{2}\boldsymbol{x}^\top\boldsymbol{x}\Big). \tag{98}$$

Then the probability density function of $\mathcal{N}(\boldsymbol{0}, k^2\boldsymbol{I}_n)$ is $\varphi(\boldsymbol{x}/k)/k$. Let $\boldsymbol{x}_t = k_t\boldsymbol{x}_0$ denotes the position of the particle at time $t$. Here, $k_t$ represents the steepness of the generator function at $\boldsymbol{x} = 0$. We investigate the evolution of the particle subject to the vector field given by $\nabla d(\boldsymbol{x})/d(\boldsymbol{x})$. Assuming the discriminator is optimal, this process is governed by the following explicit formula (Yi et al., 2023):

$$\boldsymbol{x}_{t+1} = \boldsymbol{x}_t + s \cdot \frac{\nabla r(\boldsymbol{x}_t)}{r(\boldsymbol{x}_t)(r(\boldsymbol{x}_t) + 1)}, \quad t = 1, 2, \ldots, T. \tag{99}$$

Here, $s$ denotes the step size, $T$ is the maximum time, and

$$r(\boldsymbol{x}) = \frac{\varphi(\boldsymbol{x}/k_*)/k_*}{\varphi(\boldsymbol{x}/k_t)/k_t} \tag{100}$$

---

[5] This result can be shown by taking the limit $t \to +\infty$ on both sides of the recurrence relation.

is the ratio of the probability density function of $p_{\text{data}}$ and $p_g$. By the formula of $\varphi(\boldsymbol{x})$, we deduce that $\nabla\varphi(\boldsymbol{x}) = -\varphi(\boldsymbol{x})\boldsymbol{x}$. Below we compute $\nabla r(\boldsymbol{x})$ by the chain rule:

$$
\begin{aligned}
\nabla r(\boldsymbol{x}) &= \frac{k_t}{k_*} \cdot \frac{\nabla\varphi(\boldsymbol{x}/k_*) \cdot \varphi(\boldsymbol{x}/k_t) - \varphi(\boldsymbol{x}/k_*)\nabla\varphi(\boldsymbol{x}/k_t)}{\varphi(\boldsymbol{x}/k_t)^2} \\
&= \frac{k_t}{k_*} \cdot \Big(\frac{1}{k_t^2} - \frac{1}{k_*^2}\Big)\frac{\varphi(\boldsymbol{x}/k_*)}{\varphi(\boldsymbol{x}/k_t)} \cdot \boldsymbol{x}.
\end{aligned}
\tag{101}
$$

Using $\boldsymbol{x}_t = k_t\boldsymbol{x}_0$, we derive the following recurrent formula for $\{k_t\}_{t=0}^T$:

$$
k_{t+1} = k_t + s\Big(\frac{1}{k_t^2} - \frac{1}{k_*^2}\Big) \cdot \frac{1}{1 + \frac{k_t\varphi(k_t\boldsymbol{x}_0/k_*)}{k_*\varphi(\boldsymbol{x}_0)}}.
\tag{102}
$$
$\square$

In the next theorem C.6, we analyze the setting where $p_{\text{data}}$ is a symmetric mixture of Gaussians and derive an evolution equation in continuous time for the steepness of the generator function $g_t$ at $x = 0$, i.e., $g_t'(0)$. This equation provides two key implications: (i) when the third-order derivative of $g_t$ at 0, i.e., $g_t^{(3)}(0)$, is small compared to $g_t'(0)$ (for example, when $g_t$ can be well-approximated by a linear function near $x = 0$, as fig. 3 depicts), the steepness $g_t'(0)$ monotonically increases; and (ii) at an early stage of training, the dominant terms in the numerator and denominator of the right-hand side of the evolution equation are $(\mu^2 - \sigma^2)/\sigma^4 \cdot (g_t'(0))^3$ and $(g_t'(0))^2$, respectively. As a result, $g_t'(0)$ initially grows exponentially with rate $(\mu^2 - \sigma^2)/\sigma^4$. This suggests that when the modes are well separated ($\mu \gg \sigma$), the generator rapidly increases its steepness to match the target distribution.

**Theorem C.6.** *Suppose $p_{data}$ is a symmetric mixture of Gaussians*

$$
p_{data} \sim 0.5\mathcal{N}(-\mu, \sigma^2) + 0.5\mathcal{N}(\mu, \sigma^2)
\tag{103}
$$

*where $\mu \gg \sigma$, i.e., the modes are well separated, and that the discriminator is optimal, i.e., the discriminator consistently provides the precise moving direction for the particle. Let $g_t\colon \mathbb{R} \to \mathbb{R}$ be a one-dimensional generator evolving in continuous time $t \geq 0$ according to the gradient-flow limit of the update*

$$
g_{t+\Delta t}(z) = g_t(z) + \Delta t \cdot \frac{d_t'(g_t(z))}{d_t(g_t(z))},
\tag{104}
$$

*as in algorithm 1, where*

$$
d_t(x) = \frac{p_{data}(x)}{p_{data}(x) + p_{g_t}(x)},
\tag{105}
$$

*and $p_{g_t}$ is the push-forward of the standard normal $\mathcal{N}(0,1)$ under $g_t$. Assume that $g_t$ is continuously differentiable for all $t \geq 0$ with $g_0$ being an odd function and $g_0'(0) > 0$. Then the steepness of $g$ at $z = 0$, namely $g_t'(0)$, satisfies the ODE*

$$
\frac{\mathrm{d}}{\mathrm{d}t}g_t'(0) = \frac{(\mu^2 - \sigma^2)/\sigma^4 \cdot (g_t'(0))^3 + g_t'(0) + g_t^{(3)}(0)}{\exp(-\mu^2/(2\sigma^2))/\sigma \cdot (g_t'(0))^3 + (g_t'(0))^2}.
\tag{106}
$$

*Proof.* We split the argument into the following steps.

**(i) Deriving the continuous-time limit and PDE.** As $\Delta t \to 0$, the discrete update

$$
g_{t+\Delta t}(z) = g_t(z) + \Delta t \cdot \frac{d_t'(g_t(z))}{d_t(g_t(z))}
\tag{107}
$$

yields, in the limit, the gradient-flow PDE

$$
\frac{\partial}{\partial t}g_t(z) = \frac{d_t'(g_t(z))}{d_t(g_t(z))},
\tag{108}
$$

where $d_t(x) = \dfrac{p_{\text{data}}(x)}{p_{\text{data}}(x) + p_{g_t}(x)}$ and $p_{g_t}$ is the push-forward of the standard normal under $g_t$.

**(ii) Proving $g_t$ remains an odd function.** By assumption, $g_0$ is an odd function. Because the update rule involves only derivatives and ratios that preserve the symmetry, it follows by a symmetry argument that $g_t$ remains odd for all $t$. In particular, this implies $g_t(0) = 0$ and $g_t''(0) = 0$.

**(iii) Differentiating with respect to $z$.** Differentiate both sides with respect to $z$. By the chain rule,

$$\frac{\partial}{\partial t} g_t'(z) = \frac{\partial}{\partial z}\left(\frac{d_t'(g_t(z))}{d_t(g_t(z))}\right) = \left(\frac{d_t''(g_t(z))}{d_t(g_t(z))} - \frac{(d_t'(g_t(z)))^2}{(d_t(g_t(z)))^2}\right) g_t'(z). \tag{109}$$

Evaluating at $z = 0$ gives

$$\frac{\mathrm{d}}{\mathrm{d}t} g_t'(0) = \left(\frac{d_t''(0)}{d_t(0)} - \frac{(d_t'(0))^2}{(d_t(0))^2}\right) g_t'(0). \tag{110}$$

**(iv) Computing $p_{g_t}(0)$, $p_{g_t}'(0)$, $p_{g_t}''(0)$ and likewise for $p_{\text{data}}$.** By the density transformation formula,

$$p_{g_t}(x) = \frac{p_z\big(g_t^{-1}(x)\big)}{\big|g_t'\big(g_t^{-1}(x)\big)\big|}, \quad \text{where } p_z = \mathcal{N}(0, 1). \tag{111}$$

Setting $x = 0$ and noting that $g_t^{-1}(0) = 0$ (since $g_t$ is odd) yields

$$p_{g_t}(0) = \frac{1}{\sqrt{2\pi} g_t'(0)}. \tag{112}$$

Differentiating again and using $p_z'(0) = 0$ along with $g_t''(0) = 0$, we obtain

$$p_{g_t}'(0) = 0,$$
$$p_{g_t}''(0) = -\frac{1}{\sqrt{2\pi}} \cdot \frac{g_t'(0) + g_t^{(3)}(0)}{\big(g_t'(0)\big)^4}. \tag{113}$$

For the mixture distribution $p_{\text{data}}(x)$, we have

$$p_{\text{data}}(0) = \frac{1}{\sqrt{2\pi\sigma^2}} \cdot \exp\left(-\frac{\mu^2}{2\sigma^2}\right),$$
$$p_{\text{data}}'(0) = 0, \tag{114}$$
$$p_{\text{data}}''(0) = \frac{\mu^2 - \sigma^2}{\sigma^4 \sqrt{2\pi\sigma^2}} \cdot \exp\left(-\frac{\mu^2}{2\sigma^2}\right).$$

Then, by definition,

$$d_t'(0) = \left.\frac{p_{\text{data}}'(x) p_{g_t}(x) - p_{\text{data}}(x) p_{g_t}'(x)}{\big(p_{\text{data}}(x) + p_{g_t}(x)\big)^2}\right|_{x=0} = 0, \tag{115}$$
$$d_t''(0) = \frac{p_{\text{data}}''(0) p_{g_t}(0) - p_{\text{data}}(0) p_{g_t}''(0)}{\big(p_{\text{data}}(0) + p_{g_t}(0)\big)^2}. \tag{116}$$

**(v) Assembling the ODE for $g_t'(0)$.** Substitute the above expressions into

$$\frac{\mathrm{d}}{\mathrm{d}t} g_t'(0) = \left(\frac{d_t''(0)}{d_t(0)} - \frac{(d_t'(0))^2}{(d_t(0))^2}\right) g_t'(0) \tag{117}$$

to get

$$\frac{\mathrm{d}}{\mathrm{d}t} g_t'(0) = \frac{p_{\text{data}}''(0) p_{g_t}(0) - p_{\text{data}}(0) p_{g_t}''(0)}{\big(p_{\text{data}}(0) + p_{g_t}(0)\big) \cdot p_{\text{data}}(0)} g_t'(0). \tag{118}$$

Plugging in the explicit forms, we obtain

$$\frac{\mathrm{d}}{\mathrm{d}t} g_t'(0) = \frac{(\mu^2 - \sigma^2)/\sigma^4 \cdot (g_t'(0))^3 + g_t'(0) + g_t^{(3)}(0)}{\exp(-\mu^2/\sigma^2)/\sigma \cdot (g_t'(0))^3 + (g_t'(0))^2}. \tag{119}$$

This completes the proof. $\square$

## C.8 Quantitative Results on How Steepness Impacts the Severity of Mode Mixture

**Theorem C.7.** *Assume that the real data distribution $p_{data}(x)$ satisfies assumption 2.1 with $n = 1$ and separation condition $\Delta = 6\sigma$. Furthermore, assume that the generator function $g$ is increasing and satisfies $\sup_{x \in \mathbb{R}} \mathcal{S}_g(x) \leq k$. Additionally, assume that*

$$g^{-1}\left(\frac{x_i + x_{i+1}}{2}\right) = \Phi^{-1}\left(\Psi\left(\frac{x_i + x_{i+1}}{2}\right)\right), \tag{120}$$

*where $\Phi(x)$ denotes the cumulative distribution function (CDF) of the standard normal distribution $\mathcal{N}(0, 1)$, and $\Psi(x)$ is the CDF of the distribution $p_{data}(x)$. Then, the probability that the particles fall into the interval*

$$\bigcup_{i=1}^{N-1} [x_i + 3\sigma, x_{i+1} - 3\sigma], \tag{121}$$

*which indicates mode mixture, is at least*

$$\sum_{i=1}^{N-1} \left(\Phi\left(\Phi^{-1}\left(\Psi\left(\frac{x_i + x_{i+1}}{2}\right)\right) + \frac{x_{i+1} - x_i - 3\sigma}{2k}\right) - \Phi\left(\Phi^{-1}\left(\Psi\left(\frac{x_i + x_{i+1}}{2}\right)\right) - \frac{x_{i+1} - x_i - 3\sigma}{2k}\right)\right). \tag{122}$$

*Proof.* Given $x \sim \mathcal{N}(0, 1)$, we need to calculate the probability that

$$x \in \bigcup_{i=1}^{N-1} [g^{-1}(x_i + 3\sigma), g^{-1}(x_{i+1} - 3\sigma)]. \tag{123}$$

Since $g^{-1}\big((x_i + x_{i+1})/2\big)$ is identical to its optimal counterpart, it suffices to analyze how $g^{-1}(x_i + 3\sigma)$ and $g^{-1}(x_{i+1} - 3\sigma)$ deviate from this value. In other words, we only need to compute the maximum value of $g^{-1}(x_i + 3\sigma)$ and the minimum value of $g^{-1}(x_{i+1} - 3\sigma)$, as the probability that a standard Gaussian variable falls within an interval decreases with respect to its left endpoint and increases with respect to its right endpoint. Using the property that $\sup_{x \in \mathbb{R}} \mathcal{S}_g(x) \leq k$, we have:

$$g^{-1}(x_i + 3\sigma) \leq g^{-1}\left(\frac{x_i + x_{i+1}}{2}\right) - \frac{x_{i+1} - x_i - 3\sigma}{2k}, \tag{124}$$

and

$$g^{-1}(x_{i+1} - 3\sigma) \geq g^{-1}\left(\frac{x_i + x_{i+1}}{2}\right) + \frac{x_{i+1} - x_i - 3\sigma}{2k}. \tag{125}$$

By summing over all intervals, we derive that the probability that particles fall into

$$\bigcup_{i=1}^{N-1} [x_i + 3\sigma, x_{i+1} - 3\sigma] \tag{126}$$

is at least

$$\sum_{i=1}^{N-1} \left(\Phi\left(g^{-1}\left(\frac{x_i + x_{i+1}}{2}\right) + \frac{x_{i+1} - x_i - 3\sigma}{2k}\right) - \Phi\left(g^{-1}\left(\frac{x_i + x_{i+1}}{2}\right) - \frac{x_{i+1} - x_i - 3\sigma}{2k}\right)\right)$$
$$= \sum_{i=1}^{N-1} \left(\Phi\left(\Phi^{-1}\left(\Psi\left(\frac{x_i + x_{i+1}}{2}\right)\right) + \frac{x_{i+1} - x_i - 3\sigma}{2k}\right) - \Phi\left(\Phi^{-1}\left(\Psi\left(\frac{x_i + x_{i+1}}{2}\right)\right) - \frac{x_{i+1} - x_i - 3\sigma}{2k}\right)\right). \tag{127}$$

Note that for the case that $N = 2$ and $-x_1 = x_2 = x$, this probability simplifies to

$$\Phi\left(\frac{2x - 3\sigma}{2k}\right) - \Phi\left(-\frac{2x - 3\sigma}{2k}\right). \tag{128}$$

$\square$

### C.9 A Local Analysis of Steepness at Collapse

**Theorem C.8.** *Following the notations in algorithm 1, assume that after the update step, the generator is optimal in the sense that $g_{\theta'}(z_i) = \hat{Z}_i$. Further assume there are infinitely many particles and that the step size $s > 0$ is sufficiently small. Then, the Jacobian $J_{g_{\theta'}}(z)$ of the updated generator $g_{\theta'}$ satisfies*

$$J_{g_{\theta'}}(z) = J_{g_\theta}(z) + s \cdot \nabla_x\Big(\frac{\nabla d_\omega}{d_\omega}\Big)\big(g_\theta(z)\big) \cdot J_{g_\theta}(z), \tag{129}$$

*where $\nabla_x(\nabla d_\omega/d_\omega)(x)$ is the Jacobian of the vector field $\nabla d_\omega/d_\omega$ evaluated at $x$.*

*Proof.* By the algorithm, particles are updated as

$$\hat{Z} = Z + s \cdot \frac{\nabla d_\omega(Z)}{d_\omega(Z)}, \quad \text{where } Z_i = g_\theta(z). \tag{130}$$

Assuming the generator is optimal after the update, the new generator satisfies

$$g_{\theta'}(z) = \hat{Z} = g_\theta(z) + s \cdot \frac{\nabla d_\omega(g_\theta(z))}{d_\omega(g_\theta(z))}. \tag{131}$$

Differentiating both sides with respect to $z$, we obtain

$$J_{g_{\theta'}}(z) = J_{g_\theta}(z) + s \cdot \nabla_z\Big(\frac{\nabla d_\omega(g_\theta(z))}{d_\omega(g_\theta(z))}\Big). \tag{132}$$

Applying the chain rule to compute the Jacobian of the velocity field:

$$\nabla_z\Big(\frac{\nabla d_\omega(g_\theta(z))}{d_\omega(g_\theta(z))}\Big) = \nabla_x\Big(\frac{\nabla d_\omega}{d_\omega}\Big)\big(g_\theta(z)\big) \cdot J_{g_\theta}(z), \tag{133}$$

where $\nabla_x(\nabla d_\omega/d_\omega)(x)$ is the Jacobian of the vector field evaluated at $x$. $\qquad\square$

**Theorem C.9.** *Assume that the real data distribution $p_{data}(x)$ satisfies assumption 2.1 with separation condition $\Delta = 8\sigma$. Suppose the current generator is $g_\theta$, and the current discriminator $d(x)$ satisfies the linear model assumption 4.1 near a certain mode $x_i$, specifically $d(x) = 1/2 - \|x - x_i\|_2/(8\sigma)$ for all $x \in B_{4\sigma}(x_i)$. Under the same conditions as in theorem 4.1, the steepness of the updated generator $g_{\theta'}$ satisfies*

$$\mathcal{S}_{g_{\theta'}}(z) \le \Big(1 - \frac{s}{(4\sigma - r)^2}\Big) \cdot \mathcal{S}_{g_\theta}(z), \tag{134}$$

*for all latent vectors $z$ such that $g_\theta(z) \in B_{2\sigma}(x_i)$, where $r = \|g_\theta(z) - x_i\|_2$, provided that the step size $s < (4\sigma - r)^2$ is sufficiently small.*

*Proof.* Given the discriminator's linear behavior near the mode $x_i$, expressed as

$$d(x) = \frac{1}{2} - \frac{1}{8\sigma} \cdot \|x - x_i\|_2, \tag{135}$$

for $x \in B_{2\sigma}(x_i)$, the gradient and Hessian (where the Hessian specifically refers to $\nabla^2 \log d(x)$) of $d(x)$ are:

$$\nabla d(x) = -\frac{y}{8\sigma r}, \quad \nabla_x\Big(\frac{\nabla d(x)}{d(x)}\Big) = -\frac{I}{r(4\sigma - r)} + \frac{yy^\top}{r^3(4\sigma - r)} - \frac{yy^\top}{r^2(4\sigma - r)^2}, \tag{136}$$

where $r = \|x - x_i\|_2$ and $y = x - x_i$. Denoting the Hessian by $H$, the updated generator's Jacobian satisfies

$$J_{g_{\theta'}}(z) = (I + sH) \cdot J_{g_\theta}(z). \tag{137}$$

By the submultiplicative property of the spectral norm, we have

$$\mathcal{S}_{g_{\theta'}}(z) = \|J_{g_{\theta'}}(z)\|_2 \le \|I + sH\|_2 \cdot \|J_{g_\theta}(z)\|_2 = \|I + sH\|_2 \cdot \mathcal{S}_{g_\theta}(z). \tag{138}$$

Next, we analyze $\|\boldsymbol{I} + s\boldsymbol{H}\|_2$. Since $\boldsymbol{y}\boldsymbol{y}^\top$ is rank-1 with the only nonzero eigenvalue $r^2$, the eigenvalues of $\boldsymbol{H}$ are

$$-\frac{1}{r(4\sigma - r)} \quad (\text{multiplicity } n-1), \quad -\frac{1}{r(4\sigma - r)} + \frac{4\sigma - 2r}{r(4\sigma - r)^2} \quad (\text{multiplicity } 1). \tag{139}$$

The eigenvalues of $\boldsymbol{I} + s\boldsymbol{H}$ are therefore

$$1 - \frac{s}{r(4\sigma - r)} \quad (\text{multiplicity } n-1), \quad 1 - \frac{s}{r(4\sigma - r)} + \frac{s(4\sigma - 2r)}{r(4\sigma - r)^2} \quad (\text{multiplicity } 1). \tag{140}$$

Since $\boldsymbol{I} + s\boldsymbol{H}$ is a symmetric matrix, the spectral norm of $\boldsymbol{I} + s\boldsymbol{H}$ is the largest eigenvalue

$$\|\boldsymbol{I} + s\boldsymbol{H}\|_2 = 1 - \frac{s}{r(4\sigma - r)} + \frac{s(4\sigma - 2r)}{r(4\sigma - r)^2} = 1 - \frac{s}{(4\sigma - r)^2}, \tag{141}$$

where we implicitly use the condition that the step size $s < (4\sigma - r)^2$, so that $\boldsymbol{I} + s\boldsymbol{H}$ is positive definite. Substituting back, we obtain

$$\mathcal{S}_{g_{\theta'}}(\boldsymbol{z}) \le \left(1 - \frac{s}{(4\sigma - r)^2}\right) \cdot \mathcal{S}_{g_\theta}(\boldsymbol{z}), \tag{142}$$

as required. $\qquad\square$

# D   Analysis of a Class of Suboptimal Discriminators

## D.1   The Class of Suboptimal Discriminators

In this section, we analyze a class of suboptimal discriminators that can be expressed as

$$\hat{d}_\omega(\boldsymbol{x}) = \frac{p_{\text{data}}(\boldsymbol{x})}{p_{\text{data}}(\boldsymbol{x}) + f(r(\boldsymbol{x})) \cdot p_g(\boldsymbol{x})}, \tag{143}$$

where $r(\boldsymbol{x}) = p_{\text{data}}(\boldsymbol{x})/p_g(\boldsymbol{x})$ represents the density ratio, and $f$ is a scalar function. The optimal discriminator, as established by Goodfellow et al. (2014), corresponds to the case where $f \equiv 1$. This formulation naturally arises from the training process of the discriminator, which effectively functions as a binary classifier distinguishing between real and generated data. During training, gradient descent approximates the density ratio $r(\boldsymbol{x})$, and deviations from the true value are captured by the function $f(r(\boldsymbol{x}))$.

The term $f(r(\boldsymbol{x}))$ quantifies the error in the estimation of the density ratio. Specifically, the suboptimal discriminator can be rewritten as:

$$\hat{d}_\omega(\boldsymbol{x}) = \frac{1}{1 + f(r(\boldsymbol{x})) \cdot \left(p_{\text{data}}(\boldsymbol{x})/p_g(\boldsymbol{x})\right)^{-1}} = \frac{1}{1 + \left(r(\boldsymbol{x})/f(r(\boldsymbol{x}))\right)^{-1}}. \tag{144}$$

This highlights the role of $f(r(\boldsymbol{x}))$ as a measure of deviation from the optimal case. When $f(r(\boldsymbol{x})) = 1$, the discriminator achieves optimality, perfectly distinguishing between real and generated samples. However, deviations from $f(r(\boldsymbol{x})) = 1$ reflect imperfections in the discriminator, introducing bias or error into the classification process. Such a modeling allows us to analyze and understand the behavior of suboptimal discriminators and their impact on the overall performance of GANs.

## D.2   The Influence of the Suboptimal Discriminator to the Vector Field

In this subsection, we investigate the influence of the suboptimal discriminator on the vector field that governs the movement of particles. This analysis complements the discussion in section 3.

**Proposition D.1.** *Assume that $f \in C^2(0, +\infty)$. Then, at a point $\boldsymbol{x}$ where $p_{data}(\boldsymbol{x})p_g(\boldsymbol{x}) > 0$, the cosine of the angle $\theta$ between the suboptimal vector $\nabla\hat{d}_\omega(\boldsymbol{x})/(2\hat{d}_\omega(\boldsymbol{x}))$ and the optimal vector $\nabla d_*(\boldsymbol{x})/(2d_*(\boldsymbol{x}))$ is given by*

$$\cos\theta = \frac{\langle \nabla\hat{d}_\omega(\boldsymbol{x}), \nabla d_*(\boldsymbol{x})\rangle}{\left\|\nabla\hat{d}_\omega(\boldsymbol{x})\right\|_2 \left\|\nabla d_*(\boldsymbol{x})\right\|_2} = \text{sign}\left(\frac{f(r(\boldsymbol{x}))}{r(\boldsymbol{x})} - f'(r(\boldsymbol{x}))\right). \tag{145}$$

*Consequently, there exists $\delta > 0$ that depends on $f$ such that whenever $r(\boldsymbol{x}) < \delta$, the two vectors are in the same direction.*

*Proof.* To calculate the angle between two vectors, we can ignore their scalar coefficients. Therefore, we only need to determine the angle between $\nabla \hat{d}_\omega(\boldsymbol{x})$ and $\nabla d_*(\boldsymbol{x})$. Using the results derived in theorem C.1, this calculation reduces to finding the angle between

$$-p_{\text{data}}(\boldsymbol{x})\nabla p_g(\boldsymbol{x}) + p_g(\boldsymbol{x})\nabla p_{\text{data}}(\boldsymbol{x}) \tag{146}$$

and

$$
\begin{aligned}
&- p_{\text{data}}(\boldsymbol{x})\nabla\big(\alpha(\boldsymbol{x}) \cdot p_g(\boldsymbol{x})\big) + \big(\alpha(\boldsymbol{x}) \cdot p_g(\boldsymbol{x})\big)\nabla p_{\text{data}}(\boldsymbol{x}) \\
&= \alpha(\boldsymbol{x})\big( - p_{\text{data}}(\boldsymbol{x})\nabla p_g(\boldsymbol{x}) + p_g(\boldsymbol{x})\nabla p_{\text{data}}(\boldsymbol{x})\big) - p_{\text{data}}(\boldsymbol{x})p_g(\boldsymbol{x})\nabla\alpha(\boldsymbol{x}),
\end{aligned}
\tag{147}
$$

where $\alpha(\boldsymbol{x}) = f(r(\boldsymbol{x}))$. We use the same technique and divide both vectors by the scalar $p_{\text{data}}(\boldsymbol{x})p_g(\boldsymbol{x})\alpha(\boldsymbol{x})$. By applying the chain rule, we only need to compute the angle between

$$-\nabla \log p_g(\boldsymbol{x}) + \nabla \log p_{\text{data}}(\boldsymbol{x}) = \nabla \log r(\boldsymbol{x}) \tag{148}$$

and

$$\nabla \log r(\boldsymbol{x}) - \nabla \log \alpha(\boldsymbol{x}). \tag{149}$$

We proceed with the final calculations:

$$\cos\theta = \frac{\langle \nabla \log r(\boldsymbol{x}), \nabla \log(r(\boldsymbol{x})/f(r(\boldsymbol{x})))\rangle}{\|\nabla \log r(\boldsymbol{x})\|_2 \|\nabla \log(r(\boldsymbol{x})/f(r(\boldsymbol{x})))\|_2}. \tag{150}$$

For the numerator, we have $\nabla \log r(\boldsymbol{x}) = \nabla r(\boldsymbol{x})/r(\boldsymbol{x})$, and

$$\nabla \log f(r(\boldsymbol{x})) = \frac{f'(r(\boldsymbol{x}))}{f(r(\boldsymbol{x}))} \cdot \nabla r(\boldsymbol{x}), \tag{151}$$

implying that $\nabla \log r(\boldsymbol{x})$ and $\nabla \log(r(\boldsymbol{x})/f(r(\boldsymbol{x})))$ are both parallel to $\nabla r(\boldsymbol{x})$. Therefore,

$$
\begin{aligned}
\cos\theta &= \text{sign}\Big(\frac{1}{r(\boldsymbol{x})} - \frac{f'(r(\boldsymbol{x}))}{f(r(\boldsymbol{x}))}\Big) \\
&= \text{sign}\Big(\frac{f(r(\boldsymbol{x}))}{r(\boldsymbol{x})} - f'(r(\boldsymbol{x}))\Big).
\end{aligned}
\tag{152}
$$

By the continuity of $f''$, there exists $\varepsilon > 0$ such that for $x \in [0, \varepsilon)$, we have $|f''(x)| < M$. As a result, for $\boldsymbol{x}$ such that $r(\boldsymbol{x}) < \delta := \min\big(\varepsilon, \sqrt{2f(0)/M}\big)$, we have

$$\cos\theta = \text{sign}\big(f(r(\boldsymbol{x})) - r(\boldsymbol{x})f'(r(\boldsymbol{x}))\big) = \text{sign}\big(f(0) + r(\boldsymbol{x})^2 f''(\xi)/2\big) = 1 \tag{153}$$

for some $\xi \in (0, r(\boldsymbol{x}))$, where we use Taylor's expansion with the Lagrange remainder. $\square$

We now briefly discuss the implications of proposition D.1. Firstly, this proposition considers $f \equiv 1$ as a special case, in which $\cos\theta = 1$ for any choice of $\boldsymbol{x}$. Secondly, although the proposition seems to hold only for $\boldsymbol{x}$ where $r(\boldsymbol{x})$ is small, this is sufficient for our purposes. In this subsection, we are focusing on the fitting phase, where $r(\boldsymbol{x})$ is typically small for $\boldsymbol{x} \sim p_g(\boldsymbol{x})$. Finally, it may seem counter-intuitive that the vector field of the suboptimal discriminator aligns perfectly with that of the optimal discriminator. However, it is important to note that while the directions of these two vector fields may be the same, their magnitudes can differ. We choose not to delve further into this topic because the magnitudes can be adjusted by varying the step sizes.

### D.3 The Influence of the Suboptimal Discriminator to the Evolution of Steepness

In this subsection, we investigate the influence of the suboptimal discriminator on the evolution of steepness. This analysis complements the discussion in section 3.2.

**Proposition D.2.** *Assume that $p_{data} \sim \mathcal{N}(\mathbf{0}, k_*^2 \mathbf{I}_n)$ and that the discriminator is suboptimal and takes the form*

$$\hat{d}_\omega(\boldsymbol{x}) = \frac{p_{data}(\boldsymbol{x})}{p_{data}(\boldsymbol{x}) + f(r(\boldsymbol{x})) \cdot p_g(\boldsymbol{x})}, \tag{154}$$

*where $r(\boldsymbol{x}) = p_{data}(\boldsymbol{x})/p_g(\boldsymbol{x})$, and $f$ is a function measuring the deviation of $\hat{d}_\omega(\boldsymbol{x})$ from the optimal discriminator. Then $k_t$, the steepness of $g$ at $\boldsymbol{x} = 0$ at discrete time step $t$ satisfies*

$$k_{t+1} = k_t + s\left(\frac{1}{k_t^2} - \frac{1}{k_*^2}\right) \cdot \frac{f(r(k_t\boldsymbol{x}_0)) - r(k_t\boldsymbol{x}_0)f'(r(k_t\boldsymbol{x}_0))}{r(k_t\boldsymbol{x}_0) + f(r(k_t\boldsymbol{x}_0))}, \tag{155}$$

*where $0 \leq t \leq T$, and $T$ is the maximum time. Here, $\varphi$ is the probability density function of $\mathcal{N}(\mathbf{0}, \mathbf{I}_n)$ and*

$$r(k_t\boldsymbol{x}_0) = \frac{k_t\varphi(k_t\boldsymbol{x}_0/k_*)}{k_*\varphi(\boldsymbol{x}_0)}. \tag{156}$$

*Proof.* Let $\varphi(\boldsymbol{x})$ be the probability density function of the $n$-dimensional standard Gaussian distribution

$$\varphi(\boldsymbol{x}) = \frac{1}{(2\pi)^{n/2}} \cdot \exp\left(-\frac{1}{2}\boldsymbol{x}^\top\boldsymbol{x}\right). \tag{157}$$

Then the probability density function of $\mathcal{N}(\mathbf{0}, k^2\mathbf{I}_n)$ is $\varphi(\boldsymbol{x}/k)/k$. Let $\boldsymbol{x}_t = k_t\boldsymbol{x}_0$ denotes the position of the particle at time $t$. Here, $k_t$ represents the steepness of the generator function. We investigate the evolution of the particle subject to the vector field given by $\nabla\hat{d}_\omega(\boldsymbol{x})/\hat{d}_\omega(\boldsymbol{x})$, which can be written in terms of $r(\boldsymbol{x})$ as

$$\boldsymbol{x}_{t+1} = \boldsymbol{x}_t + s \cdot \frac{\big(f(r(\boldsymbol{x}_t)) - r(\boldsymbol{x}_t)f'(r(\boldsymbol{x}_t))\big)\nabla r(\boldsymbol{x}_t)}{r(\boldsymbol{x}_t)\big(r(\boldsymbol{x}_t) + f(r(\boldsymbol{x}_t))\big)}, \quad t = 1, 2, \ldots, T. \tag{158}$$

By the formula of $\varphi(\boldsymbol{x})$, we deduce that $\nabla\varphi(\boldsymbol{x}) = -\varphi(\boldsymbol{x})\boldsymbol{x}$. Below we compute $\nabla r(\boldsymbol{x})$ by the chain rule:

$$\begin{aligned}
\nabla r(\boldsymbol{x}) &= \frac{k_t}{k_*} \cdot \frac{\nabla\varphi(\boldsymbol{x}/k_*) \cdot \varphi(\boldsymbol{x}/k_t) - \varphi(\boldsymbol{x}/k_*)\nabla\varphi(\boldsymbol{x}/k_t)}{\varphi(\boldsymbol{x}/k_t)^2} \\
&= \frac{k_t}{k_*} \cdot \left(\frac{1}{k_t^2} - \frac{1}{k_*^2}\right) \cdot \frac{\varphi(\boldsymbol{x}/k_*)}{\varphi(\boldsymbol{x}/k_t)} \cdot \boldsymbol{x}.
\end{aligned} \tag{159}$$

Using $\boldsymbol{x}_t = k_t\boldsymbol{x}_0$, we derive the following recurrent formula for $\{k_t\}_{t=0}^T$:

$$k_{t+1} = k_t + s\left(\frac{1}{k_t^2} - \frac{1}{k_*^2}\right) \cdot \frac{f(r(k_t\boldsymbol{x}_0)) - r(k_t\boldsymbol{x}_0)f'(r(k_t\boldsymbol{x}_0))}{r(k_t\boldsymbol{x}_0) + f(r(k_t\boldsymbol{x}_0))}, \tag{160}$$

where

$$r(k_t\boldsymbol{x}_0) = \frac{k_t\varphi(k_t\boldsymbol{x}_0/k_*)}{k_*\varphi(\boldsymbol{x}_0)}. \tag{161}$$

$\square$

Note that this proposition considers $f \equiv 1$ as a special case, leading to the same conclusion as in theorem 3.3.

# E  Disparity Among Modes Across Different Datasets

## E.1  MNIST

**Preprocessing.** We first transform the images in MNIST by sequentially resizing the images to $64 \times 64$ pixels, converting them to PyTorch tensors, and normalizing the tensor values to the range of $[-1, 1]$.

**Computation.** We calculate the average image tensor for each label based on a set of 10 image tensors sharing the same label. Next, we compute the pairwise distances between these average tensors using the Frobenius norm. The resulting distances are visualized as a heatmap in fig. 8.

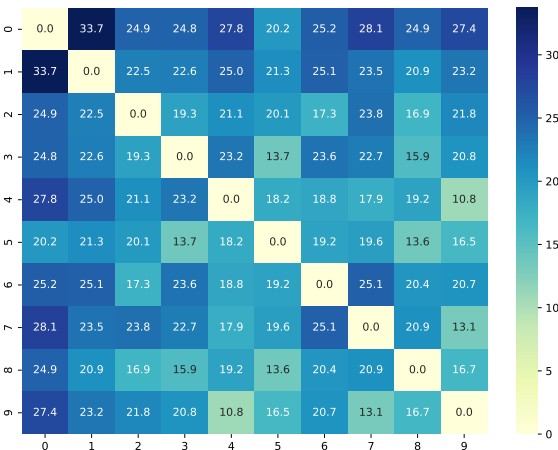

Figure 8: Frobenius distances between different modes in MNIST. The tensor of the modes are approximated by taking the average of image tensors that share the same label.

## E.2 Fashion MNIST

**Preprocessing.** We first transform the images in Fashion MNIST by first resizing the images to $64 \times 64$ pixels, converting them to PyTorch tensors, and normalizing the tensor values to the range of $[-1, 1]$.

**Computation.** We calculate the average image tensor for each label based on a set of 10 image tensors sharing the same label. Next, we compute the pairwise distances between these average tensors using the Frobenius norm. The resulting distances are visualized as a heatmap in fig. 9.

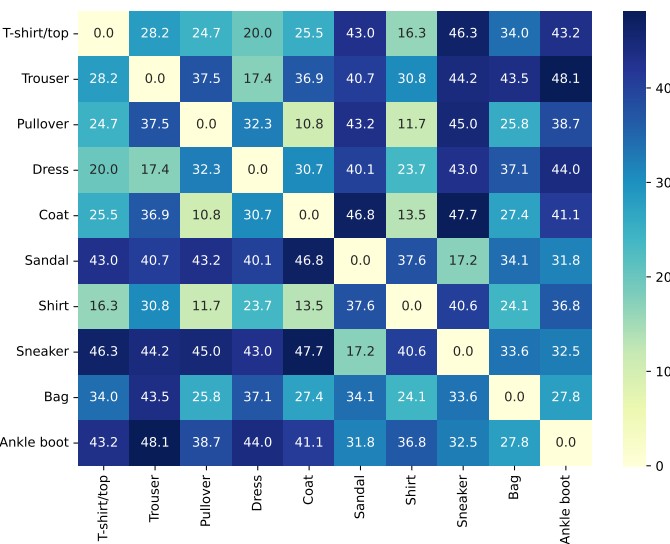

Figure 9: Frobenius distances between different modes in Fashion MNIST. The tensor of the modes are approximated by taking the average of image tensors that share the same label.

## E.3 CIFAR-10

**Preprocessing.** We first transform the images in CIFAR-10 by sequentially resizing the images to $64 \times 64$ pixels, converting them to PyTorch tensors, and normalizing the tensor values to the range of $[-1, 1]$.

**Computation.** We calculate the average image tensor for each label based on a set of 10 image tensors sharing the same label. Next, we compute the pairwise distances between these average tensors using the Frobenius norm. The resulting distances are visualized as a heatmap in fig. 10.

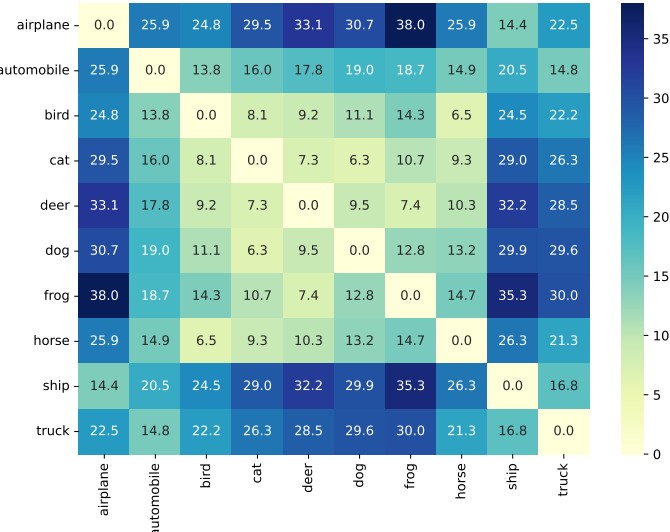

Figure 10: Frobenius distances between different modes in CIAFR-10. The tensor of the modes are approximated by taking the average of image tensors that share the same label.

## F  Detailed Experimental Settings

All codes are provided in the supplementary material.

### F.1  Verifying Fitting

**Methodology.** We demonstrate that the fitting phase exist in real-world datasets. To do this, we use a classification network $q(\boldsymbol{x})$ that takes an image tensor $\boldsymbol{x}$ as an input and outputs a 10-dimensional vector,

$$(p_0, p_1, \ldots, p_9), \tag{162}$$

where each $p_i \in [0, 1]$ denotes the likelihood of $\boldsymbol{x}$ corresponding to the $i$th category (e.g., the 1st category in MNIST corresponds to the handwritten digit 1 and the 2nd category in Fashion MNIST represents pullovers). Our focus gravitates towards those $p_i$'s that exhibit significant magnitudes. For discernibility, a threshold $\tau$ is set to $10^{-2}$. In other words, if $p_i > 10^{-2}$, then there is a notable probability that $\boldsymbol{x}$ belongs to the $i$th category. Empirical observations suggest that seldom do more than three $p_i$'s surpass the designated threshold. Hence, for any $\boldsymbol{x}$, we may pair $(i, j)$ when both $p_i$ and $p_j$ exceed $\tau$. By pairing, the intuition is that such $\boldsymbol{x}$ potentially resides *between* modes $i$ and $j$. In scenarios where only a single $p_i$ surpasses $\tau$, $i$ is paired with itself, implying that the $\boldsymbol{x}$ predominantly belongs to the $i$th category. We count the occurrences of the pairings $(i, j)$ $(0 \leq i, j \leq 9)$ in a batch of size 256 and visualize them with heatmaps in fig. 4, fig. 14 and fig. 15. In these figures, the value of the entry $(i, j)$ represents the logarithmically transformed occurrence frequency of pair $(i, j)$ within a batch, adjusted by one, thereby mitigating the impact of dominant diagonal values on the colorbar.

**Classification networks.** We use the MNIST classification network in MNIST classification network and the Fashion MNIST classification network in Fashion MNIST classification network.

**Number of training runs.** We conducted our experiments at least 50 times and consistently observed similar patterns across all trials. Therefore, we randomly selected two of these experiments to present in this paper.

## F.2 Early Stopping

**Early stopping on 3-dimensional Gaussian mixture.** In this part, our codes borrow heavily from NSGAN. Both the generator and the discriminator are implemented as full-connected neural networks with SGD optimizers. Now we elaborate on how to calculate the thresholds defined in algorithm 2. The discriminator threshold is given by $k_d/(2\sigma)$. We set $k_d = 2$, the distance between two nearest modes in the 3-dimensional Gaussian mixture dataset. For $\sigma$, it equals $\sqrt{0.0125}$ in our setting. Therefore the threshold is

$$k_d/(2\sigma) = 2/(2 \times \sqrt{0.0125}) \approx 8.9. \tag{163}$$

We set the generator threshold $k_g = -0.5$. As for the warm-up training iteration parameter $N_w$, we set it to 50.

**Early stopping on MNIST.** In this part, our generator and discriminator architectures borrow heavily from NSGAN on MNIST. Both the generator and the discriminator are implemented as convolutional neural networks with Adam optimizers. Now we elaborate on how to calculate the threshold defined in algorithm 2. The discriminator threshold is given by $k_d/(2\sigma)$. We set $k_d = 33.7$, the distance between two farthest modes in MNIST (please refer to appendix E). For $\sigma$, we first compute the population variance of the images from each label, arriving at 10 values. Then we compute their average value, and divide this value by $64 \times 64 \times 1$, i.e., the total number of dimensions. Therefore the threshold is

$$k_d/(2\sigma) = 33.7/(2 \times \sqrt{0.33/64^2}) \approx 1877. \tag{164}$$

We set the generator threshold $k_g = -0.5$. As for the warm-up training iteration parameter $N_w$, we set it to 20.

**Early stopping on Fashion MNIST.** In this part, our generator and discriminator architectures borrow heavily from NSGAN on Fashion MNIST. Both the generator and the discriminator are implemented as convolutional neural networks with Adam optimizers. Now we elaborate on how to calculate the threshold defined in algorithm 2. The discriminator threshold is given by $k_d/(2\sigma)$. We set $k_d = 48.1$, the distance between two farthest modes in Fashion MNIST (please refer to appendix E). For $\sigma$, we first compute the population variance of the images from each label, arriving at 10 values. Then we compute their average value, and divide this value by $64 \times 64 \times 1$, i.e., the total number of dimensions. Therefore the threshold is

$$k_d/(2\sigma) = 48.1/(2 \times \sqrt{0.33/64^2}) \approx 2679. \tag{165}$$

We set the generator threshold $k_g = -0.5$. As for the warm-up training iteration parameter $N_w$, we set it to 50.

**Early stopping on CIFAR-10.** In this part, our generator and discriminator architectures borrow heavily from NSGAN on CIFAR-10. Both the generator and the discriminator are implemented as convolutional neural networks with Adam optimizers. Now we elaborate on how to calculate the threshold defined in algorithm 2. The discriminator threshold is given by $k_d/(2\sigma)$. We set $k_d = 38.0$, the distance between two farthest modes in CIFAR-10 (please refer to appendix E). For $\sigma$, we first compute the population variance of the images from each label, arriving at 10 values. Then we compute their average value, and divide this value by $64 \times 64 \times 3$, i.e., the total number of dimensions. Therefore the threshold is

$$k_d/(2\sigma) = 38.0/(2 \times \sqrt{0.23/(64^2 \times 3)}) \approx 4391. \tag{166}$$

We set the generator threshold $k_g = -0.5$. As for the warm-up training iteration parameter $N_w$, we set it to 50.

**Number of training runs.** On all of the datasets mentioned above, we conducted our experiments at least 100 times. We observed similar patterns across all trials, although the point at which the GANs collapsed varied. Therefore, we choose to present those that collapsed before a certain threshold to ensure consistency in our reported results. It is important to note that the generated samples eventually collapsed in our experiments, either sooner or later, without contradicting the findings in our paper.

# G   Additional Experimental Results

## G.1   The Behavior of the Discriminator at the Collapse Phase

In this subsection, we elaborate on the optimal discriminator's behavior outlined in section 4.1.

**Verification on a toy example.** We first consider the following synthetic dataset

$$p_{\text{data}} \sim \frac{1}{4}\mathcal{N}([1,1], 0.0125\boldsymbol{I}_2) + \frac{1}{4}\mathcal{N}([1,-1], 0.0125\boldsymbol{I}_2) + \frac{1}{4}\mathcal{N}([-1,1], 0.0125\boldsymbol{I}_2) + \frac{1}{4}\mathcal{N}([-1,-1], 0.0125\boldsymbol{I}_2),$$
(167)

and train the discriminator until optimal. We plot the values of the optimal discriminator in fig. 11. We observe that discriminator values are close to 0.5 (red) in the central regions of the modes and approach zero (dark blue) in regions far from the modes. The contours between these regions are approximately equally spaced, indicating a linear decline in discriminator values. The distance from the mode centers to where discriminator values vanish is approximately 0.4, aligning with the predicted $4\sigma = 4 \cdot \sqrt{0.0125} \approx 0.45$ in assumption 4.1.

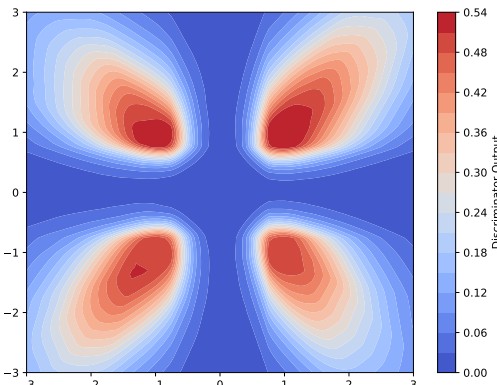

Figure 11: The values of the optimal discriminator. The discriminator values are close to 0.5 in the central regions of the modes (i.e., $[\pm1, \pm1]$) and vanish in the regions far from the modes. Between them, the discriminator values smoothly change from 0.5 to 0.

**Verification in real datasets.** We justify assumption 4.1 on MNIST and Fashion MNIST. We use saved checkpoints prior to mode collapse as the basis for our analysis, which are the 43rd epoch and the 124th epoch, respectively, for MNIST and Fashion MNIST, under the same setting as in figs. 5, 6 and 17 to 19. Specifically, we randomly sampled 10k points from MNIST and Fashion MNIST, and applied the $k$-means algorithm to cluster these points into 100 clusters, treating the cluster centers as representative modes of the data. We then computed the discriminator values at these cluster centers and took their average as the baseline measurement. To evaluate the impact of perturbations, we introduced random standard Gaussian noise with varying scales to these cluster centers. The maximum perturbation scale was determined according to the $4\sigma/\text{dim}$ criterion described in assumption 4.1 where $\sigma$ represents the average population variance of the labels in the dataset, and dim is the dimension of the standard Gaussian noise. This ensures that

the largest expected distance to the cluster centers equals $4\sigma$. Finally, we plotted the mean discriminator values as a function of the perturbation scale in fig. 12, providing a visualization of how the discriminator responds to deviations from the original mode centers. The results demonstrate that the discriminator values approximately follow a linear trend near the mode centers, decreasing gradually from 0.5 to 0 as the distance from the mode centers increases.

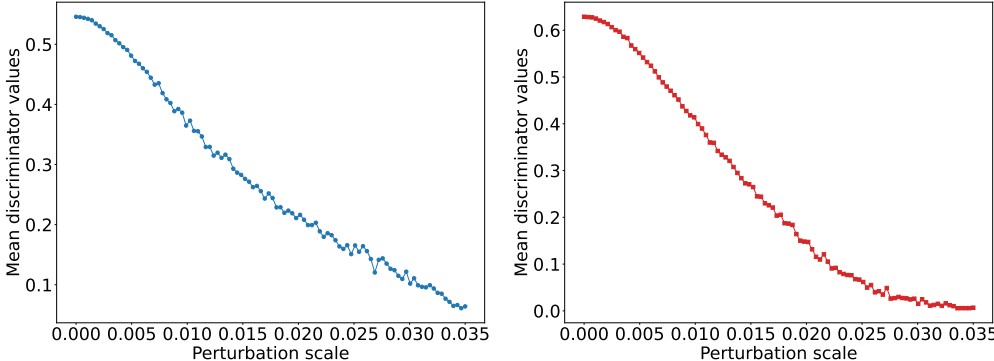

Figure 12: Mean discriminator values as a function of distance from the mode centers for MNIST (left) and Fashion MNIST (right). These results support assumption 4.1 by illustrating how the discriminator values vary with increasing distance from the mode centers. The mode centers are estimated using the $k$-means algorithm applied to each dataset. To introduce perturbations, random standard Gaussian noise with varying scales is added to these cluster centers. The maximum perturbation scale is set to $4\sigma/\text{dim}$, where $\sigma$ is the average population variance across dataset labels, and $\text{dim} = 64 \times 64$ represents the dimensionality of the perturbation. The plots reveal that the discriminator values approximately follow a linear trend near the mode centers, decreasing from 0.5 to 0 as the perturbation scale increases.

**Visualization of the discriminator gradient field.**  We plot the discriminator gradient field $\nabla d(x)/d(x)$ under assumption 4.1 in fig. 13. In the left panel, the unscaled vector field demonstrates the existence of particles that experience large-magnitude gradients, which could propel them away from one mode to another. In the right panel, the zoomed-in view shows the vector lengths reduced to one-tenth of their original size, yet some vectors still pass through the centers of the modes. This serves to verify the discussions in section 4.1.

## G.2 Verifying Fitting

**Annotated heatmaps for MNIST.**  We verify the existence of fitting on MNIST. Annotated heatmaps are employed to track the evolution of pairings $(i, j)$ occurrence within batches of size 256. The values depicted in these heatmaps represent the logarithm of occurrence counts plus 1, with darker colors indicating higher values. Each heatmap includes epoch numbers ranging from 0 to 38 displayed at the bottom. Initially, the heatmap has few nonzero entries, indicating limited sample diversity at the beginning of the fitting phase. As training advances, more entries became nonzero, reflecting a broader distribution of generated samples across the mode space. Notably, the values of off-diagonal entries signifies the severity of mode mixture, which gradually decrease over the course of training, validating the fitting phase. However, the issue of mode mixture persists even at the end of fitting. By the 36th epoch, the heatmap only has two nonzero entries, suggesting the collapse phase, where the generated samples become less diverse and concentrate around few modes. These observations provide empirical evidence for our proposed two phases of GAN training.

**Verifying fitting in Fashion MNIST.**  We verify the existence of fitting in Fashion MNIST using annotated heatmaps. The heatmap values are the logarithm of pairings $(i, j)$ occurrence plus 1 in batches of size 256, with darker colors indicating higher values. Each epoch is divided into 5 collections of batches, denoted as $e, b$ where $e$ is the epoch and $b$ is the batch collection within the epoch. Initially, there are only two nonzero entries, which suggests limited sample diversity. As training progresses, more entries become nonzero, indicating a broader sample distribution across the mode space during the fitting phase. Notably,

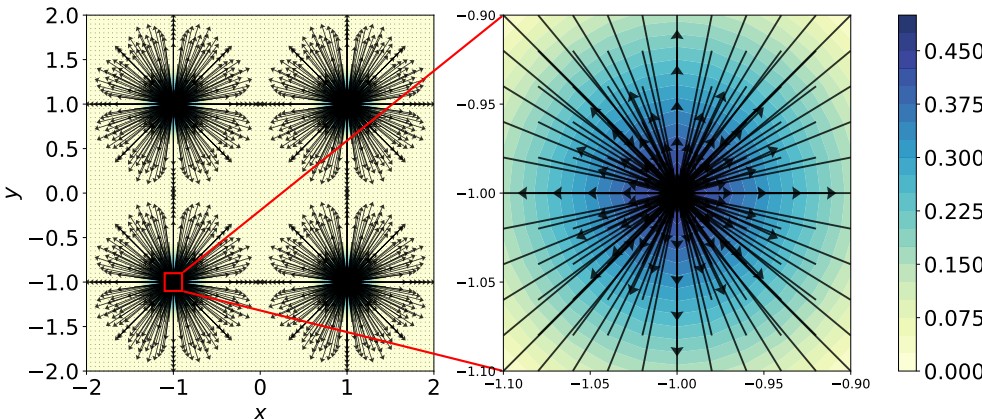

Figure 13: The discriminator gradient field $\nabla d(x)/d(x)$ under assumption 4.1 on the toy example. The left panel shows the unscaled vector field, where particles experience large-magnitude gradients, with vectors *passing through the mode centers and pointing outward*. This could trigger propulsion away to another mode. The right panel provides a zoomed-in view around the bottom-left mode, with the vector lengths reduced to one-tenth of their original size. Even with this scaling, some vectors are seen passing through the centers of the modes.

unlike MNIST, the phase of fitting in Fashion MNIST occur quickly, evidenced by the rapid stabilization of off-diagonal values. It is important to note that the large values in some off-diagonal entries do not necessarily imply severe mode mixture. For example, "T-shirt", "Pullover", and "Shirt" are frequently confused in Fashion MNIST classification tasks.

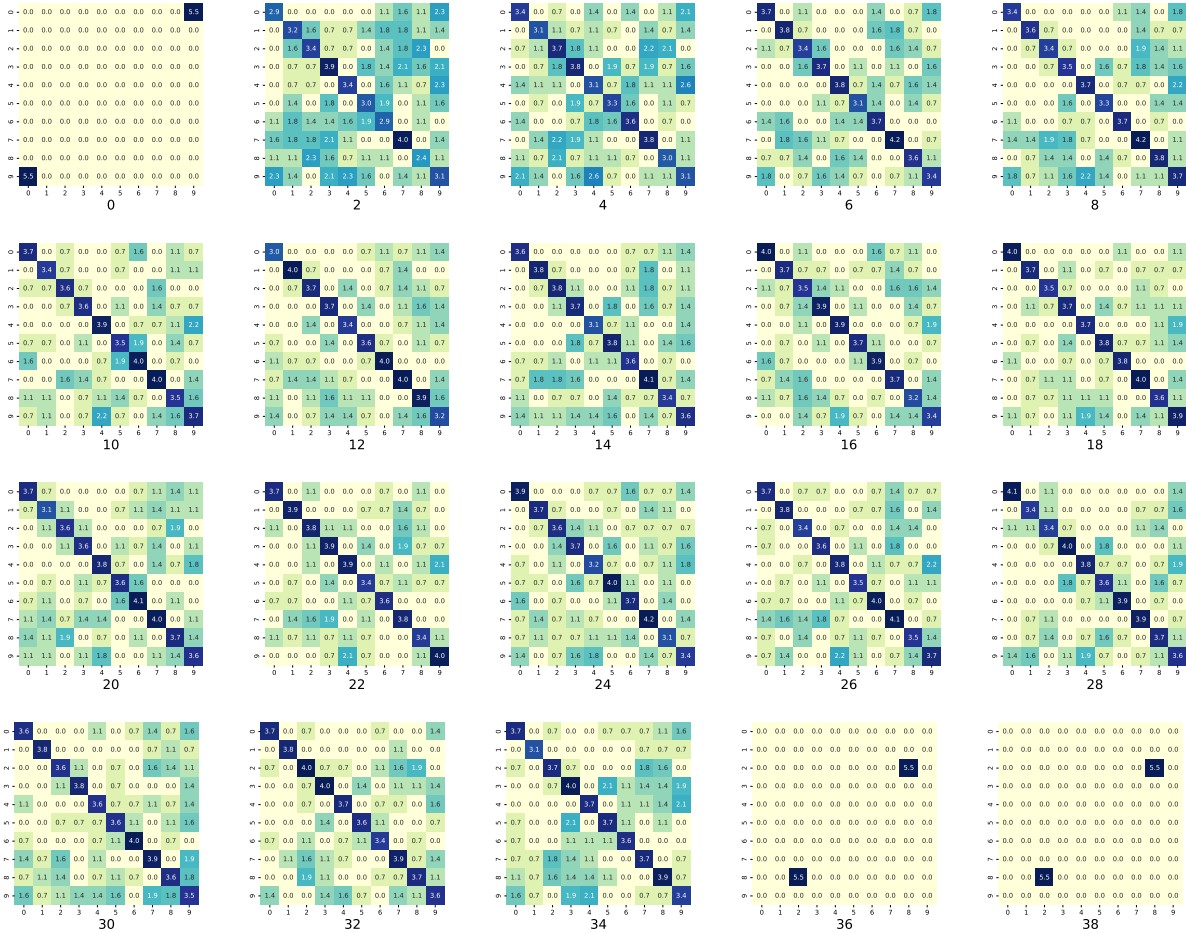

Figure 14: Annotated heatmaps for verifying fitting in MNIST. The values are the logarithm of the occurrence of pairings $(i, j)$ plus 1 in a batch of size 256. Darker colors indicate higher values. The epochs, ranging from 0 to 38, are displayed at the bottom of each heatmap. Initially, there are few nonzero entries, suggesting limited sample diversity. As training progresses, more entries become nonzero, indicating wider sample distribution across mode space, which corresponds to the fitting phase. Off-diagonal entries reflect mode mixture, which diminishes over training, confirming the fitting phase. Remarkably, mode mixture persists even at the closure of the fitting phase. Note that by the 36th epoch, only two entries remain nonzero, indicating the collapse phase.

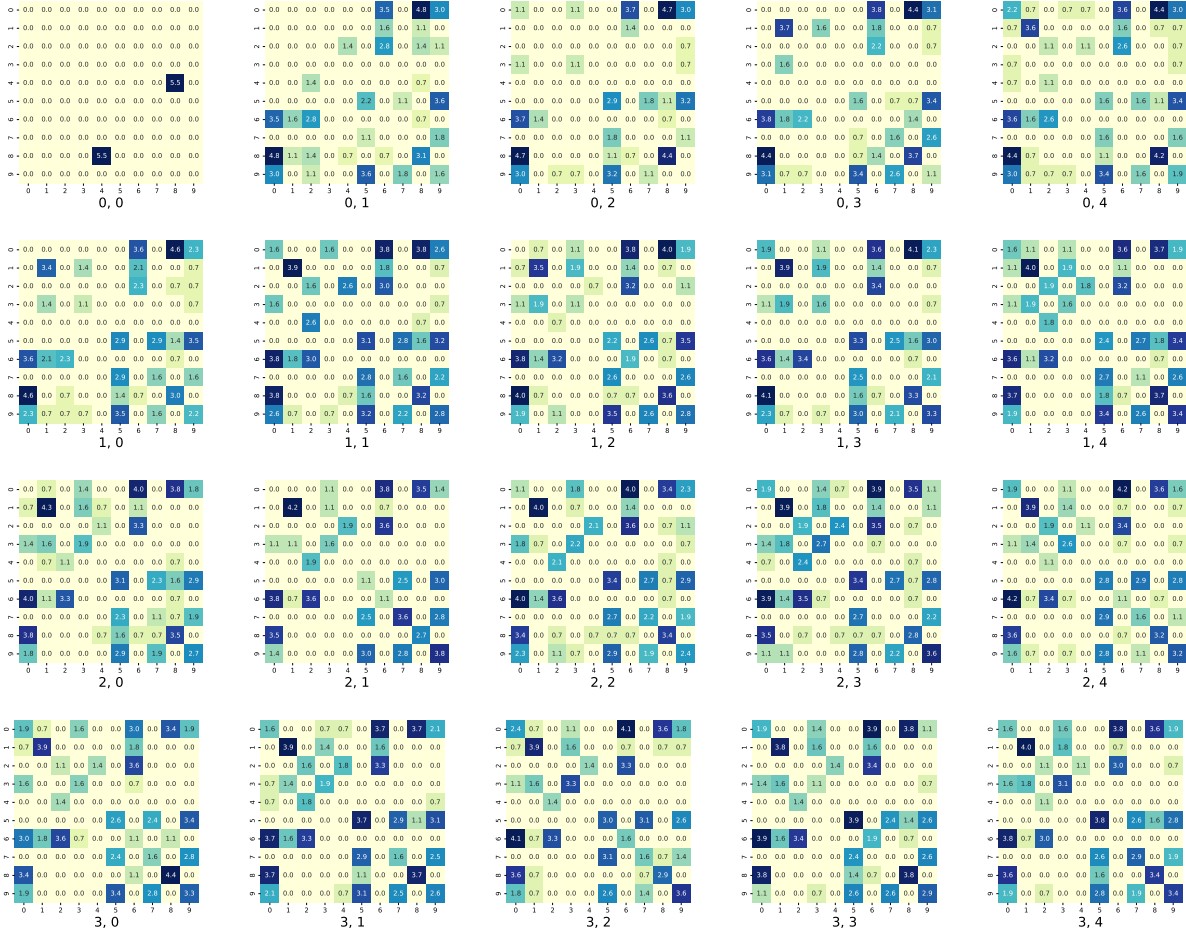

Figure 15: Annotated heatmaps for verifying fitting in Fashion MNIST. The labels 0 to 9 mean "T-shirt/top", "Trouser", "Pullover", "Dress", "Coat", "Sandal", "Shirt", "Sneaker", "Bag", and "Ankle boot", respectively. The values are the logarithm of the occurrence of pairings $(i, j)$ plus 1 in a batch of size 256. Darker colors indicate higher values. Each epoch is equally divided into 5 collection of batches. The label "$e, b$" at the bottom of each heatmap denotes the $b$th collection within the $e$th epoch. Therefore, the heatmaps displayed are for the first 4 epochs only. Initially, the few nonzero entries indicate limited sample diversity. As training progresses, more entries became nonzero, reflecting a broader sample distribution across the mode space, which corresponds to the fitting phase. Unlike MNIST, the phase of fitting in Fashion MNIST take place rapidly because the off-diagonal values stabilize quickly. It is important to note that the large values of some off-diagonal entries do not necessarily imply severe mode mixture; for instance, "T-shirt", "Pullover", and "Shirt" are often confused in Fashion MNIST classification tasks.

### G.3 Illustration of Particle Dynamics in Real Dataset

In this subsection, we use the CIFAR-10 dataset to illustrate particle dynamics in real dataset, particularly focusing on their relationship with mode collapse.

**Methods.** To investigate particle dynamics, we set fixed noise vectors and track their corresponding generated images (referred to as particles) across different epochs. For each particle, we compute two key metrics: (i) the discriminator gradient norm, $\|\nabla d(x)/d(x)\|_2$; and (ii) the steepness of the generator, $\|J_g(x)\|_2$. We visualize the generated images across epochs and analyze how these metrics evolve during training.

**Results.** From the visualizations, we observe that particles initially appear vague and gradually align with the modes, which supports our discussion in section 3.1. Over time, they move closer to the modes, as described in section 3.2, and eventually collapse. For the steepness metric, $\|J_g(x)\|_2$, we find that it increases in the early stages of training, likely due to the generator aligning the noise prior with the modes, consistent with the analysis in theorem 3.3. Later, it stabilizes but drops sharply to near zero at collapse theorem 4.2. In contrast, the discriminator gradient norm, $\|\nabla d(x)/d(x)\|_2$, shows oscillatory behavior throughout training but spikes sharply at the point of collapse, aligning with our findings in section 4.1. These results provide particle-level evidence that our theoretical findings approximately hold in real datasets, further validating our analysis.

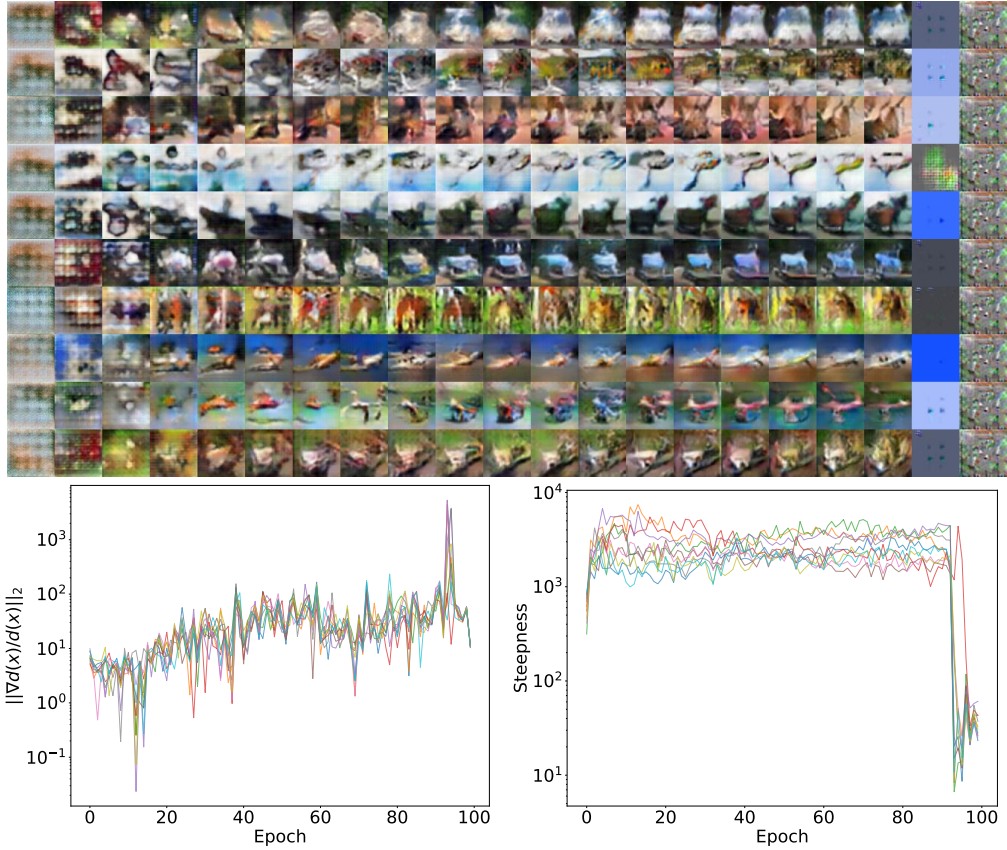

Figure 16: Visualization of particle dynamics on the CIFAR-10 dataset. The top row shows the evolution of particles (generated images) across training epochs, specifically at epochs 0, 4, 9, 14, ..., and 99. The bottom left plot illustrates the discriminator gradient norm, $\|\nabla d(x)/d(x)\|_2$, for each particle over training epochs (indicated by different colors), while the bottom right plot shows the steepness of the generator, $\|J_g(x)\|_2$. These visualizations highlight the transition of particles from initial noise to alignment with modes, and eventually to collapse, validating our theoretical findings.

### G.4 Comparison with the FID score

We present a comparison between $\|\nabla d(\boldsymbol{x})/d(\boldsymbol{x})\|_2$ and the FID score in fig. 17, complementing the analysis in section 6, where steepness was compared with the FID score. A connection emerges between the behavior of $\|\nabla d(\boldsymbol{x})/d(\boldsymbol{x})\|_2$ and the FID score: During the early training epochs, $\|\nabla d(\boldsymbol{x})/d(\boldsymbol{x})\|_2$ remains small, corresponding to a rapid decline in the FID score. As training progresses, $\|\nabla d(\boldsymbol{x})/d(\boldsymbol{x})\|_2$ begins to oscillate while gradually increasing, with the FID score decreasing at a slower rate. This trend likely reflects the discriminator's behavior approaching the condition implied by assumption 4.1. For MNIST and CIFAR-10, the stopping point is triggered by a proportional drop in steepness, without a concurrent rapid increase in both $\|\nabla d(\boldsymbol{x})/d(\boldsymbol{x})\|_2$ and the FID score. In the case of Fashion MNIST, our early stopping algorithm halts training when $\|\nabla d(\boldsymbol{x})/d(\boldsymbol{x})\|_2$ surpasses a predefined threshold, effectively preventing a deterioration in sample quality.

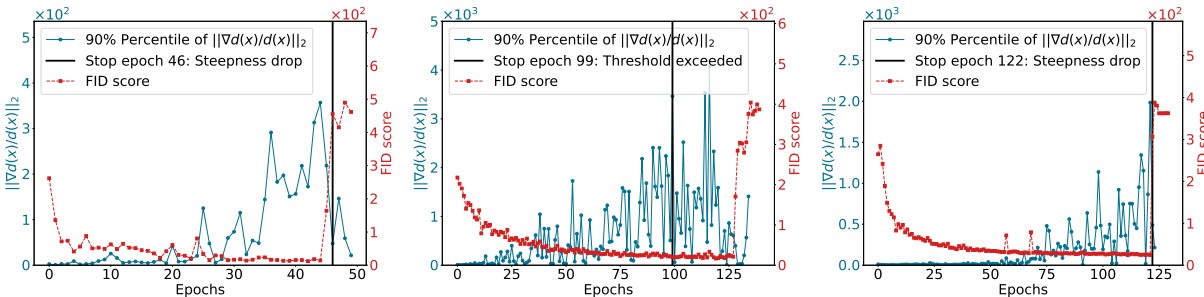

Figure 17: The tendency of $\|\nabla d(\boldsymbol{x})/d(\boldsymbol{x})\|_2$ and FID score for MNIST, Fashion MNIST, and CIFAR-10, presented from left to right. Blue circled for $\|\nabla d(\boldsymbol{x})/d(\boldsymbol{x})\|_2$ and red square-shaped for the FID score. For MNIST and CIFAR-10, since the stopping point is primarily triggered by steepness dropping below the threshold, we do not observe a concurrent rapid increase in $\|\nabla d(\boldsymbol{x})/d(\boldsymbol{x})\|_2$ and the FID score. In contrast, for Fashion MNIST, when $\|\nabla d(\boldsymbol{x})/d(\boldsymbol{x})\|_2$ surges past its threshold, we halt training at a point where a deterioration in sample quality is prevented.

### G.5 Comparison with the Duality Gaps

We compare our metrics, $\|\nabla d(\boldsymbol{x})/d(\boldsymbol{x})\|_2$ and steepness, with the duality gap (Grnarova et al., 2019), along with its improved counterpart, the perturbed duality gap (Sidheekh et al., 2021). We first briefly introduce the two metrics, and then show the results in fig. 18.

**Duality gap.** The duality gap is an optimization concept that measures the difference between the primal and dual forms of a problem. In GANs, it quantifies the suboptimality of the current generator and discriminator. For parameters $(\theta_g, \theta_d)$ at a given iteration, the duality gap is defined as:

$$\mathrm{DG}(\theta_g, \theta_d) = \max_{\theta_d' \in \Theta_d} F(\theta_g, \theta_d') - \min_{\theta_g' \in \Theta_g} F(\theta_g', \theta_d), \tag{168}$$

where $\Theta_d$ and $\Theta_g$ are the parameter spaces for the discriminator and generator, respectively, and $F$ is the objective function of the Vanilla GAN: $F(\theta_g, \theta_d) = \mathbb{E}_{\boldsymbol{x} \sim p_{\mathrm{data}}}[\log d(\boldsymbol{x})] + \mathbb{E}_{\boldsymbol{z} \sim p_z}[\log(1 - d(g(\boldsymbol{z})))]$. In practice, Grnarova et al. (2019) proposed to estimate the duality gap through the following steps:

1. Train the GAN to iteration $t$, obtaining parameters $(\theta_g^t, \theta_d^t)$.

2. Find the worst-case discriminator and generator by optimizing one while keeping the other fixed:

$$\theta_d^{\mathrm{worst}} \approx \arg\max_{\theta_d' \in \Theta_d} F(\theta_g^t, \theta_d'), \quad \theta_g^{\mathrm{worst}} \approx \arg\min_{\theta_g' \in \Theta_g} F(\theta_g', \theta_d^t). \tag{169}$$

3. Estimate the duality gap as: $\mathrm{DG}(\theta_g^t, \theta_d^t) \approx F(\theta_g^t, \theta_d^{\mathrm{worst}}) - F(\theta_g^{\mathrm{worst}}, \theta_d^t)$.

**Perturbed duality gap.** The perturbed duality gap, introduced by Sidheekh et al. (2021), improves upon the standard duality gap by more effectively distinguishing between Nash and non-Nash critical points. This metric performs local perturbations to the parameters $(\theta_g^t, \theta_d^t)$ with Gaussian noise before the second optimization step, helping the model escape from saddle points. This ensures the subsequent optimization does not get stuck in suboptimal regions.

**Results.** We compare $\|\nabla d(\boldsymbol{x})/d(\boldsymbol{x})\|_2$ and the steepness with the vanilla and perturbed duality gaps across three datasets: MNIST, Fashion MNIST, and CIFAR-10, as shown in fig. 18. In the first row, $\|\nabla d(\boldsymbol{x})/d(\boldsymbol{x})\|_2$ is plotted alongside the duality gaps. In the second row, the steepness is compared against the duality gaps. Prior to the collapse, both duality gaps decrease with oscillations, while $\|\nabla d(\boldsymbol{x})/d(\boldsymbol{x})\|_2$ increase with oscillations. After collapse, the vanilla duality gap drops to zero, mirroring the behavior of steepness. In contrast, the perturbed duality gap oscillates, making it difficult to pinpoint the beginning of collapse. These results demonstrate the robustness of our metrics, which consistently and clearly detect the collapse phase.

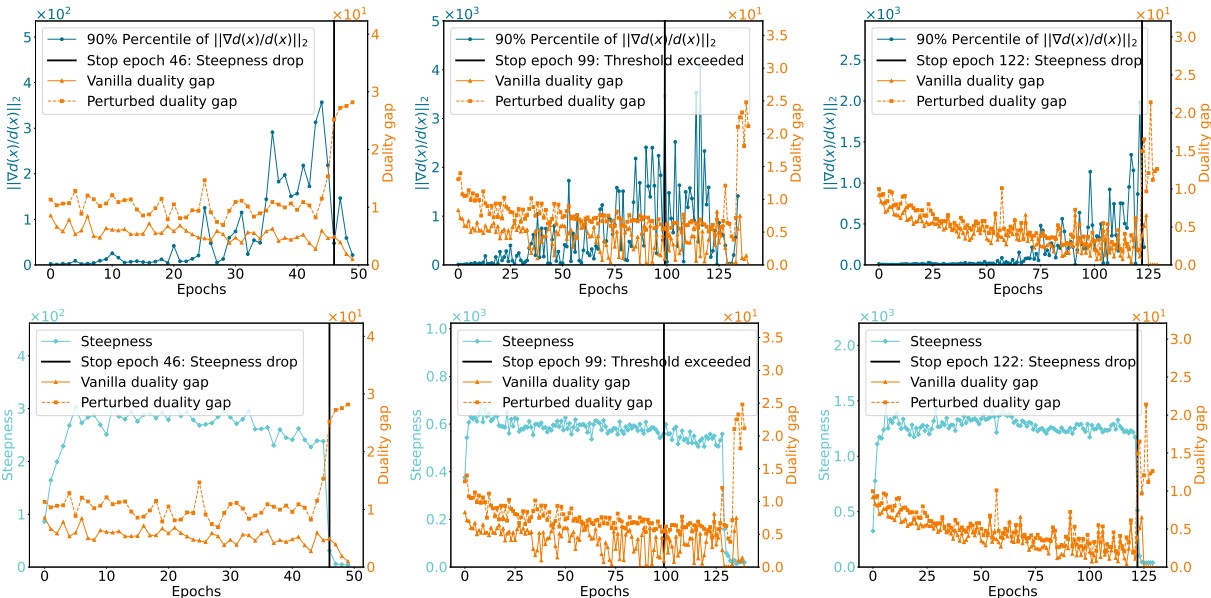

Figure 18: The first row compares $\|\nabla d(\boldsymbol{x})/d(\boldsymbol{x})\|_2$ with the duality gaps for MNIST, Fashion MNIST, and CIFAR-10, from left to right. The second row compares the steepness with the duality gaps for the same datasets. Blue circled represents $\|\nabla d(\boldsymbol{x})/d(\boldsymbol{x})\|_2$, light blue square-shaped represents steepness, while orange triangle-shaped and orange square-shaped represent the vanilla and perturbed duality gaps, respectively. Prior to the collapse, $\|\nabla d(\boldsymbol{x})/d(\boldsymbol{x})\|_2$ exhibits a similar trend with the perturbed duality gap. After collapse, the vanilla duality gap often drops to zero, mirroring the behavior of steepness, while the perturbed duality gap oscillates, making it difficult to determine whether collapse has occurred.

## G.6 GAN Training Losses

We present the loss curves under the same experimental settings as in figs. 5, 6 and 17 to 19. The loss functions are defined as

$$\begin{aligned}
\mathcal{L}_{d_\omega} &= -\mathbb{E}_{\boldsymbol{x}\sim p_{\text{data}}}[\log(d_\omega(\boldsymbol{x}))] - \mathbb{E}_{\boldsymbol{z}\sim p_z}[\log(1 - d_\omega(g_\theta(\boldsymbol{z})))], \\
\mathcal{L}_{g_\theta} &= -\mathbb{E}_{\boldsymbol{z}\sim p_z}[\log(d_\omega(g_\theta(\boldsymbol{z})))],
\end{aligned} \quad (170)$$

and are computed using `torch.nn.functional.binary_cross_entropy`. The blue circled markers stand for discriminator losses, while the red square-shaped markers represent generator losses. Notably, after the collapse, the discriminator and generator losses either escalate to large values or drop to zero. While this divergence behavior is consistent across experiments, it poses challenges when used as an early stopping

criterion, particularly in encoding a practical termination condition. The simplest approach is perhaps monitoring both losses and terminating when they increase or decrease beyond a certain threshold. However, this strategy comes with several limitations: (i) the interpretability of GAN losses is limited. GAN losses do not directly reflect the quality of generated samples. For instance, an increase in discriminator loss might indicate overfitting rather than a collapse of generator performance, while an increase in generator loss could stem from the discriminator becoming overly sensitive, leading to loss fluctuations without a corresponding drop in sample quality. (ii) threshold values may be difficult to generalize across experiments. The dynamics of GAN training are highly sensitive to hyperparameters such as learning rates, batch sizes, and architecture choices. Consequently, the patterns of loss escalation can vary significantly, making it challenging to define a universal threshold for early stopping. Nevertheless, it is interesting to see if GAN loss functions can also be used for designing alternative early stopping criteria that leverage their dynamic behaviors to predict mode collapse.

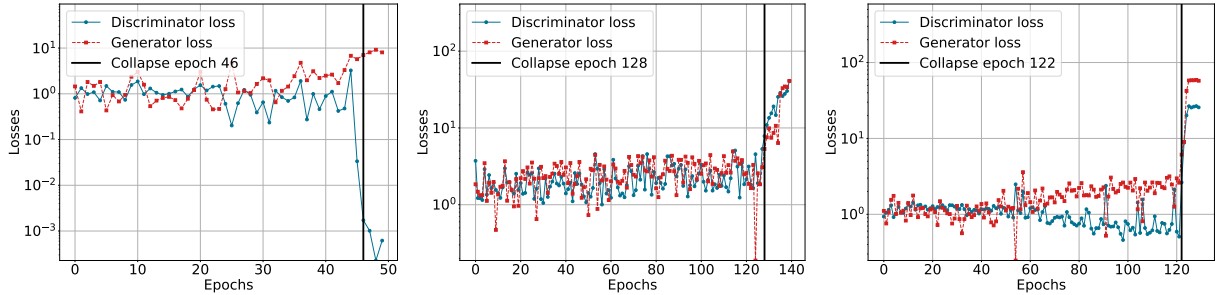

Figure 19: Loss curves of the discriminator and generator in experiments on MNIST, Fashion MNIST, and CIFAR-10 (from left to right). Blue circled represent discriminator losses, while red square-shaped represent generator losses. The black vertical lines indicate the stopping epochs determined by our early stopping algorithm. The exact collapse epochs are 46, 128 and 122, for the respective datasets. Notably, after the collapse, both losses either escalate to large values or drops to zero.

### G.7 Impact on the Early Stopping Metric after Applying Techniques to Mitigate Mode Collapse

In this subsection, we validate our early stopping metric's effectiveness by demonstrating that injecting noise into the intermediate layers of the discriminator combats mode collapse and pushes back the metric.

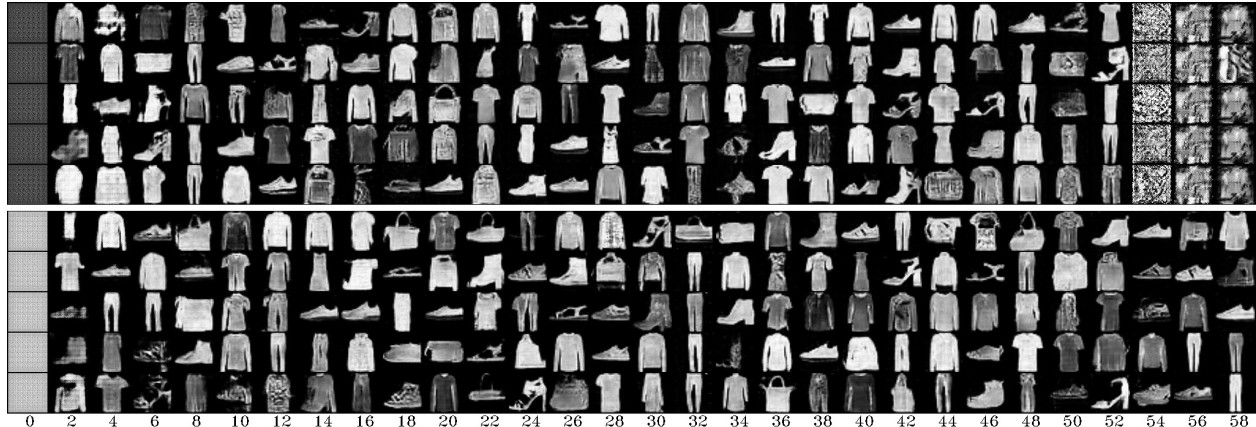

Figure 20: The generated images from the noise-free GAN and the noised GAN. **Upper**: Noise-free GAN. **Lower**: Noised GAN. The noise-free GAN collapses at the 54th epoch, whereas the noised GAN consistently produces high-quality images.

**Experimental setup.** We devise two generator models of identical architecture and implement two discriminators, one adhering to the original design (which we will refer to as "noise-free") and the other modified to incorporate Gaussian noise with a standard deviation of 0.1 before forwarding the input to the subsequent layer (which we will refer to as "noised"). Both generators and discriminators are initialized using the same random seed. During training, the four networks are concurrently trained, with each generator paired with a discriminator. We present the generated images of the two models on Fashion MNIST in fig. 20 and histograms of $\|\nabla d(\boldsymbol{x})/d(\boldsymbol{x})\|_2$ in fig. 21.

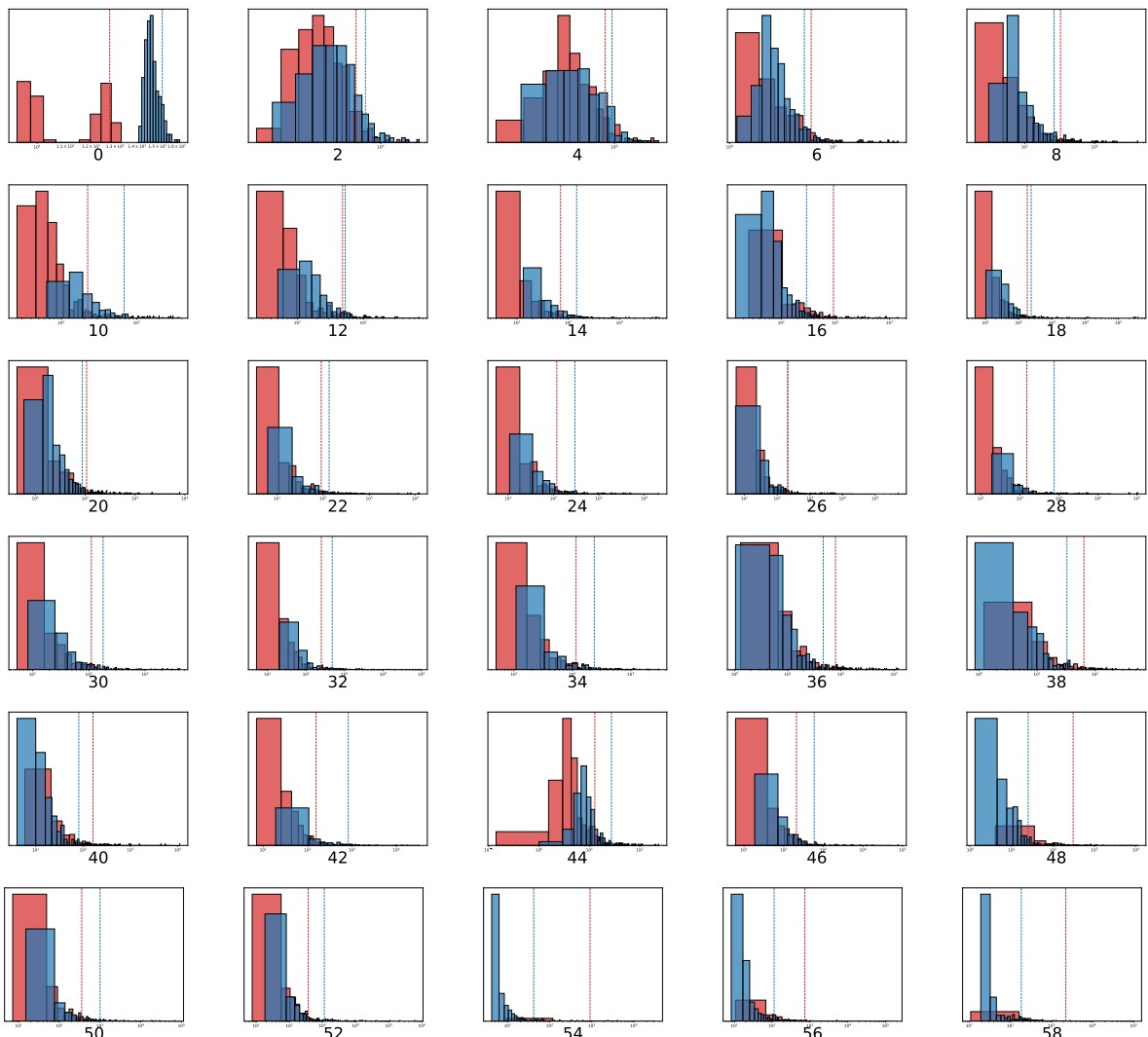

Figure 21: Histograms of the values of $\|\nabla d(\boldsymbol{x})/d(\boldsymbol{x})\|_2$ and their 90th percentile across epochs. The epochs are displayed at the bottom of each histogram. The $x$-axis represents $\|\nabla d(\boldsymbol{x})/d(\boldsymbol{x})\|_2$ values on a *logarithmic* scale, while the $y$-axis denotes density. Results are differentiated by color: red for the model with noise and blue for the model without noise. Preceding the 54th epoch where the noise-free GAN collapses, the noised model nearly always exhibits lower $\|\nabla d(\boldsymbol{x})/d(\boldsymbol{x})\|_2$ values compared to its noise-free counterpart. Post 54th epoch, this relationship reverses. Notably, in the noise-free model, $\|\nabla d(\boldsymbol{x})/d(\boldsymbol{x})\|_2$ tends towards zero, contributing to this observed divergence.

**Results.** The noise-free GAN collapses at the 54th epoch, while the noised GAN consistently generates high-quality images. The introduction of noise results in an overall decrease in the $\|\nabla d(\boldsymbol{x})/d(\boldsymbol{x})\|_2$ compared to its noise-free counterpart before the 54th epoch. After the 54th epoch, the opposite trend is observed,

attributed to the vanishing of $||\nabla d(\boldsymbol{x})/d(\boldsymbol{x})||_2$ in the noise-free GAN. This indicates that the strategy of injecting noise to mitigate mode collapse leads to an overall decrease in our proposed metric, thereby validating the effectiveness of the metric.

Furthermore, from our experiments, we observe that collapse is avoided altogether in the noised GAN. Specifically, we trained the noised GAN for 1000 epochs and did not detect any signs of collapse throughout the training process. This supports the understanding that adding noise to the discriminator stabilizes GAN training by mitigating extreme gradient magnitudes near mode boundaries, which are a primary contributor to collapse. Regarding the early stopping algorithm, it remains effective in the noised GAN setting. The algorithm identifies a point in the training process where sample quality and diversity are well-retained. However, we do not observe significant sample deterioration or signs of subtle collapse even after this identified point. This suggests that the early stopping algorithm's role in the noised setting may focus more on preserving computational efficiency rather than strictly avoiding collapse. A possible explanation for this behavior is that the added noise reduces the discriminator's capacity to provide overly precise gradients, which prevents the discriminator from overfitting to specific modes. This stabilization mechanism ensures that the generator continues to explore and cover the data distribution effectively, even in later training stages. As a result, even subtle collapses do not seem to occur.

## H    Extension to Other Divergence GANs

In this section, we outline how to extended to other Divergence GANs. We focus on the $f$-GAN proposed in (Nowozin et al., 2016) with the $f$-divergence defined as

$$D_f(Q_\theta||p_{\text{data}}) = \int_{\boldsymbol{x}} p_{\text{data}}(\boldsymbol{x}) f\Big(\frac{p_{\text{data}}(\boldsymbol{x})}{Q_\theta(\boldsymbol{x})}\Big) \mathrm{d}\boldsymbol{x}. \tag{171}$$

The variational lower bound of $D_f(Q_\theta||p_{\text{data}})$ is used as the training objective:

$$F(\theta;\omega) = \mathbb{E}_{\boldsymbol{x} \sim p_{\text{data}}} \big[ g_f\big(V_\omega(\boldsymbol{x})\big)\big] + \mathbb{E}_{\boldsymbol{x} \sim Q_\theta}\big[ -f^*\big(g_f(V_\omega(x))\big)\big]. \tag{172}$$

Here, $f^*$ is the Fenchel conjugate of $f$, $g_f$ is analogous to the generator and $V_\omega$ is similar to the discriminator. We consider its variant where the objective function of the generator is modified to

$$-\mathbb{E}_{\boldsymbol{x} \sim Q_\theta}\big[g_f\big(V_\omega(\boldsymbol{x})\big)\big], \tag{173}$$

while the objective function of the discriminator remains unchanged.

**General methodology.**    The key to analyzing Divergence GANs is their interpretation as particle models. The update of the generator $Q_\theta$ can be recast as: (i) generate particles $Z_i = Q_\theta(z_i)$; and (ii) update the particles $\hat{Z}_i = Z_i + g'_f(V_\omega(Z_i))\nabla V_\omega(Z_i)$; and (iii) update $\theta$ by descending its stochastic gradient with respect to the Mean Square Error (MSE) loss betweeen $\hat{Z}_i$'s and $g(z_i)$'s.

**Fitting phase.**    We may plot the vector field $g'_f(V_\omega(Z_i))\nabla V_\omega(Z_i)$ instead of the original $\nabla d(\boldsymbol{x})/d(\boldsymbol{x})$ to visualize the updating process of particles, which promotes the fitting of the modes. Only theorems 3.3 and 3.4 in section 3.2 needs to be modified to accommodate the desired Divergence GAN.

**Collapse phase.**    In section 4.1, apart from modifying the update formula for particles, a more appropriate model for the discriminator needs to be established and a new threshold may be developed on the basis of it.

## I    Visualizing Generator Functions

This section visualizes generator functions $g$ that satisfy $g_\# p_z = p_{\text{data}}$, where $p_z \sim \mathcal{N}(0,1)$ and $p_{\text{data}}$ is a Gaussian mixture, as shown in fig. 22. For qualitative effects of the parameters in $p_{\text{data}}$, please refer to table 3. We then discuss about how to plot fig. 22. While $\Phi$ can be computed in MATLAB using the

built-in function `normcdf`, $\Psi^{-1}$ typically necessitates solving a non-linear equation at each evaluation point. To mitigate computational expenses, we choose to calculate the inverse of $g$, which is $g^{-1} = \Phi^{-1} \circ \Psi$. In this context, $\Psi$ can be computed by employing `gmdistribution` to construct a Gaussian mixture model, followed by utilizing `cdf` to assess the cumulative distribution function (CDF) of the model at a specific point. To generate a plot of $g$, a mere interchange of the $x$ and $g^{-1}(x)$ in the `plot` function suffices.

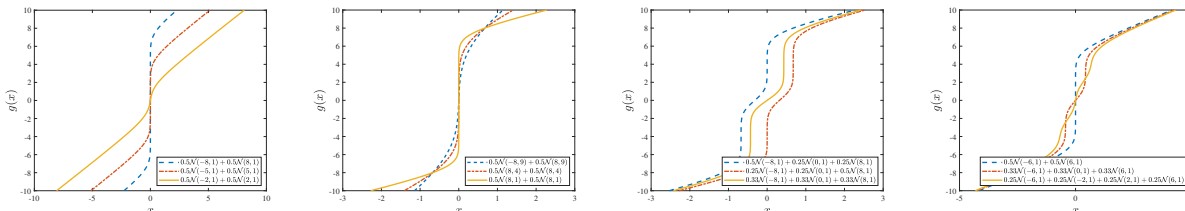

Figure 22: The functions $g$ that satisfy $g_{\#}p_z = p_{\text{data}}$, where $p_{\text{data}}$ is a Gaussian mixture. **First**: Varying the mean $\mu$. **Second**: Varying the variance $\sigma^2$. **Third**: Varying the mixing coefficients $\{\alpha_i\}_{i=1}^n$. **Fourth**: Varying the number of Gaussians $n$. Please refer to table 3 for a detailed description.

Table 3: Qualitative effects of the parameters in $p_{\text{data}} \sim \alpha_1 \mathcal{N}(\mu_1, \sigma^2) + \cdots + \alpha_n \mathcal{N}(\mu_n, \sigma^2)$ on $g$.

| Parameters | Qualitative Effects on $g$ |
|---|---|
| Means $\{\mu_i\}_{i=1}^n$ | Larger $\|\mu_i - \mu_{i+1}\|_2$ increases the magnitude of $g'$ between the two modes. |
| Variances $\sigma^2$ | Larger $\sigma^2$ increases the asymptotic slope of $g$ as $x \to \infty$. |
| Mixing coefficients $\{\alpha_i\}_{i=1}^n$ | Different combinations of $\alpha_i$ incline $g$ towards specific modes. |
| Number of Gaussians $n$ | Larger $n$ increases the number of segments in $g$. |

## J  Discussions

In this section, we provide additional intuitions and implications.

In terms of applicability scope, our theoretical findings are primarily derived from Divergence GANs, specifically NSGAN, where we can leverage their particle model interpretations. While Divergence GANs represent a significant category within GANs, they do not encompass some prominent GAN models, such as Wasserstein GAN with gradient penalty and MMD GAN. Exploring how our theoretical findings can be extended to incorporate these Integral Probability Metric (IPM) based GAN variants presents an intriguing avenue for future research.

Regarding the proposed two phases, it is important to note that not all Divergence GANs may fit neatly into the this characterization. While we often observe such empirical patterns, we acknowledge the possibility that when networks are not well-initialized or when advanced techniques are used, GAN training may deviate from the fitting phase entirely. However, these inquiries may spark independent interests and are beyond the scope of our study.

In our numerical experiments, we used relatively small-scale real-world datasets compared to modern datasets. This choice was deliberate as we aimed to assess the effectiveness of our early stopping algorithm in detecting the transition from fitting to collapse phases. Modern datasets often comprise exponentially more modes, which could potentially limit the efficacy of our algorithm, particularly considering that our algorithm takes the number of modes as an input parameter.

