# OpenReview forum: "Evolution of Discriminator and Generator Gradients in GAN Training: From Fitting to Collapse"
_TMLR — Accepted by TMLR_

### Review · Reviewer_dg58 · 2024-12-30

**Summary Of Contributions:**

I was a reviewer of the previous version of this manuscript and I appreciate the significant changes that the authors have made to the new manuscript. In particular, it addressed the original concerns that were discussed in previous reviews, particularly regarding citing related works and clarifying the novelty points.

The current version is an in-depth analysis of the training dynamics of GANs from the gradient perspective, and the proposed analysis leads to a practical algorithm that helps detect model collapse during training. I think this contribution makes a valuable contribution to the practice of GAN training.

**Audience:**

Yes

**Claims And Evidence:**

Yes

**Requested Changes:**

See above

**Strengths And Weaknesses:**

The paper demonstrates solid theoretical foundations and empirical rigor in analyzing GAN training dynamics, particularly through its well-structured verification experiments. I commend the authors' improvements in Section 2, where they've clearly laid out their assumptions and maintained consistency throughout the paper. The presentation has notably improved by focusing on the early stopping methodology and its empirical validation.

However, I have two main concerns. First, the experimental validation would be more convincing if extended beyond CIFAR-10, where the reported FID scores exceed 100 – significantly higher than current standards. I strongly encourage the authors to evaluate their proposed criterion on either modern GAN architectures that achieve FID scores of 10-30 on CIFAR-10, or on larger-scale datasets such as LSUN or ImageNet.

My second point concerns the paper's title. While the revisions have substantially improved the manuscript, the current title "Magnifying Three Phases of GAN Training" seems disconnected from the paper's core contributions. The work more accurately reflects an understanding or characterization of GAN training phases, with particular emphasis on addressing collapse issues. A title revision would better serve potential readers and more accurately represent the paper's contributions.

---

> ### Author Response · Authors · 2025-01-23
> **Thank You and a Point-by-Point Response to Comments**
>
> Dear Reviewer dg58,
>
> We sincerely thank you for your thoughtful feedback and valuable suggestions. Your comments have provided clear guidance to improve the quality and clarity of our work. We also deeply appreciate your careful evaluation of the revised manuscript and your recognition of the improvements made, especially in addressing prior concerns. Below, we respond to your comments point by point.
>
> ## Experimental validation
>
> > However, I have two main concerns. First, the experimental validation would be more convincing if extended beyond CIFAR-10, where the reported FID scores exceed 100 – significantly higher than current standards. I strongly encourage the authors to evaluate their proposed criterion on either modern GAN architectures that achieve FID scores of 10-30 on CIFAR-10, or on larger-scale datasets such as LSUN or ImageNet.
>
> Thank you for your valuable suggestion. We appreciate your feedback and have conducted additional experiments on modern GAN architectures that achieve FID scores of 10-30 on CIFAR-10. **In our experiments, the lowest FID achieved was approximately 24.** To ensure consistency across our evaluation, **we have also replaced the MNIST and Fashion MNIST results with experiments using the same architectures**. Notably, our early stopping criterion remains effective in these settings. Please refer to Figure 6 for detailed results and visualizations.
>
> ## Paper's title
>
> > My second point concerns the paper's title. While the revisions have substantially improved the manuscript, the current title "Magnifying Three Phases of GAN Training" seems disconnected from the paper's core contributions. The work more accurately reflects an understanding or characterization of GAN training phases, with particular emphasis on addressing collapse issues. A title revision would better serve potential readers and more accurately represent the paper's contributions.
>
> Thank you for your insightful suggestion regarding the paper's title. Based on your feedback, we have revised the title to **"Evolution of Discriminator and Generator Gradients in GAN Training: From Fitting to Collapse"**. This new title emphasizes the primary theoretical tools used in our analysis --- discriminator and generator gradients --- while also highlighting the phases of GAN training, which remain central to our work.
>
> Additionally, we have merged the previously defined phases of fitting and refining into a single fitting phase, based on another reviewer's feedback. This decision was motivated by the observation that the transition between these two phases was often fluid and less distinctly separable. By combining them, we present a more cohesive and streamlined characterization of GAN training dynamics, which better aligns with the overall narrative and avoids unnecessary complexity.
>
> We hope this revision effectively addresses your concern and makes the paper's contributions clearer to potential readers.
>
> ----
>
> We sincerely thank you again for your valuable feedback. If you have any additional questions or suggestions, we would be delighted to address them and engage in further discussion.
>
> Best regards,
>
> Paper3757 Authors

---

> > ### Comment · Reviewer_dg58 · 2025-01-26
> > **Acknowledgment**
> >
> > I thank the authors for the detailed reply. I think my former concerns are well addressed and I think the new title better reflects main contributions of this work. I have no more further questions.

---

### Review · Reviewer_AFrb · 2025-01-12

**Summary Of Contributions:**

This paper breaks down non-saturating GAN training into three phases (fitting, refining, collapsing) and analyses each of them using the spatial gradients of the generator and the discriminator with the following conclusions. In the first phase (fitting), discriminator gradients encourage the generator to cover all modes of the data distribution. In the second phase (refining), the generator's steepness (related to its Lipschitz constant) provably needs to increase in order to fit a multimodal data distribution. In the third phase (collapsing), mode collapse occurs, attributed to an exploding discriminator gradient; together with decreased steepness of the generator, the discriminator's gradient magnitude constitutes a criterion for detecting mode collapse in a proposed early stopping strategy.

**Audience:**

Yes

**Claims And Evidence:**

Yes

**Requested Changes:**

Most of the aforementioned issues are crucial for acceptance at TMLR. I would strongly recommend the authors to:
- improve the writing to better highlight conclusions and provide interpretations in the overall context of the paper;
- clarify or amend the unsupported claims, especially on the phases' characterization;
- discuss prior work on generator steepness and training collapse.

Details are provided above.

---

## Post-rebuttal update

As highlighted in the discussion with the authors below, the vast majority of my concerns have been successfully addressed in the revisions of the paper. Writing is significantly improved, the (modified) claims are now correctly supported, and close related work has been discussed.

**Disclaimer.** Before providing a recommendation, I would like to state that I did not have the time to assess the proofs of theoretical results in the appendix.

**Strengths And Weaknesses:**

## Strengths

The paper highlights **interesting insights into GAN training**. To my knowledge, this is the first paper decomposing GAN training into phases. This allows the paper to present mode collapse in a global picture as the outcome of the optimization process trying to alleviate mode mixture.

**The proposed early stopping algorithm is interesting and novel**, grounded in the theoretical and empirical observations using the paper's framework. It may serve as a useful tool for practitioners to easily detect the collapse of GAN training.

I appreciate the balanced mix of theoretical results and empirical observations in toy and real-world examples made in the paper that **grounds the conclusions into GAN practice**.

## Weaknesses

The paper suffers from three important issues, described below.

### Writing and Clarity

While the paper is easy to read, the message it conveys is not clearly conveyed in the analysis of the three phases. **Most results, while potentially interesting, lack contextualization and an explicit conclusion** that would help the reader interpret them in the overall context of the paper. This is particularly the case in Section 3 where I do not know what to conclude from the empirical observations, and in Section 4.2 where there is no interpretation of the theoretical results.

### Insufficiently Supported Claims

Partly due to this writing problem, several of the paper's central claims are not sufficiently supported.

**(1.)** First of all, in contradiction with the abstract, introduction and conclusion, the paper **does not characterize GAN training into three phases**. Such a characterization would necessitate a clear separation between phases and a precise definition of each phase, unlike the high-level description of Table 1. While collapsing is clearly identified in Section 5 and by the early stopping criterion, **the other two phases and not properly identifiable**. Fitting is especially blurry as explained above, and its separation with refining is never described. In its current state, the paper is more of an analysis of the training dynamics of GANs before and after collapse.

**(2.)** The paper **overclaims that Algorithm 2** "balances sample quality and diversity" (abstract) when it actually only stops training when collapse occurs as seen in Figures 5 and 6, and that it "offers enhanced sensitivity and reliability in detecting mode collapse compared to existing approaches" (Section 6.2). For the last assertion to hold, the paper should at least explain why monitoring the FID is not sufficient (as it aligns with the proposed criterion) and show whether collapse can already be detected in the GAN losses.

**(3.)** **Section 5.1's assumptions are insufficiently grounded**, making its conclusion questionable.
- There is no explanation why $p_g \approx p_{\mathrm{data}}$ close to modes.
- Assumption 5.1 artificially makes the discriminator ill-conditioned without proper justification. The referenced Appendix G does not show such a sharp transition in-between modes for the discriminator.

**(4.)** **The steepness and the identification of mode collapse as the last stage of GAN training** are presented as novel in page 1, but the paper **lacks sufficient discussion of the literature** for this claim to hold. Details are given below.

### Prior Work Discussion

To my understanding, **the notion of generator steepness, including its link with GAN trainability and minimal steepness requirements, has already been studied in the literature**, with a few variations, by, notably, Odena et al. (2018), Tanielian et al. (2020) and Salmona et al. (2022). Given the potentially strong links with the paper's contributions, a discussion of the paper's novelty w.r.t. the existing literature should be included.

Furthermore, the paper **undermines pre-existing knowledge of GAN training and collapse in the literature**, for example via the assertion "[the paper] challenges the conventional view that mode collapse signifies GAN training failure" (page 2) -- which should be motivated by one or several references. I would like to point out in particular the seminal paper of Brock et al. (2019) documenting and analyzing collapse in the final training stages.

Brock et al. Large scale GAN training for high fidelity natural image synthesis. ICLR 2019.\
Odena et al. Is generator conditioning causally related to GAN performance? ICML 2018.\
Salmona et al. Can push-forward generative models fit multimodal distributions? NeurIPS 2022.\
Tanielian et al. Learning disconnected manifolds: a no GAN’s land. ICML 2020.

### Other Issues

- Algorithm 2 necessitates to know the number of modes of the data distribution and to estimate the generator's steepness. The paper would benefit from a discussion on the feasibility of these choices.
- It is not clear how Assumption 2.1 (Gaussian smoothing of the data) is related to minibatch training. I believe this assumption can be justified more directly by noticing that data smoothing has now become a standard even beyond diffusion models.
- I assume that $\omega$ denotes the discriminator's parameters, but this is not specified in the main paper.

---

> ### Author Response · Authors · 2025-01-23
> **Thank You and a Point-by-Point Response to Comments (Part 1)**
>
> Dear Reviewer AFrb,
>
> We sincerely thank you for your thoughtful feedback and valuable suggestions. Your comments have provided clear guidance to improve the quality and clarity of our work. We deeply appreciate the time and effort you have devoted to reviewing our paper. Below, we address your concerns point by point.
>
> ## Writing and Clarity
>
> > While the paper is easy to read, the message it conveys is not clearly conveyed in the analysis of the three phases. Most results, while potentially interesting, lack contextualization and an explicit conclusion that would help the reader interpret them in the overall context of the paper. This is particularly the case in Section 3 where I do not know what to conclude from the empirical observations, and in Section 4.2 where there is no interpretation of the theoretical results.
>
> We appreciate your constructive feedback regarding the clarity and contextualization of our analysis across the three phases. In response, we have made several revisions to address your concerns and improve the readability and interpretability of our results. Below, we elaborate on the improvements made in Section 3 (now Section 4.1) and Section 4.2, followed by a summary of the broader changes regarding the discussion of theoretical results.
>
> **For Section 3 (now Section 4.1)**, we acknowledge that its conclusions were previously scattered due to the varied scenarios in the fitting phase. To improve clarity, we added:
>   - **An introduction paragraph** to clarify the section’s purpose.
>   - **Additional descriptions** to each case for better context.
>   - **A modified Figure 2**, which now includes a zoomed-in view of the vector field.
>   - **A summary paragraph** at the end of Section 4.1 to provide key takeaways. Similar summary paragraphs have also been added to Sections 4.2, 5.1, and 5.2 to ensure consistency throughout the manuscript.
>
> **For Section 4.2**, we have substantially expanded this section and integrated it into the revised Section 4.2. The updates include:
>   - **Motivation and implications** for each theorem, clarifying their relevance and connection to GAN training dynamics.
>   - **A summary paragraph** at the end of Section 4.2, synthesizing the results and providing explicit conclusions.
>
> **Additional changes regarding the discussion of theoretical results** include:
>
> * **Expanded contextualization of theorems**:
>    Each theorem now includes detailed explanations to clarify its context, motivation, implications, and relationships with other results. Specifically:
>    - **Motivation**: We elaborated on Theorems 4.3, 4.4, and 5.2, explaining their necessity in understanding GAN training dynamics.
>    - **Implications**: Theorems 4.1, 4.2, 4.3, 4.4, and 5.2 now highlight both practical and theoretical implications, particularly their roles in understanding steepness evolution and mode collapse.
>    - **Inter-theorem relationships**: Connections between related results, such as Theorems 4.1 and 4.2 (steepness of 1-dimensional and n-dimensional measure-preserving maps) and Theorems 5.1 and 5.2 (steepness behavior during collapse), are emphasized to form a cohesive framework.
>    - **Practical relevance**: We discussed how Theorems 4.2, 4.4, and 5.2 inform actionable methods for improving GAN training and detecting mode collapse.
>
> * **Summarizing key roles of discriminator gradients and steepness**:
>    We added summary paragraphs at the end of the first two subsections in Section 4 (fitting phase) and Section 5 (collapse phase), synthesizing the roles of discriminator gradients and steepness in these phases. These summaries provide a cohesive understanding of how these metrics evolve and interact.
>
> * **Descriptive titles for theorems**:
>    Concise, descriptive titles have been assigned to each theorem to help readers quickly grasp their core ideas and relevance.
>
> By incorporating these changes, we hope that the revised manuscript provides a clearer and more detailed discussion of the theoretical results while strengthening the connection between theory and practice, thereby addressing your concern.

---

> ### Author Response · Authors · 2025-01-23
> **Thank You and a Point-by-Point Response to Comments (Part 2)**
>
> ## Insufficiently Supported Claims
>
> > First of all, in contradiction with the abstract, introduction and conclusion, the paper does not characterize GAN training into three phases. Such a characterization would necessitate a clear separation between phases and a precise definition of each phase, unlike the high-level description of Table 1. While collapsing is clearly identified in Section 5 and by the early stopping criterion, the other two phases and not properly identifiable. Fitting is especially blurry as explained above, and its separation with refining is never described. In its current state, the paper is more of an analysis of the training dynamics of GANs before and after collapse.
>
> Thank you for pointing this out. We have carefully considered your feedback and have reorganized the manuscript by **merging the previous "fitting" and "refining" phases into a single "fitting" phase**, to avoid potential ambiguity. In the revised Section 4, we focus our analysis on the dynamics of discriminator gradients and steepness during the fitting phase, which precedes the onset of collapse. This restructuring aims to provide a clearer and more cohesive narrative.
>
> As a remark, we observe in Figure 6 that the steepness metric exhibits a sharp initial increase followed by stable oscillations. We hypothesize that this initial rise corresponds to particles transitioning from the noise prior to the modes (previously referred to as the original fitting phase). During this process, the generator requires greater steepness to ensure that particles close to each other can be effectively separated. In the original refining phase, since particles are already near the modes, the steepness stabilizes. However, rigorously demonstrating that these two phases occur sequentially and identifying the transition has proven challenging, which ultimately led us to merge them into a single phase in the revised manuscript.

---

> ### Author Response · Authors · 2025-01-23
> **Thank You and a Point-by-Point Response to Comments (Part 3)**
>
> > The paper overclaims that Algorithm 2 "balances sample quality and diversity" (abstract) when it actually only stops training when collapse occurs as seen in Figures 5 and 6, and that it "offers enhanced sensitivity and reliability in detecting mode collapse compared to existing approaches" (Section 6.2). For the last assertion to hold, the paper should at least explain why monitoring the FID is not sufficient (as it aligns with the proposed criterion) and show whether collapse can already be detected in the GAN losses.
>
> Thank you for pointing this out. We acknowledge that the original wording in the abstract and Section 6.2 may have overstated the claims. To address this concern, we have revised the language to more accurately reflect the purpose of Algorithm 2:
>
> - **Abstract**:
>    We have replaced the phrase "*balances* sample quality and diversity" with "*retains* sample quality and diversity" to clarify that the algorithm primarily aims to prevent collapse by stopping training at an appropriate point. The revised wording reflects the algorithm's role in maintaining both sample quality and diversity up to the point of collapse, rather than actively balancing these aspects throughout training.
>
> - **Section 6.2**:
>    In this section, instead of claiming that our early stopping metric is "offers enhanced sensitivity and reliability in detecting mode collapse compared to existing approaches", we have chosen to present a multi-perspective comparison to highlight the complementary aspects of our metric and existing approaches, especially the FID score in the main text. Below, we address this concern by comparing the two methods across three key aspects:
>
>     * **Sensitivity to mode collapse**:
>        As shown in Figure 6, our metric, steepness, closely aligns with the FID score in detecting mode collapse. Specifically, steepness exhibits a sharp decline during collapse phases, corresponding to a rapid increase in FID scores. Both metrics effectively signal the onset of mode collapse, indicating that steepness can serve as a reliable and sensitive indicator of this transition.
>
>     * **Insight into the training dynamics**:
>       Our metric provides insights into the training dynamics that complements the FID score. Notably, steepness increases during the early epochs of training, corresponding to the phase where FID decreases most rapidly, which reflects the transition from prior noise to the modes of the real data distribution. Steepness then stabilizes and oscillates, which corresponds to the FID score slowly decreasing. This reflects the particles moving closer to the modes and improving the sample quality gradually. Toward the later stages of training, steepness begins to decline sharply, signaling mode collapse, which coincides with a rapid increase in the FID score.
>
>     * **Applicability during training**:
>        The FID score is frequently used for retrospective evaluation, where generator checkpoints are saved periodically, and the FID score is computed post hoc. This process typically requires generating a large number of samples (e.g., 10k or more) and feeding them through a pretrained Inception network, making it computationally intensive. In contrast, our algorithm can be integrated directly into the training process, requiring batch-level gradient computations for both the generator and discriminator. This enables real-time monitoring and early stopping, potentially reducing the frequency of checkpoint saving in resource-intensive training scenarios.
>
> - **GAN training losses**:
>   We have added Appendix G.6 to present the GAN training losses under the same experimental settings as in Figures 5-6. We observe that GAN losses often exhibit divergence after collapse, either escalating to large values or dropping to zero. As discussed in Appendix G.6, while these patterns can provide some insights into training dynamics, their use as early stopping criteria can be challenging due to limited interpretability and sensitivity to hyperparameters. Exploring how GAN losses might complement other metrics for detecting collapse could be a valuable direction for further study.
>
> We hope these revisions address your concern by aligning our claims with the evidence presented in the paper.

---

> ### Author Response · Authors · 2025-01-23
> **Thank You and a Point-by-Point Response to Comments (Part 4)**
>
> > Section 5.1's assumptions are insufficiently grounded, making its conclusion questionable. There is no explanation why $p_g\approx p_{data}$ close to modes. Assumption 5.1 artificially makes the discriminator ill-conditioned without proper justification. The referenced Appendix G does not show such a sharp transition in-between modes for the discriminator.
>
> Thank you for pointing out this concern. We have added an explanation on the assumption that $p_g\approx p_{data}$ to clarify its basis. **The main idea is that it can be understood through the dynamics discussed in Section 4.1.** Specifically, particles near high-density regions are naturally attracted to these regions (Case 2), while overaccumulation of particles triggers redistribution toward alternative modes (Case 3), ensuring balanced coverage of the data distribution. These dynamics collectively support the validity of the assumption near mode centers.
>
> To further address concerns about Assumption 5.1 and its empirical grounding, **we conducted a verification process on MNIST and Fashion MNIST in Appendix G.1** using saved checkpoints prior to mode collapse as the basis for our analysis. Specifically, we randomly sampled $10,000$ points from each dataset and applied the $k$-means algorithm to cluster these points into $100$ clusters, treating the cluster centers as representative modes of the data. We then computed the discriminator values at these cluster centers and took their average as the baseline measurement. To evaluate the impact of perturbations, we introduced random standard Gaussian noise with varying scales to these cluster centers. The maximum perturbation scale was determined according to the $4\sigma/dim$ criterion described in Assumption 5.1, where $\sigma$ represents the average population variance of the labels in the dataset, and $dim$ is the dimension of the standard Gaussian noise. This ensures that the largest expected distance to the cluster centers equals $4\sigma$. By plotting the mean discriminator values as a function of the perturbation scale, **we observe that the discriminator values approximately follow a linear trend near the mode centers, gradually decreasing from $0.5$ to $0$ as the perturbation scale increases**, justifying Assumption 5.1. Please refer to Appendix G.1 for more details.
>
> As for Figure 11 in Appendix G.1, we provide additional explanations regarding the discriminator values and the colormap. For further details, please refer to Appendix G.1.

---

> ### Author Response · Authors · 2025-01-23
> **Thank You and a Point-by-Point Response to Comments (Part 5)**
>
> ## Prior Work Discussion
>
> > The steepness and the identification of mode collapse as the last stage of GAN training are presented as novel in page 1, but the paper lacks sufficient discussion of the literature for this claim to hold. Details are given below. To my understanding, the notion of generator steepness, including its link with GAN trainability and minimal steepness requirements, has already been studied in the literature, with a few variations, by, notably, Odena et al. (2018), Tanielian et al. (2020) and Salmona et al. (2022). Given the potentially strong links with the paper's contributions, a discussion of the paper's novelty w.r.t. the existing literature should be included. Furthermore, the paper undermines pre-existing knowledge of GAN training and collapse in the literature, for example via the assertion "[the paper] challenges the conventional view that mode collapse signifies GAN training failure" (page 2) --- which should be motivated by one or several references. I would like to point out in particular the seminal paper of Brock et al. (2019) documenting and analyzing collapse in the final training stages.
>
> We sincerely thank you for pointing out the relevant literature. We have included a detailed comparison with these works in Section 2: Related Work, and have modified the assertation in the main text. Below, we briefly outline how our work complements and extends the cited studies:
>
> - **Regarding final-stage mode collapse**, we appreciate your reference to the work by Brock et al. (2019), which documents later-stage mode collapse observed when scaling up GAN architectures using BigGAN on high-resolution, large-scale datasets. They highlight that "settings which were stable in previous works become unstable when applied at scale." While we recognize the significance of their contributions, our study complements these findings by demonstrating that similar phenomena can occur in NSGAN trained on relatively smaller datasets. Furthermore, our analysis takes a distinct approach by focusing on the generator’s overall steepness and the $L_2$-norm of the discriminator gradients $\nabla d(x)/d(x)$, providing a complementary perspective to the layer-specific singular value analysis employed in BigGAN. Notably, while Brock et al. (2019) primarily focus on stabilizing large-scale GAN training to enhance performance, our work emphasizes detecting such collapses through quantitative metrics.
>
> - **Regarding related concepts with steepness**, we appreciate your reference to the works of (Odena et al., 2018; Tanielian et al., 2020; Salmona et al., 2022). In terms of definition, they approach the Jacobian of the generator in different ways, including the condition number (Odena et al., 2018) (which is the ratio of its largest to smallest singular value) and the global Lipschitz constant (Tanielian et al., 2020; Salmona et al., 2022) (which may be seen as the global supremum of steepness). In terms of main findings, Odena et al. (2018) focus on empirical investigations of the relationship between the condition number and the GAN's well-being, proposing Jacobian Clamping as a technique to stabilize training. On the other hand, Tanielian et al. (2020) introduce a Jacobian-based truncation method aimed at reducing off-manifold samples and improving precision in the context of disconnected manifolds, while Salmona et al. (2022) derive theoretical bounds showing that a high Lipschitz constant is necessary to fit well-separated multimodal distributions. Our work complements these studies by explicitly deriving the lower bound for multi-modal distributions and employing steepness, which is locally defined, as a theoretical tool to analyze mode dynamics. This local perspective allows us to identify the decline of steepness as a potential indicator of mode collapse and offers complementary insights into the dynamics of mode mixture during GAN training.
>
> As for the assertion "challenges the conventional view that mode collapse signifies GAN training failure", we have modified the wording to: "*mode collapse is **frequently** viewed as an indicator of training failure (Arjovsky et al., 2017; Luo & Yang, 2024), **and the current literature primarily focuses on techniques to prevent or mitigate this phenomenon,***" to provide a clearer and more precise explanation. We hope this revision aligns better with the existing literature and improves the clarity of our discussion.

---

> ### Author Response · Authors · 2025-01-23
> **Thank You and a Point-by-Point Response to Comments (Part 6)**
>
> ## Other Issues
>
> > Algorithm 2 necessitates to know the number of modes of the data distribution and to estimate the generator's steepness. The paper would benefit from a discussion on the feasibility of these choices.
>
> Thank you for highlighting this point. We acknowledge that Algorithm 2 relies on knowing the number of modes in the data distribution and estimating the generator's steepness. Below, we address the feasibility of these requirements:
>
> - **Knowing the number of modes**:
>    * In practice, many datasets used in GAN training (e.g., MNIST, Fashion MNIST, CIFAR-10) have well-defined modes corresponding to distinct classes or clusters in the data distribution. For such datasets, the number of modes is typically known or can be reasonably estimated.
>    * In cases where the number of modes is unknown, clustering techniques (e.g., k-means or Gaussian Mixture Models) can provide a practical approximation of the mode count. While these methods may not always yield perfect accuracy, they offer a reasonable baseline for implementing Algorithm 2 in more general scenarios.
>
> - **Estimating the generator's steepness**:
>    * The steepness of the generator is defined in terms of the norm of the Jacobian of the mapping from the noise space to the data space. This can be computed using automatic differentiation tools in modern deep learning frameworks (e.g., PyTorch). Although the computation may be resource-intensive, it is feasible for low-dimensional noise spaces or in settings with moderate computational budgets.
>    * For more efficient estimation, one could subsample the noise space or use stochastic approximation techniques to estimate steepness over a representative subset of noise vectors.
>
> We have incorporated this discussion in the revised manuscript to clarify the feasibility and address potential concerns regarding these aspects of Algorithm 2.
>
> > It is not clear how Assumption 2.1 (Gaussian smoothing of the data) is related to minibatch training. I believe this assumption can be justified more directly by noticing that data smoothing has now become a standard even beyond diffusion models.
>
> Thank you for your feedback. We acknowledge that the connection between Assumption 2.1 and minibatch training may not have been immediately clear. Based on your suggestion, we have revised the discussion as follows:
>
> *Gaussian smoothing of the data, as applied in assumption 2.1, is a commonly used approach in machine learning to transform discrete datasets into continuous probability distributions. This method enables mathematical analysis, aligns with standard data preprocessing practices, and is widely adopted in generative modeling (Goldfeld et al., 2020; Ho et al., 2020; Song et al., 2021; Karras et al., 2022)...*
>
> > I assume that $\omega$ denotes the discriminator's parameters, but this is not specified in the main paper.
>
> You are correct. We have updated Algorithm 1 to explicitly clarify that $\omega$ and $\theta$ represent the parameters of the discriminator and the generator, respectively.
>
> ----
>
> We sincerely thank you again for your valuable feedback. If you have any additional questions or suggestions, we would be delighted to address them and engage in further discussion.
>
> Best regards,
>
> Paper3757 Authors

---

> ### Comment · Reviewer_AFrb · 2025-01-26
> **Acknowledgement and follow-up**
>
> I would like to thank the authors for their extensive response and revision that addresses many of the raised concerns.
>
> I particularly appreciate the **clear improvements in writing and clarity**, the **added discussion on close related work**, as well as the significant changes to **make claims better formulated or supported**. Merging the first two phases (fitting and refining) into one (fitting) now clearly fits the paper's message and results.
>
> Before providing a recommendation, I would like to pursue the discussion with the authors on **a few remaining issues** -- although less critical than in the first version -- after assessing again the full submission, described below.
>
> ### Contribution of Section 4 (fitting phase)
>
> Overall, Section 4's contributions to describe fitting are **either questionable or not clearly different from the related work**.
>
> **(1.)** **Section 4.1** describes how generated particles are pushed towards modes by the discriminator's gradients on toy examples. While this case study and visualization are interesting, they **do not provide new insights** compared to previous works describing the particle perspective of GANs (Franceschi et al., 2023; Yi et al., 2023). The convergence towards the data distribution is already captured by the fact that the optimal discriminator yields a Wasserstein gradient flow on the particles.
>
> **(2.)** **Results on steepness**, although different from prior work, **mostly convey the same insights** as Tanielian et al. (2020) and Salmona et al. (2022): fitting a multimodal distribution requires large steepness. To my understanding, bounds on steepness or related metrics could be derived from these works as well. I understand when the authors insist that their definition is local, but theoretical results in the section (Theorems 4.1, 4.2 and 4.4) only provide bounds on maximum steepness, similarly to these works as well. The discussion of the related work should be extended in more technical details to highlight the added value of this section.
>
> **(3.)** **Theorem 4.3 and the accompanying discussion on the evolution of steepness lack relevance.** The theorem only tackles data distributions with a single mode: I understand the complexity of the task, but this makes the result unrelated with mode mixture -- especially as the steepness is considered at the mode's center. Furthermore, the conclusions are partially contradicted by the experiments: while steepness does increase during the first epochs, it then slightly but consistently decreases for a longer period e.g. in Figure 5.
>
> ### Minor issues
>
> The revision introduced a few issues (non-exhaustive list below) that would require proofreading.
> - I am not sure the clarity of the added Section 2 is ideal for a new reader. The section mentions several notions described later in the paper. Either the section should be moved, or its content should be scattered in the other relevant sections (i.e. Sections 1, 3, and 4).
> - Appendix J still mentions three phases instead of two.
> - Formatting:
>   - an inline reference to Brock et al. (2019) should be done without parentheses (p. 3);
>   - "where $p\_{\mathrm{data}}$ composed" should be "where $p\_{\mathrm{data}}$ is composed" (p. 8);
>   - the readability of Figures 5, 6, 16, 17 and 18 could be improved (text is very small and color combinations are not ideal, to be checked in grayscale).

---

> > ### Author Response · Authors · 2025-01-30
> > **Thank You and a Point-by-Point Response to Remaining Comments (Part 1)**
> >
> > Dear Reviewer AFrb:
> >
> > We sincerely thank you for your thoughtful reassessment of our work and for your kind recognition of our revisions. We are delighted that you found improvements in writing, the additional discussion on closely related work, and the reformulation of claims to be effective. We also appreciate your positive feedback on merging the fitting and refining phases. Below, we provide point-by-point responses to the remaining issues you raised.
> >
> > ## Contribution of Section 4 (fitting phase)
> >
> > > (1.) Section 4.1 describes how generated particles are pushed towards modes by the discriminator's gradients on toy examples. While this case study and visualization are interesting, they do not provide new insights compared to previous works describing the particle perspective of GANs (Franceschi et al., 2023; Yi et al., 2023). The convergence towards the data distribution is already captured by the fact that the optimal discriminator yields a Wasserstein gradient flow on the particles.
> >
> > Thank you for your thoughtful feedback. We appreciate your observation regarding the connection between optimal discriminator gradients and the Wasserstein gradient flow, as established in prior works (Franceschi et al., 2023; Yi et al., 2023). These studies provide a solid theoretical foundation by demonstrating how an optimal discriminator ensures global convergence of particles toward the data distribution. In our revised manuscript, **we have explicitly restated their relevance in Section 3.1 to emphasize their contributions**.
> >
> > **Our work seeks to complement this theoretical understanding by investigating how configurations of $p_g$ (generator distribution) and $p_{data}$ (data distribution) governs gradient-driven particle dynamics in specific scenarios.** While prior studies focus on global convergence guarantees, they do not explicitly analyze how local mode structures—such as clustered or isolated modes—affect the discriminator’s ability to guide particles. To address this gap, we designed experiments that isolate two distinct regimes of mode configuration:
> >
> > - **Locally Clustered Modes (Cases 1–3):**
> >    Our results demonstrate that discriminator gradients in these cases are pointed toward the nearest modes, with gradient magnitudes modulated by particle density and mode proximity (Figure 2, top row). For instance, in Case 3, intensified gradients around unoccupied modes lead to the redistribution of particles toward underrepresented regions. This localized effect, while consistent with global convergence principles, cannot be fully inferred from the Wasserstein gradient flow framework.
> >
> > - **Globally Separated Modes (Case 4):**
> >    In scenarios with isolated modes, such as Case 4, we observe a significant weakening of gradient magnitudes near distant modes (Figure 2, bottom right). This highlights the practical importance of proper initialization and balanced training to overcome mode separation challenges.
> >
> > By restricting our analysis to the fitting phase and assuming discriminator optimality, we disentangle the geometric effects of mode structure from later-stage collapse dynamics (Section 4). This focused analysis reveals that mode configuration itself can impose bottlenecks on particle movement, as observed in Case 4. These nuances may not be easily captured in end-to-end convergence analyses.
> >
> > In our revised manuscript, we have incorporated these discussions to better acknowledge the contributions of Franceschi et al. (2023) and Yi et al. (2023), while emphasizing our focus on the interplay between mode configurations and particle dynamics. We have also reorganized Section 3.1 by summarizing the configurations of $p_g$ and $p_{data}$ in Table 2 to provide a more compact presentation.

---

> > ### Author Response · Authors · 2025-01-30
> > **Thank You and a Point-by-Point Response to Remaining Comments (Part 2)**
> >
> > > (2.) Results on steepness, although different from prior work, mostly convey the same insights as Tanielian et al. (2020) and Salmona et al. (2022): fitting a multimodal distribution requires large steepness. To my understanding, bounds on steepness or related metrics could be derived from these works as well. I understand when the authors insist that their definition is local, but theoretical results in the section (Theorems 4.1, 4.2 and 4.4) only provide bounds on maximum steepness, similarly to these works as well. The discussion of the related work should be extended in more technical details to highlight the added value of this section.
> >
> > Thank you for your thoughtful feedback. **We have expanded the discussion in the Related Work section (now Section 5) to provide more technical details.** Specifically, our approach differs from Tanielian et al. (2020) and Salmona et al. (2022) in three key aspects:
> >
> > - We employ the density transformation formula as a central tool, directly linking the generator’s push-forward density to the noise distribution. In contrast, Tanielian et al. (2020) and Salmona et al. (2022) rely on the Gaussian isoperimetric inequality to analyze divergence measures.
> >
> > - We analyze the evolution of steepness during training (in updated Theorem 3.3 for a symmetric mixture of Gaussians) and establish a quantitative relationship between steepness and mode mixture severity (Theorem 3.4). This allows us to show that mode mixture severity decreases over time, whereas prior works assume a static generator with a fixed Lipschitz constant and focus on theoretical guarantees under this assumption.
> >
> > - Our localized approach leverages steepness to analyze mode dynamics in specific regions of the data space. This enables us to examine how steepness declines near individual modes during collapse, offering complementary insights into mode collapse dynamics.
> >
> > We hope this expanded discussion clarifies the contributions of our work in relation to existing studies.
> >
> > > (3.) Theorem 4.3 (now Theorem 3.3) and the accompanying discussion on the evolution of steepness lack relevance. The theorem only tackles data distributions with a single mode: I understand the complexity of the task, but this makes the result unrelated with mode mixture -- especially as the steepness is considered at the mode's center. Furthermore, the conclusions are partially contradicted by the experiments: while steepness does increase during the first epochs, it then slightly but consistently decreases for a longer period e.g. in Figure 5.
> >
> > Thank you for your insightful feedback. In response, **we have updated Theorem 4.3 (now Theorem 3.3) by modifying its setting to a symmetric mixture of Gaussians**, which better aligns with the multi-modal nature of the target distribution.
> >
> > **This theorem establishes a continuous-time evolution equation for the steepness $g_t'(0)$ of the generator at $x=0$.** While a closed-form analytical solution for $g_t'(0)$ is not available, the equation provides key qualitative insights: under certain conditions—such as when the third-order derivative $g_t^{(3)}(0)$ is small relative to $g_t'(0)$—the steepness exhibits a **monotonic increase**. While our analysis focuses on the bimodal case, the underlying principle extends to more general multimodal settings, though the corresponding evolution equations become significantly more complex. We believe this generalization warrants a dedicated discussion. This suggests that, at least in this simplified setting, the generator naturally increases its steepness to adapt to the bimodal structure of the data. Please refer to Theorem 3.3 and the preceding discussion for more details.
> >
> > Regarding the observation that steepness slightly but consistently decreases after an initial increase in the experiments (e.g., Figure 5), we acknowledge this discrepancy and believe it may arise due to several factors. First, in practical training, the evolution of steepness can be influenced by minibatch effects, stochastic gradients, and adaptive optimization techniques, leading to deviations from the idealized theoretical trajectory. Additionally, once the generator has sufficiently captured the modes, an excessive increase in steepness may not be beneficial and could be counteracted by stabilization mechanisms inherent in the learning process.
> >
> > We appreciate your comments and hope these modifications improve the clarity and relevance of our analysis.

---

> > ### Author Response · Authors · 2025-01-30
> > **Thank You and a Point-by-Point Response to Remaining Comments (Part 3)**
> >
> > ## Minor Issues
> >
> > > I am not sure the clarity of the added Section 2 is ideal for a new reader. The section mentions several notions described later in the paper. Either the section should be moved, or its content should be scattered in the other relevant sections (i.e. Sections 1, 3, and 4).
> >
> > Thank you for your suggestion. We agree that introducing notions before they are properly explained could be confusing for new readers. In response, **we have moved this section to later in the manuscript, now as Section 5, just before the experiments section**. This ensures that all relevant notions and concepts have been introduced and explained in detail beforehand. We believe this adjustment improves the logical flow and makes the content more accessible to readers unfamiliar with the subject.
> >
> > > Appendix J still mentions three phases instead of two. An inline reference to Brock et al. (2019) should be done without parentheses (p. 3); "where composed" should be "where is composed" (p. 8).
> >
> > Thank you for your comment. We have addressed these points: Appendix J has been updated to reflect two phases instead of three, the inline reference to Brock et al. (2019) has been adjusted as specified (now p. 12), and "where composed" has been removed due to the updates in Theorem 3.3 and the preceding discussion.
> >
> > > The readability of Figures 5, 6, 16, 17 and 18 could be improved (text is very small and color combinations are not ideal, to be checked in grayscale).
> >
> > Thank you for your comment. We have improved the readability of the figures by **increasing the legend font size** and **adjusting the color scheme** according to their luminance to enhance contrast and grayscale readability.
> >
> > ----
> >
> > We sincerely thank you again for your constructive feedback and thoughtful comments. Your insights have greatly contributed to improving the clarity and quality of our manuscript. We hope our responses and the revisions address all your concerns, but we are happy to further clarify or refine any aspect if needed.
> >
> > Best regards,
> >
> > Paper3757 Authors

---

> > > ### Comment · Reviewer_AFrb · 2025-02-03
> > > **Acknowledgement (2)**
> > >
> > > I would like to thank the authors for their follow-up response, which successfully clarified my main remaining concerns. I can now provide a recommendation.
> > >
> > > For the next version of this paper -- after the decision --, I would recommend the authors to address the following points.
> > >
> > > ### New Theorem 3.3
> > >
> > > The assumption in the new Theorem 3.3 that the generator evolves according to the gradient-flow limit is strong, even though understandable. The theorem would benefit from a discussion of this hypothesis, notably what it requires for the generator.
> > >
> > > ### Steepness and related work
> > >
> > > - Now that the related work follows Sections 3 and 4, it would be best to reference it when the authors introduce definitions and results on steepness for an easier contextualization when reading the paper linearly.
> > > - I still think the claim on steepness locality should be toned down, as Tanielian et al. (2020) also rely on local steepness for their rejection method.
> > >
> > > ### Formatting
> > >
> > > - Equations should be numbered.
> > > - "For case 1", p. 5, should be "Case 1".
> > > - Font size in figures remain small, especially in Figure 5. Since there is no strong page limit, it would be best for readability to increase it.

---

> > > > ### Author Response · Authors · 2025-02-13
> > > > **Thank You and Acknowledgment of Your Suggestions**
> > > >
> > > > Dear Reviewer AFrb,
> > > >
> > > > We sincerely appreciate your thoughtful follow-up and your valuable feedback throughout the review process. We are pleased that our clarifications addressed your main concerns. **Regarding your suggestions for the next version of the paper, we will carefully take them into account and implement the corresponding improvements.** Thank you again for your insightful comments and for your support in helping us improve the manuscript.
> > > >
> > > > Best regards,
> > > >
> > > > Paper3757 Authors

---

### Review · Reviewer_JU7e · 2025-01-13

**Summary Of Contributions:**

- This paper proposes to study the training dynamics of GAN models using a set of gradient based tools.
- They first describe these tools: 1) the discriminator gradient, i.e. the gradient of the discriminator with respect to samples, as well as 2) steepness; i.e. the spectral norm of the Jacobian of the generator's gradient with respect to the samples.
- The suggest that these tools can help us to better identify different regimes of GAN training, which they describe as fitting, refining, and collapse.
- The authors next present a series of theoretical results and simplified data analysis settings which are nominally specific to each phase of GAN training:
	1. For the first fitting phase, they describe how discriminator gradients would encourage movement of data samples under different configurations of the true and simulated data distributions. The novelty of these results lies in the two toy configurations of true/generated data distributions that they present, as well as a qualitative analysis of discriminator gradient vector fields under these conditions.
	2. For the second refining phase, the authors:
		1. provide a theoretical analysis of steepness, which measures the ability of a generator to map similar points in the prior to different samples in the data space.
		2. They next provide a lower bound on steepness under simplifying assumptions on the true data distribution, as well as analyses of how generator steepness evolves with time.
		3. Finally, they empirically correlate simple generators having higher steepness with less severe mode mixture on a gaussian mixture dataset, and support this empirical evidence with theoretical results in one dimension.
	3. In the third collapsing phase, the authors propose two characteristics of discriminator gradients and steepness which they suggest are indicative of mode collapse.
		1. They show that under simplified conditions and discriminators of a certain form, samples which are close to boundaries of data modes will experience large discriminator gradients which can trigger the onset of mode collapse. They suggest that monitoring the norm of discriminator gradients is therefore a useful metric to detect the onset of mode collapse.
		2. They show that under these same simplified conditions and discriminators, the generator will map samples which are close to a true data sample to a smaller neighborhood of that sample.
		3. Finally, the authors suggest an early stopping metric, montoring discriminator gradients and steepness evaluated on generated samples to monitor for the onset of mode collapse.
- Finally, the authors present experimental results to corroborate their findings, as follows:
	- The authors evaluate image generation using a GAN model trained on MNIST (as well as Fashion MNIST in the appendix), and consider the probability of sample classification (as evaluated by a separate network $q(x)$ throughout the course of network training. They suggest that the dynamics of how generated samples are classified reflects the time course of the fitting and refining organization.
	- The authors then present empirical results for their early stopping algorithm, showing that the combination of discriminator gradients and steepness appears to identify a threshold before which generated samples are diverse and realistic, whereas samples generated after the threshold appear to be generated from collapsed modes. They also present the FID score (as well as the duality gap/perturbed duality gap in the appendix) and show that the metrics that they suggest to detect mode collapse appear to correlate well with these established metrics.
- Finally, the authors study the effect of adding noise to the discriminator, showing that doing so appears to mitigate fluctuations in the norm of the discriminator gradient.
- In the appendix, the authors provide an extensive review of the literature around GAN models, as well as proofs for theoretical results and additional experimental results.

**Audience:**

Yes

**Broader Impact Concerns:**

I do not have concerns about the ethical implications of this work.

**Claims And Evidence:**

Yes

**Requested Changes:**

I have organized this section to mirror the weaknesses described above.
## Support for claims
- I believe that more evidence is required to support the claim that fitting and refining as described in the paper currently are sequential phases of GAN training. In particular, understanding how we might identify these phases quantitatively and measure the transition between them would be critical. Addressing this concern would be crucial for me to recommend this paper for acceptance in its current form.
-  If the authors will make the claim that their early stopping metric is more reliable and sensitive than existing approaches, I think this claim requires a more thorough explanation, especially in comparison to the FID score as is presented in the main text. What aspects of selecting an early stopping criterion based on FID score would be less reliable/sensitive than the algorithm presented? Addressing this concern would be crucial for me to recommend this paper for acceptance in its current form.
	-  Additionally, given the results presented Figure 7, it would be useful to understand if the early stopping algorithm still works well on noised GANs. Is it the case that collapse is avoided altogether? Or rather, is there a more subtle collapse that can still be detected. Addressing this concern would strengthen this work.
-  It would be useful to have more illustrative examples to study mode collapse in real datasets. For example, is it possible to track a small number of particles through training, and measure steepness/discriminator gradients on it in order to verify that the mechanistic dynamics described on toy datasets do indeed generalize to the behavior of samples in real data? Addressing this concern would strengthen the paper.
## Paper organization
- I believe this paper is significant for its study of mode collapse, without necessarily describing separate fitting and refining stages of GAN training. If the authors were to reorganize sections 3 and 4 to be studies of discriminator gradients and steepness, and focus their study on the analysis of the transition to mode collapse, I believe it would be much easier to support the claims relating to mode collapse with the existing results. Addressing this concern (which is the same as the first "support for claims" concern) would be critical for me to recommend this paper for acceptance.
- A discussion of each theoretical result presented, as well as descriptions of how they do/could inform practical algorithms or tools to study GAN training would greatly strengthen sections 3-5 in my opinion.
## Clarity
I believe that clarification on the points below are all critical for me to recommend this paper for acceptance, unless otherwise indicated.
- What do the colors represent in Figure 2? I initially thought this was the density of $p_{data}$, but it should be uniform across all four modes in all presented cases. I then thought it was the norm of the vectors (which is very difficult to see- please consider subsampling) but the colors would then be inversely correlated to the scale provided.
- Regarding the early stopping algorithm: In Figure 5, I assume the epoch indicated by the horizontal line in this Figure (also Figure 15) is the epoch where the early stopping algorithm would halt training. Are the samples indicated in red boxes those which are selected to optimize the GAN cost? If not, can you show these samples instead, as well as the samples generated immediately prior to the epoch where your early stopping algorithm is triggered?
- In Figure 7, epoch 54, it appears that the histogram of discriminator gradient norms for noised GAN is entirely to the left of the 90% threshold. Where is the mass of the histogram for the noised GAN? I am also unable to read the axis labels for this figure.
  - I also have concerns about the interpretation of this figure. My understanding was that mode collapse is indicated by **increases** in the norm of the discriminator gradient, whereas here it seems that mode collapse is indicated by a small gradient norm. Please elaborate on these results.

**Strengths And Weaknesses:**

## Strengths
- This paper presents some compelling and interesting findings about the training dynamics of GANs (in particular, around the phenomena of mode collapse and mode mixture) which are to my understanding, novel. The gradient based tools described in this work appear to be well suited to detect mode collapse, and have already proved fruitful in identifying successful periods of GAN training prior to collapse.
- It also contains a large quantity of theoretical and experimental results, some of which I believe could be of interest to the broad machine learning community.
- The methods described here could certainly lead to many interesting follow up works to better understand the dynamics of training GAN model.

## Weaknesses
My main concerns for the paper are regarding support for the main claims of the paper. Many of these concerns are related to weaknesses in paper organization and clarity. I have categorized my concerns below.

### Support for claims.
- One of the main claims of this paper is that GAN training can be understood as three sequential phases of training, designated as fitting, refining, and collapsing. I found that there is very compelling evidence to support the claim that mode collapse often follows training that we would otherwise deem successful (as verified by the early stopping algorithm), but very little evidence to support fitting and refining as independent and sequential phases of training.
	- The empirical evidence to support the existence of a fitting and refining phase is presented in Figures 1, 4, 12 and 13. I do not find strong evidence in these figures to suggest that fitting and refining must occur sequentially, as opposed to being concurrent features of what we might call "successful GAN training", as the idea that these states are sequential is often suggested based on qualitative descriptions of the data.
	- Furthermore, it is not clear to me what quantitative confirmation of the fitting and refining stages would even consist of. Sections 3 and 4 are titled as fitting and refining, but describe the discriminator and generator gradient tools respectively, which are then applied to all stages of GAN training. Are there clear features of fitting vs. refining which can be detected with the gradient based tools described here? If not, how can we evaluate the claim that these stages exist in real data? I have provided further detail for these concerns in the paper organization section below.
- It is not clear how to understand the performance of the early stopping algorithm in comparison with other evaluation metrics for GANs, including those presented here (i.e. FID score and duality gap). Based on the presented results, the early stopping algorithm presented appears to align closely with FID score, but I do not find evidence that "The combined use of
our two proposed metrics offers enhanced sensitivity and reliability in detecting mode collapse compared to
existing approaches", as stated on page 12 of the main text. More careful quantitative results or explanation are necessary in order to support this claim.
- Many of the theoretical results in this paper are concerned with detailed, mechanistic descriptions of particle (e.g. generated sample) evolution over the course of GAN training. For example, the description of mode collapse in section 5 suggests that during collapse individual data modes are gradually vacated of samples, as 1) samples around the boundary of single mode are ejected and attracted towards other modes and 2) sample closer to the center of a data mode become more similar. While these claims may be difficult to justify in real world datasets, the empirical results (while broadly consistent with the theoretical/toy models), do not offer conclusive evidence in support of many aspects of the mechanistic theory.

## Paper Organization.
I believe there should be a correspondence between the section titles describing the three putative phases of GAN training (3-5), and the corresponding results presented.
- Section 3 is labeled as fitting, which I understood to be the first phase when particles are putatively pushed from initialization across the entire support of the true data. The analysis presented within this section offers an analysis which features discriminator gradients and does not seem scoped to this first fitting phase. What features of the fitting phase specifically are we meant to understand from this section? Are there features of discriminator gradients which would identify this fitting phase, as distinct from a refining phase in real data?
- Section 4 is labeled as refining, which I understand to be the second phase of training when the primary focus of GAN training is to reduce mode mixture. While steepness appears to be well suited as a technique to identify refining, it is not clear to me which aspects of steepness would help us to identify the transition from the fitting to the refining stage of training.

There are also several theoretical results which are presented on toy datasets, without a great deal discussion of how they generalize to real data. It would be useful to understand if and how theoretical results presented inform the particular early stopping algorithm described, or other potential applications/avenues of study in the future.

## Clarity.
I think this paper could also be greatly improved with greater clarity in the description of results and figures. I have made my specific suggestions in the requested changes section below.

---

> ### Author Response · Authors · 2025-01-23
> **Thank You and a Point-by-Point Response to Comments (Part 1)**
>
> Dear Reviewer JU7e,
>
> We sincerely thank you for your thoughtful feedback and valuable suggestions. Your comments have provided clear guidance to improve the quality and clarity of our work. We deeply appreciate the time and effort you have devoted to reviewing our paper. Below, we address your concerns point by point.
>
> ## Support for claims
>
> > I believe that more evidence is required to support the claim that fitting and refining as described in the paper currently are sequential phases of GAN training. In particular, understanding how we might identify these phases quantitatively and measure the transition between them would be critical. Addressing this concern would be crucial for me to recommend this paper for acceptance in its current form.
>
> Thank you for your insightful feedback. **We have adopted your suggestion and merged the first two phases, "fitting" and "refining," into a single phase, now referred to as the "fitting phase."** Detailed discussions regarding this change can be found in the responses below (please refer to the first response in the Paper Organization section).
>
> As a remark, we observe in Figure 6 that the steepness metric exhibits a sharp initial increase followed by stable oscillations. We hypothesize that this initial rise corresponds to particles moving from the noise prior to the modes (previously defined as the original fitting phase). During this process, the generator requires greater steepness to separate particles that are close to each other. In the original refining phase, since particles are already near the modes, the steepness stabilizes. Nevertheless, rigorously demonstrating that these two phases occur sequentially remains a challenge, which led us to merge them into one.
>
> > If the authors will make the claim that their early stopping metric is more reliable and sensitive than existing approaches, I think this claim requires a more thorough explanation, especially in comparison to the FID score as is presented in the main text. What aspects of selecting an early stopping criterion based on FID score would be less reliable/sensitive than the algorithm presented? Addressing this concern would be crucial for me to recommend this paper for acceptance in its current form.
>
> Thank you for highlighting this important concern. Instead of claiming that our early stopping metric is "more reliable and sensitive" than existing approaches like the FID score, we have chosen to present a multi-perspective comparison to highlight the complementary aspects of our metric and the FID score. Below, we address this concern by comparing the two methods across three key aspects:
>
> - **Sensitivity to mode collapse**:
>    * As shown in Figure 6, our metric, steepness, closely aligns with the FID score in detecting mode collapse. Specifically, steepness exhibits a sharp decline during collapse phases, corresponding to a rapid increase in FID scores. Both metrics effectively signal the onset of mode collapse, indicating that steepness can serve as a reliable and sensitive indicator of this transition.
>
> - **Insight into the training dynamics**:
>    * Our metric provides insights into the training dynamics that complements the FID score. Notably, steepness increases during the early epochs of training, corresponding to the phase where FID decreases most rapidly, which reflects the transition from prior noise to the modes of the real data distribution. Steepness then stabilizes and oscillates, which corresponds to the FID score slowly decreasing. This reflects the particles moving closer to the modes and improving the sample quality gradually. Toward the later stages of training, steepness begins to decline sharply, signaling mode collapse, which coincides with a rapid increase in the FID score.
>
> - **Applicability during training**:
>    * The FID score is frequently used for retrospective evaluation, where generator checkpoints are saved periodically, and the FID score is computed post hoc. This process typically requires generating a large number of samples (e.g., 10k or more) and feeding them through a pretrained Inception network, making it computationally intensive. In contrast, our algorithm can be integrated directly into the training process, requiring batch-level gradient computations for both the generator and discriminator. This enables real-time monitoring and early stopping, potentially reducing the frequency of checkpoint saving in resource-intensive training scenarios.
>
> In summary, while the FID score remains a widely used and effective measure for evaluation of GAN performance, our proposed metric offers a reliable and complementary perspective. It is sensitive to collapse phases, provides interpretability during training, and computationally efficient. We believe that these attributes make our metric a practical tool for real-time training monitoring and intervention.

---

> ### Author Response · Authors · 2025-01-23
> **Thank You and a Point-by-Point Response to Comments (Part 2)**
>
> > Additionally, given the results presented Figure 7, it would be useful to understand if the early stopping algorithm still works well on noised GANs. Is it the case that collapse is avoided altogether? Or rather, is there a more subtle collapse that can still be detected. Addressing this concern would strengthen this work.
>
> Thank you for your comment. From our experiments, we observe that **collapse is avoided altogether in the noised GAN**. Specifically, we trained the noised GAN for 1000 epochs, and no signs of collapse were detected throughout the training process. This aligns with the established understanding that adding noise to the discriminator helps stabilize training by mitigating extreme gradient magnitudes near mode boundaries, which are a primary contributor to collapse.
>
> **Regarding the early stopping algorithm, it still identifies a point in the training process for the noised GAN where sample quality and diversity are well-retained**. However, we note that in the noised GAN, we do not observe significant sample deterioration or other signs of subtle collapse even after this identified point. This suggests that while the early stopping algorithm remains effective, its role in the noised setting may be more about preserving computational efficiency rather than strictly avoiding collapse.
>
> A possible explanation for this behavior is that the added noise reduces the discriminator's capacity to provide overly precise gradients, which in turn prevents the discriminator from overfitting to specific modes. This stabilization mechanism ensures that the generator continues to explore and cover the data distribution effectively, even in later stages of training. As a result, even subtle collapses do not seem to occur.
>
> We have include these observations and discussions in the revised manuscript to address this concern and highlight the broader applicability of our method in both noised and noise-free GAN settings.
>
> > It would be useful to have more illustrative examples to study mode collapse in real datasets. For example, is it possible to track a small number of particles through training, and measure steepness/discriminator gradients on it in order to verify that the mechanistic dynamics described on toy datasets do indeed generalize to the behavior of samples in real data? Addressing this concern would strengthen the paper.
>
> Thank you for this insightful suggestion. We agree that incorporating illustrative examples from real datasets can provide valuable insights into mode collapse and help validate the generalizability of our theoretical findings.
>
> To address this concern, **we have included an illustration in Appendix G.3 where we use the CIFAR-10 dataset to track the dynamics of a fixed set of particles throughout training** (i.e., the generated images from a fixed set of noises using generators with different epochs). Specifically, we visualize these particles across epochs and compute the steepness $\|J_g(x)\|_2$ and the discriminator gradient norm $\|\nabla d(x)/d(x)\|_2$. This allows us to observe how particles evolve from noisy states to align with modes and eventually experience collapse, consistent with our observations in toy datasets. As for the metrics, $\|J_g(x)\|_2$ shows an initial increase, likely due to the generator aligning the noise prior with the data modes, before stabilizing and eventually dropping sharply during collapse. On the other hand, $\|\nabla d(x)/d(x)\|_2$ exhibits oscillatory behavior throughout training but spikes significantly at the collapse point, highlighting its sensitivity to this phenomenon. Please refer to Appendix G.3 for more details.
>
> These results demonstrate that the mechanistic dynamics described in the toy dataset extend to real datasets, providing additional empirical support for our theoretical analysis.

---

> ### Author Response · Authors · 2025-01-23
> **Thank You and a Point-by-Point Response to Comments (Part 3)**
>
> ## Paper organization
>
> > I believe this paper is significant for its study of mode collapse, without necessarily describing separate fitting and refining stages of GAN training. If the authors were to reorganize sections 3 and 4 to be studies of discriminator gradients and steepness, and focus their study on the analysis of the transition to mode collapse, I believe it would be much easier to support the claims relating to mode collapse with the existing results. Addressing this concern (which is the same as the first "support for claims" concern) would be critical for me to recommend this paper for acceptance.
>
> We appreciate your insightful comments and thoughtful suggestions regarding the organization of Sections 3 and 4. In response, we have reorganized the manuscript by **merging the previous "fitting" and "refining" phases into a single "fitting" phase** to streamline the discussion. In the revised Section 4, we focus our analysis on the dynamics of discriminator gradients and steepness during the fitting phase, prior to the onset of collapse. We intentionally present the dynamics of the discriminator and generator together, rather than in separate sections, to maintain consistency with our analysis of the collapse phase, where the interaction between these components is similarly emphasized.
>
> > A discussion of each theoretical result presented, as well as descriptions of how they do/could inform practical algorithms or tools to study GAN training would greatly strengthen sections 3-5 in my opinion.
>
> We appreciate your suggestion to enhance Sections 3-5 (now revised as Sections 4 and 5) by providing detailed discussions of the theoretical results and their implications for practical algorithms or tools to study GAN training. In response, we have made several revisions to strengthen these sections:
>
> * **Expanded contextualization of theorems**:
>    For each theorem, we have added detailed explanations to clarify their context, motivation, implications, and connections with other results. Specifically:
>    - **Motivation**: We elaborated on the motivation for Theorems 4.3, 4.4, and 5.2, explaining their necessity and relevance to understanding GAN training dynamics.
>    - **Implications**: For Theorems 4.1, 4.2, 4.3, 4.4, and 5.2, we highlighted their practical and theoretical implications, particularly how they inform steepness evolution and mode collapse.
>    - **Inter-theorem relationships**: We emphasized the connections between related results, such as Theorems 4.1 and 4.2 (steepness of 1-dimensional and n-dimensional measure-preserving maps) and Theorems 5.1 and 5.2 (steepness behavior during collapse), demonstrating how these results collectively form a cohesive framework.
>    - **Practical relevance**: We discussed how specific theorems, particularly Theorems 4.2, 4.4, and 5.2, suggest existing/actionable methods for improving GAN training and detecting mode collapse.
>
> * **Summarizing key roles of discriminator gradients and steepness**:
>    At the end of the first two subsections in Section 4 (fitting phase) and Section 5 (collapse phase), we included summary paragraphs synthesizing the roles of discriminator gradients and steepness in these phases. These summaries provide a comprehensive understanding of how these metrics evolve and interact.
>
> * **Descriptive titles for theorems**:
>    To improve readability and accessibility, we assigned concise, descriptive titles to each theorem, helping readers quickly grasp their core ideas and relevance.
>
> By incorporating these changes, we hope that the revised manuscript provides a clearer and more detailed discussion of the theoretical results while strengthening the connection between theory and practice, thereby addressing your concern.

---

> ### Author Response · Authors · 2025-01-23
> **Thank You and a Point-by-Point Response to Comments (Part 4)**
>
> ## Clarity
>
> > What do the colors represent in Figure 2? I initially thought this was the density of $p_{data}$, but it should be uniform across all four modes in all presented cases. I then thought it was the norm of the vectors (which is very difficult to see --- please consider subsampling) but the colors would then be inversely correlated to the scale provided.
>
> We apologize for any confusion. The colors (and the colorbars) in Figure 2 represent the values of the optimal discriminator, $d_*(x)$. Regarding the difficulty in seeing the vectors, we acknowledge that they may be hard to discern, especially near the centers of the modes. This aligns with our discussion in the main text and Theorem C.1 in the appendix, where we noted that particles near the mode centers tend to remain stationary. To improve clarity, we have resampled the vector fields and included zoomed-in views around the bottom left mode (we have used a different colormap to enhance the visual contrast, so the colors may not match the original plots). Please refer to the updated Figure 2 for these details.
>
> > Regarding the early stopping algorithm: In Figure 5, I assume the epoch indicated by the horizontal line in this Figure (also Figure 15) is the epoch where the early stopping algorithm would halt training. Are the samples indicated in red boxes those which are selected to optimize the GAN cost? If not, can you show these samples instead, as well as the samples generated immediately prior to the epoch where your early stopping algorithm is triggered?
>
> Yes, you are correct. **The epoch indicated by the horizontal line in Figure 5 is indeed the epoch where the early stopping algorithm would halt training**.
>
> We would like to clarify that the horizontal line was originally intended to represent the threshold of $\|\nabla d(x) / d(x)\|_2$. Since our early stopping algorithm consists of two metrics—the proportional drop in steepness and the absolute threshold for $\|\nabla d(x) / d(x)\|_2$—a single horizontal line cannot fully capture the decision criteria. Therefore, in the revised version of Figure 5, we now use **a vertical line** to indicate the specific epoch where the early stopping algorithm halts training.
>
> Additionally, as suggested by another reviewer, we have updated Figure 5 to include experiments on more modern GAN architectures that achieve lower FID scores, providing a stronger validation of our approach.
>
> **Regarding the samples in red boxes, they indeed correspond to those selected to optimize the GAN cost in practice.** These samples are presented because this aligns with the common practice of periodically saving model checkpoints and selecting the best-performing one.
>
> In response to your suggestion, we have also included the generated samples immediately prior to the epoch where the early stopping algorithm is triggered. These samples are now shown as the last image in each column for comparison.
>
> > In Figure 7, epoch 54, it appears that the histogram of discriminator gradient norms for noised GAN is entirely to the left of the 90% threshold. Where is the mass of the histogram for the noised GAN? I am also unable to read the axis labels for this figure.
>
> Thank you for pointing this out. Please allow us to clarify the details of Figure 7 at epoch 54.
>
> In Figure 7, the x-axis represents the **logarithmic scale** of $\|\nabla d(x) / d(x)\|_2$, with tick marks at $10^1$, $10^2$, $10^3$, $10^4$. This captures the magnitude of the discriminator gradient norms across samples. The y-axis represents the **density**, which is normalized such that the total area under each histogram sums to 1.
>
> From the figure, the majority of the mass for the noised GAN histogram is concentrated to the left of the 90% threshold, indicating that most samples have relatively small discriminator gradient norms in this scenario. However, there are **outliers** in the region to the right of the 90% threshold. These outliers are not visually prominent because they span a compressed range on the logarithmic x-axis. Despite this, they correspond to large x-axis values ($10^4$), which means that their contribution to the total area under the curve is significant. Combined, these outliers account for 10% of the total area.
>
> In conclusion, the use of a logarithmic x-axis compresses the right tail of the histogram, making the outliers appear visually negligible. This can create the impression that the histogram mass is entirely to the left of the 90% threshold. However, quantitatively, the total area to the right of the threshold is indeed 10%.
>
> We sincerely appreciate your feedback and have added clarifications in the main text to avoid potential confusion in interpreting the figure.

---

> ### Author Response · Authors · 2025-01-23
> **Thank You and a Point-by-Point Response to Comments (Part 5)**
>
> > I also have concerns about the interpretation of this figure. My understanding was that mode collapse is indicated by increases in the norm of the discriminator gradient, whereas here it seems that mode collapse is indicated by a small gradient norm. Please elaborate on these results.
>
> Thank you for pointing this out. This phenomenon arises from our evaluation method, which is performed on the last batch of each epoch, as well as the images of the generated samples.
>
> The $54$th epoch represents the **aftermath of collapse**, where the generator has already collapsed into a limited mode. The discriminator's gradient norm in this context may deviate from what might be anticipated during at the beginning of the collapse phase.
>
> Prior to the $54$th epoch, during the **lead-up to collapse**, the discriminator gradient norm is expected to be large according to the algorithm. This behavior is consistent with our observations, such as in the $52$th epoch, where the discriminator gradient norms are large.
>
> We appreciate your insightful comment and have emphasized that "at the end of the $54$th epoch, the noise-free model collapse ..." in the main text to clarify this point.
>
> ----
>
> We sincerely thank you again for your valuable feedback. If you have any additional questions or suggestions, we would be delighted to address them and engage in further discussion.
>
> Best regards,
>
> Paper3757 Authors

---

> > ### Comment · Reviewer_JU7e · 2025-01-28
> > **Thank you**
> >
> > I believe the work has been greatly improved by the new organization, additional clarifying structure, and experimental results. I have some remaining comments that I have described below.
> >
> > ## Section 2
> > - I appreciate the discussion of closely related work in section 2. However, given the discussion of technical concepts introduced later in this paper, I believe it would be preferable to defer the inclusion of this section to after the current section 3 (or later).
> > ## Section 3
> > - I appreciate the new organization of section 3, and I believe it's much clearer how these two gradient based tools relate to one another.
> > ## Section 4
> > - A lot of space in this section is spent on describing phenomena in a toy model in Figure 4. In general this is fine, as it seems useful to build intuition on what successful GAN training may look like through the lens of these gradient based tools. However, one important point here is that there is no part of Figure 2, Case 3 which would suggest redistribution of particles, as is suggested in the main text here and later in section 5. Namely, the critical point at which particles experiment large-magnitude gradients which would trigger propulsion away to another mode are not apparent, and it is not clear how the vector field intensity would relate to this described phenomena.
> > 	- A suggestion to  replacing Figure 2 with a toy example which is capable of demonstrating the redistribution which is referenced in 5.1.
> > ## Section 5
> > - I think the references back to Figure 2 in section 5.1, and the specific cases presented there help to tie the paper together.  Please make sure to reference the figure when you do so.
> > 	- However, once again, I will emphasize that it is unclear how Case 3 suggests redistribution mechanisms.  From the figure it is difficult to discern how anything about Case 3 suggests disproportionately large gradients near the boundaries of the real data modes. See my suggestion above.
> > - In general sections 4 and 5 are much clearer, and I appreciate the narrative context around the proofs.
> >
> > ## Section 6:
> > - While the results in Figure 4 appear technically correct, it is difficult to understand what they are meant to convey, given that the two phases of fitting and refining have been collapsed into 1. If there is some interesting correspondence between the structure presented here and the discriminator gradients/steepness presented in figure 5, panel 2, that would be good to know. Otherwise "verifying the existence of the fitting phase" seems equivalent to showing that NSGANs can be fit on MNIST.
> > - The discussion of FID score thoroghly addresses my previous concerns.
> > - I found the particle based experiments very interesting, and I appreciate the additional discussions around discussion of the noised GAN results.

---

> > > ### Author Response · Authors · 2025-01-30
> > > **Thank You and a Point-by-Point Response to Remaining Comments (Part 1)**
> > >
> > > Dear Reviewer JU7e:
> > >
> > > We sincerely thank you for your thoughtful reassessment of our work and for your kind recognition of our revisions. We are delighted that you found improvements in organization, additional clarifying structure, and experimental results. Below, we provide point-by-point responses to the remaining issues you raised.
> > >
> > > ## Section 2 (Now Section 5)
> > >
> > > > I appreciate the discussion of closely related work in section 2. However, given the discussion of technical concepts introduced later in this paper, I believe it would be preferable to defer the inclusion of this section to after the current section 3 (or later).
> > >
> > > Thank you for your suggestion. In response, **we have moved this related work section to later in the manuscript, now as Section 5, just before the experiments section**. This ensures that all relevant notions and concepts have been introduced and explained in detail beforehand.
> > >
> > > ## Section 4 (Now Section 3)
> > >
> > > > A lot of space in this section is spent on describing phenomena in a toy model in Figure 4. In general this is fine, as it seems useful to build intuition on what successful GAN training may look like through the lens of these gradient based tools. However, one important point here is that there is no part of Figure 2, Case 3 which would suggest redistribution of particles, as is suggested in the main text here and later in section 5. Namely, the critical point at which particles experiment large-magnitude gradients which would trigger propulsion away to another mode are not apparent, and it is not clear how the vector field intensity would relate to this described phenomena. A suggestion to replacing Figure 2 with a toy example which is capable of demonstrating the redistribution which is referenced in 5.1.
> > >
> > > Thank you for pointing this out and for the thoughtful feedback. We apologize for any confusion caused. In Figure 2, Case 3, **we are specifically discussing the behavior of GANs *during the fitting phase***, where we assume an optimal discriminator (as opposed to the collapse phase, which operates under Assumption 4.1).
> > >
> > > The purpose of Case 3 is to demonstrate that even if the generated particles do not initially cover all modes, the vector field intensity can influence the particles to eventually cover all modes during the fitting phase. It is important to note that Case 3 represents a snapshot in time—after an update step, the vector field will adapt and change accordingly.
> > >
> > > Regarding the collapse phase and the redistribution of particles, we acknowledge the need for further clarification. We address this directly in the following response, with an additional illustration to better support the discussion.
> > >
> > > ## Section 5 (Now Section 4)
> > >
> > > > I think the references back to Figure 2 in section 5.1, and the specific cases presented there help to tie the paper together. Please make sure to reference the figure when you do so. However, once again, I will emphasize that it is unclear how Case 3 suggests redistribution mechanisms. From the figure it is difficult to discern how anything about Case 3 suggests disproportionately large gradients near the boundaries of the real data modes. See my suggestion above.
> > >
> > > Thank you for your suggestions. As mentioned in our previous response, **we have added an illustration in the last part of Appendix G.1, Figure 13, that aims to demonstrate the redistribution mechanism**. Additionally, we have included a hyperlink to this section in the paragraph titled "Discriminator gradients near mode boundaries" (Section 4.1) for better accessibility.
> > >
> > > In Figure 13, we illustrate that for particles near the mode boundaries, the discriminator gradients are disproportionately large, causing the vectors to cross the centers of the modes and reach the opposite side. This gives the impression that the vector field points outward from the mode centers. To clarify this further, we have included zoomed-in visualizations in the appendix. These figures reveal that the vector field is not actually pointing outward but rather appears so due to the large magnitude of the gradients.
> > >
> > > We hope this addition clarifies the redistribution mechanism in Case 3.

---

> > > ### Author Response · Authors · 2025-01-30
> > > **Thank You and a Point-by-Point Response to Remaining Comments (Part 2)**
> > >
> > > ## Section 6:
> > >
> > > > While the results in Figure 4 appear technically correct, it is difficult to understand what they are meant to convey, given that the two phases of fitting and refining have been collapsed into 1. If there is some interesting correspondence between the structure presented here and the discriminator gradients/steepness presented in figure 5, panel 2, that would be good to know. Otherwise "verifying the existence of the fitting phase" seems equivalent to showing that NSGANs can be fit on MNIST.
> > >
> > > Thank you for pointing this out. We have added a discussion at the end of Section 6.1 to address the correspondence between the structure presented in Figure 4 and the discriminator gradients/steepness shown in Figure 5, panel 2. Specifically, we now highlight how the evolution of the heatmap structure links with the trends in the discriminator gradients and steepness.
> > >
> > > For your convenience, we provide the relevant snippet here: "*At the stage when more entries appear in the heatmap, the discriminator gradient's magnitude remains relatively small, while its steepness increases rapidly... As training continues and off-diagonal entries begin to decrease in magnitude, the steepness stabilizes, and the discriminator gradient starts to oscillate...*"
> > >
> > > ----
> > >
> > > We sincerely thank you again for your constructive feedback and thoughtful comments. Your insights have greatly contributed to improving the clarity and quality of our manuscript. We hope our responses and the revisions address all your concerns, but we are happy to further clarify or refine any aspect if needed.
> > >
> > > Best regards,
> > >
> > > Paper3757 Authors

---

> > > > ### Comment · Reviewer_JU7e · 2025-02-03
> > > > **Thank you**
> > > >
> > > > I thank the authors for their responses to my comments. I have reviewed the changes described above, and I will update my recommendation to reflect the revisions made.

---

### Author Response · Authors · 2025-01-23
**Changes Since Last Submission --- Major Changes Grouped by Topic**

**Below, we present the major changes *grouped by topic*, as a complementary to section-specific changes.**

- **Merged Phases**
  * The original "fitting" and "refining" phases have been merged into a single "fitting" phase (Section 4: The Fitting Phase: Gradient Dynamics).

- **Theoretical Modifications**
  * **Expanded contextualization of theorems**:
    Added detailed explanations for the motivation, implications, and interrelationships of Theorems 4.1-4.4 and 5.1-5.2, emphasizing their practical and theoretical relevance. Discussed actionable insights derived from Theorems 4.2, 4.4, and 5.2, highlighting their implications for improving GAN training and detecting mode collapse.
  * **Summarized key roles**:
    Included summary paragraphs in Sections 4 and 5 to synthesize the roles of discriminator gradients and steepness during the fitting and collapse phases.
  * **Descriptive theorem titles**:
    Assigned concise and descriptive titles to all theorems to improve readability and accessibility.

- **Related Work**
  * Added a discussion on two key aspects: (1) The phenomenon of final-stage mode collapse in GAN training (Brock et al., 2019). (2) Related concepts regarding generator steepness (Odena et al., 2018; Tanielian et al., 2020; Salmona et al., 2022) (Section 2: Related Work).

- **Experimental Validation**
  * **Improved GAN architectures**:
    Conducted additional experiments on modern GAN architectures achieving FID scores of 10–30 on CIFAR-10. Replaced MNIST and Fashion MNIST experiments with results from the same architectures to ensure consistency (Figure 6).
  * **Detailed comparison with the FID score**:
    Added a discussion comparing FID scores from multiple perspectives (Section 6.2).
  * **Plots of GAN losses**:
    Added plots of GAN losses (Appendix G.5).
  * **Illustration of particle dynamics in real dataset**:
    Added an illustration of particle dynamics in CIFAR-10, focusing on their relationship with mode collapse.
  * **Justification for assumption on the discriminator in the collapse phase**:
    Added empirical verification on MNIST and Fashion MNIST (Appendix G.1).

---

### Decision · Action_Editor_YT27 · 2025-02-28

**Recommendation:** Accept as is

**Comment:**

This paper is a resubmission of a previously submitted work that has undergone significant improvements. Additionally, this version has been further refined based on discussions with the three reviewers.

This paper analyzes GAN training dynamics, introducing a novel approach to identifying key transitions during training using gradient-based tools. The authors successfully demonstrate how these methods can detect different epochs within GAN training, leading to meaningful practical implications.

Initially, the submission required improvement in several areas, including missing discussions on relevant related work, unclear claims, and insufficient empirical support. However, following a thorough revision process, the paper has significantly improved, effectively addressing all major concerns. The updated version now clearly articulates the transition of GAN training from a fitting phase to collapse, supported by well-established theoretical foundations. The revised manuscript is well-structured and presents a compelling argument for its contributions. The authors have refined the writing, enhanced empirical evaluations, and made the paper’s claims more transparent and well-supported.

The changes made during the revision were substantial and have ultimately strengthened the paper’s contributions. Given these improvements, I recommend its acceptance, as it meets the TMLR acceptance criteria and makes a meaningful contribution to the field.

**Audience:**

GANs remain a relevant approach in machine learning for generating samples, with widespread use in tasks such as super-resolution and image generation. A significant portion of the TMLR community is likely to find this paper valuable.

**Claims And Evidence:**

The authors propose a novel method to identify when a GAN’s generator transitions from fitting the distribution to experiencing mode collapse. Their analysis leverages both discriminator and generator gradients.

All three reviewers recommend accepting the paper, as the authors have collaborated with them to refine the manuscript to a level of sufficient quality and novelty. I concur with the reviewers regarding the quality of the final version.

The paper can be accepted as is, provided the authors incorporate their additions and address the reviewers’ comments and suggestions.